# Average Individual Fairness:
# Algorithms, Generalization and Experiments

**Michael Kearns**
University of Pennsylvania
mkearns@cis.upenn.edu

**Aaron Roth**
University of Pennsylvania
aaroth@cis.upenn.edu

**Saeed Sharifi-Malvajerdi**
University of Pennsylvania
saeedsh@wharton.upenn.edu

## Abstract

We propose a new family of fairness definitions for classification problems that combine some of the best properties of both statistical and individual notions of fairness. We posit not only a distribution over *individuals*, but also a distribution over (or collection of) *classification tasks*. We then ask that standard statistics (such as error or false positive/negative rates) be (approximately) equalized across *individuals*, where the rate is defined as an expectation over the classification tasks. Because we are no longer averaging over coarse groups (such as race or gender), this is a semantically meaningful individual-level constraint. Given a sample of individuals and problems, we design an oracle-efficient algorithm (i.e. one that is given access to any standard, fairness-free learning heuristic) for the fair empirical risk minimization task. We also show that given sufficiently many samples, the ERM solution generalizes in two directions: both to new individuals, and to new classification tasks, drawn from their corresponding distributions. Finally we implement our algorithm and empirically verify its effectiveness.

## 1 Introduction

The community studying fairness in machine learning has yet to settle on definitions. At a high level, existing definitional proposals can be divided into two groups: *statistical* fairness definitions and *individual* fairness definitions. Statistical fairness definitions partition individuals into "protected groups" (often based on race, gender, or some other binary protected attribute) and ask that some statistic of a classifier (error rate, false positive rate, positive classification rate, etc.) be approximately equalized across those groups. In contrast, individual definitions of fairness have no notion of "protected groups", and instead ask for constraints that bind on pairs of individuals. These constraints can have the semantics that "similar individuals should be treated similarly" (Dwork et al. (2012)), or that "less qualified individuals should not be preferentially favored over more qualified individuals" (Joseph et al. (2016)). Both families of definitions have serious problems, which we will elaborate on. But in summary, statistical definitions of fairness provide only very weak promises to individuals, and so do not have very strong semantics. Existing proposals for individual fairness guarantees, on the other hand, have very strong semantics, but have major obstacles to deployment, requiring strong assumptions on either the data generating process or on society's ability to instantiate an agreed-upon fairness metric.

Statistical definitions of fairness are the most popular in the literature, in large part because they can be easily checked and enforced on arbitrary data distributions. For example, a popular definition (Hardt et al. (2016); Kleinberg et al. (2017); Chouldechova (2017)) asks that a classifier's *false positive rate* should be equalized across the protected groups. This can sound attractive: in settings in which a positive classification leads to a bad outcome (e.g. incarceration), it is the *false positives* that are harmed by the errors of the classifier, and asking that the false positive rate be equalized across groups is asking that the harm caused by the algorithm should be proportionately spread across protected populations. But the meaning of this guarantee to an individual is limited, because

the word *rate* refers to an average over the population. To see why this limits the meaning of the guarantee, consider the example given in Kearns et al. (2018): imagine a society that is equally split between gender (Male, Female) and race (Blue, Green). Under the constraint that false positive rates be equalized across both race and gender, a classifier may incarcerate 100% of blue men and green women, and 0% of green men and blue women. This equalizes the false positive rate across all protected groups, but is cold comfort to any individual blue man and green woman. This effect isn't merely hypothetical — Kearns et al. (2018, 2019) showed similar effects when using off-the-shelf fairness constrained learning techniques on real datasets.

Individual definitions of fairness, on the other hand, can have strong individual level semantics. For example, the constraint imposed by Joseph et al. (2016, 2018) in online classification problems implies that the false positive rate must be equalized across all pairs of individuals who (truly) have negative labels. Here the word *rate* has been redefined to refer to an expectation over the randomness of the classifier, and there is no notion of protected groups. This kind of constraint provides a strong individual level promise that one's risk of being harmed by the errors of the classifier are no higher than they are for anyone else. Unfortunately, in order to non-trivially satisfy a constraint like this, it is necessary to make strong realizability assumptions.

## 1.1   Our results

We propose an alternative definition of individual fairness that avoids the need to make assumptions on the data generating process, while giving the learning algorithm more flexibility to satisfy it in non-trivial ways. We consider that in many applications each individual will be subject to decisions made by *many classification tasks* over a given period of time, not just one. For example, internet users are shown a large number of targeted ads over the course of their usage of a platform, not just one: the properties of the advertisers operating in the platform over a period of time are not known up front, but have some statistical regularities. Public school admissions in cities like New York are handled by a centralized match: students apply not just to one school, but to many, who can each make their own admissions decisions (Abdulkadiroğlu et al. (2005)). We model this by imagining that not only is there an unknown distribution $\mathcal{P}$ over individuals, but there is an unknown distribution $\mathcal{Q}$ over classification problems (each of which is represented by an unknown mapping from individual features to target labels). With this model in hand, we can now ask that the error rates (or false positive or negative rates) be equalized across all individuals — where now *rate* is defined as the average over *classification tasks* drawn from $\mathcal{Q}$ of the probability of a particular individual being incorrectly classified.

We then derive a new oracle-efficient algorithm for satisfying this guarantee in-sample, and prove novel generalization guarantees showing that the guarantees of our algorithm hold also out of sample. Oracle efficiency is an attractive framework in which to circumvent the worst-case hardness of even *unconstrained* learning problems, and focus on the *additional computational difficulty* imposed by fairness constraints. It assumes the existence of "oracles" that can solve weighted classification problems absent fairness constraints, and asks for efficient reductions from the fairness constrained learning problems to unconstrained problems. This has become a popular technique in the fair machine learning literature (see e.g. Agarwal et al. (2018); Kearns et al. (2018)) — and one that often leads to practical algorithms. The generalization guarantees we prove require the development of new techniques because they refer to generalization in *two orthogonal directions* — over both individuals and classification problems. Our algorithm is run on a sample of $n$ individuals sampled from $\mathcal{P}$ and $m$ problems sampled from $\mathcal{Q}$. It is given access to an oracle (in practice, implemented with a heuristic) for solving ordinary cost sensitive classification problems over some hypothesis space $\mathcal{H}$. The algorithm runs in polynomial time (it performs only elementary calculations except for calls to the learning oracle, and makes only a polynomial number of calls to the oracle) and returns a *mapping from problems to hypotheses* that have the following properties, so long as $n$ and $m$ are sufficiently large (polynomial in the VC-dimension of $\mathcal{H}$ and the desired error parameters): For any $\alpha$, with high probability over the draw of the $n$ individuals from $\mathcal{P}$ and the $m$ problems from $\mathcal{Q}$

1. *Accuracy*: the error rate (computed in expectation over new individuals $x \sim \mathcal{P}$ and new problems $f \sim \mathcal{Q}$) is within $O(\alpha)$ of the *optimal* mapping from problems to classifiers in $\mathcal{H}$, subject to the constraint that for every pair of individuals $x, x'$ in the support of $\mathcal{P}$, the error rates (or false positive or negative rates) (computed in expectation over problems $f \sim Q$) on $x$ and $x'$ differ by at most $\alpha$.

Table 1: Summary of notations for individuals vs. problems

|  | space | element | distribution | data set | sample size | empirical dist. |
|---|---|---|---|---|---|---|
| individual | $\mathcal{X}$ | $x \in \mathcal{X}$ | $\mathcal{P}$ | $X = \{x_i\}_{i=1}^n$ | $n$ | $\widehat{\mathcal{P}} = \mathcal{U}(X)$ |
| problem | $\mathcal{F}$ | $f \in \mathcal{F}$ | $\mathcal{Q}$ | $F = \{f_j\}_{j=1}^m$ | $m$ | $\widehat{\mathcal{Q}} = \mathcal{U}(F)$ |

2. *Fairness*: with probability $1 - \beta$ over the draw of new individuals $x, x' \sim \mathcal{P}$, the error rate (or false positive or negatives rates) of the output mapping (computed in expectation over problems $f \sim Q$) on $x$ will be within $O(\alpha)$ of that of $x'$.

The mapping from new classification problems to hypotheses that we find is derived from the dual variables of the linear program representing our empirical risk minimization task, and we crucially rely on the structure of this mapping to prove our generalization guarantees for new problems $f \sim Q$.

## 1.2 Additional related work

The literature on fairness in machine learning has become much too large to comprehensively summarize, but see Mitchell et al. (2018) for a recent survey. Here we focus on the most conceptually related work, which has aimed to bridge the gap between the immediate applicability of statistical definitions of fairness with the strong individual level semantics of individual notions of fairness. One strand of this literature focuses on the "metric fairness" definition first proposed by Dwork et al. (2012), and aims to ease the assumption that the learning algorithm has access to a task specific fairness metric. Kim et al. (2018a) imagine access to an oracle which can provide unbiased estimates to the metric distance between any pair of individuals, and show how to use this to satisfy a statistical notion of fairness representing "average metric fairness" over pre-defined groups. Gillen et al. (2018) study a contextual bandit learning setting in which a human judge points out metric fairness violations whenever they occur, and show that with this kind of feedback (under assumptions about consistency with a family of metrics), it is possible to quickly converge to the optimal fair policy. Yona and Rothblum (2018) consider a PAC-based relaxation of metric fair learning, and show that empirical metric-fairness generalizes to out-of-sample metric fairness. Another strand of this literature has focused on mitigating the problems that arise when statistical notions of fairness are imposed over coarsely defined groups, by instead asking for statistical notions of fairness over exponentially many or infinitely many groups with a well defined structure. This line includes Hébert-Johnson et al. (2018) (focusing on calibration), Kearns et al. (2018) (focusing on false positive and negative rates), and Kim et al. (2018b) (focusing on error rates).

## 2 Model and preliminaries

We model each individual in our framework by a vector of features $x \in \mathcal{X}$, and we let each learning problem [1] be represented by a binary function $f \in \mathcal{F}$ mapping $\mathcal{X}$ to $\{0, 1\}$. We assume probability measures $\mathcal{P}$ and $\mathcal{Q}$ over $\mathcal{X}$ and $\mathcal{F}$, respectively. In the training phase there is a *fixed* (across problems) set $X = \{x_i\}_{i=1}^n$ of $n$ individuals sampled independently from $\mathcal{P}$ for which we have available labels corresponding to $m$ tasks represented by $F = \{f_j\}_{j=1}^m$ drawn independently from $\mathcal{Q}$ [2]. Therefore, a training data set of $n$ individuals $X$ and $m$ learning tasks $F$ takes the form: $S = \{x_i, (f_j(x_i))_{j=1}^m\}_{i=1}^n$. We summarize the notations we use for individuals and problems in Table 1.

In general $\mathcal{F}$ will be unknown. We will aim to solve the (agnostic) learning problem over a hypothesis class $\mathcal{H}$, which need bear no relationship to $\mathcal{F}$. We will allow for randomized classifiers, which we model as learning over $\Delta(\mathcal{H})$, the probability simplex over $\mathcal{H}$. We assume throughout that $\mathcal{H}$ contains the constant classifiers $h^0$ and $h^1$ where $h^0(x) = 0$ and $h^1(x) = 1$ for all $x$. Unlike usual learning settings where the primary goal is to learn a single hypothesis $p \in \Delta(\mathcal{H})$, our objective is to learn a *mapping* $\psi \in \Delta(\mathcal{H})^{\mathcal{F}}$ that maps learning tasks $f \in \mathcal{F}$ represented as new labellings of the training data to hypotheses $p \in \Delta(\mathcal{H})$. We will therefore have to formally define the error rates incurred by a mapping $\psi$ and use them to formalize a learning task subject to our proposed fairness notion. For a mapping $\psi$, we write $\psi_f$ to denote the classifier corresponding to $f$ under the mapping, i.e.,

$\psi_f = \psi(f) \in \Delta(\mathcal{H})$. Notice in the training phase, there are only $m$ learning problems to be solved, and therefore, the corresponding empirical problem reduces to learning $m$ randomized classifiers. In general, learning $m$ specific classifiers for the training problems will not yield any generalizable rule mapping new problems to classifiers — but the specific algorithm we propose for empirical risk minimization will induce such a mapping, via a dual representation of the empirical risk minimizer.

**Definition 2.1** (Individual and Overall Error Rates). *For a mapping $\psi \in \Delta(\mathcal{H})^{\mathcal{F}}$ and distributions $\mathcal{P}$ and $\mathcal{Q}$: $\mathcal{E}(x, \psi; \mathcal{Q}) = \mathbb{E}_{f \sim \mathcal{Q}}\left[\mathbb{P}_{h \sim \psi_f}[h(x) \neq f(x)]\right]$ is the individual error rate of $x$ incurred by $\psi$ and $err(\psi; \mathcal{P}, \mathcal{Q}) = \mathbb{E}_{x \sim \mathcal{P}}[\mathcal{E}(x, \psi; \mathcal{Q})]$ is the overall error rate of $\psi$.*

In the body of this paper, we will focus on a fairness constraint that asks that the individual error rate should be approximately equalized across all individuals. In the supplement, we extend our techniques to equalizing false positive and negative rates across individuals.

**Definition 2.2** (Average Individual Fairness (AIF)). *We say a mapping $\psi \in \Delta(\mathcal{H})^{\mathcal{F}}$ satisfies "$(\alpha, \beta)$-AIF" (reads $(\alpha, \beta)$-approximate Average Individual Fairness) with respect to the distributions $(\mathcal{P}, \mathcal{Q})$ if there exists $\gamma \geq 0$ such that: $\mathbb{P}_{x \sim \mathcal{P}}\left(|\mathcal{E}(x, \psi; \mathcal{Q}) - \gamma| > \alpha\right) \leq \beta$.*

We briefly fix some notation: $\mathbb{1}[A]$ represents the indicator function of event $A$. For $n \in \mathbb{N}$, $[n] = \{1, 2, \ldots, n\}$. $\mathcal{U}(S)$ represents the uniform distribution over $S$. For a mapping $\psi : A \to B$ and $A' \subseteq A$, $\psi|_{A'}$ represents $\psi$ restricted to the domain $A'$. $d_{\mathcal{H}}$ denotes the VC dimension of the class $\mathcal{H}$. $CSC(\mathcal{H})$ denotes a *cost sensitive classification oracle* for $\mathcal{H}$:

**Definition 2.3** (Cost Sensitive Classification (CSC) in $\mathcal{H}$). *Let $D = \{x_i, c_i^1, c_i^0\}_{i=1}^n$ denote a data set of $n$ individuals $x_i$ where $c_i^1$ and $c_i^0$ are the costs of classifying $x_i$ as positive (1) and negative (0) respectively. Given $D$, the cost sensitive classification problem defined over $\mathcal{H}$ is the optimization problem: $\arg\min_{h \in \mathcal{H}} \sum_{i=1}^n \{c_i^1 h(x_i) + c_i^0 (1 - h(x_i))\}$. An oracle $CSC(\mathcal{H})$ takes $D = \{x_i, c_i^1, c_i^0\}_{i=1}^n$ as input and outputs the solution to the optimization problem. We use $CSC(\mathcal{H}; D)$ to denote the classifier returned by $CSC(\mathcal{H})$ on data set $D$. We say that an algorithm is* oracle efficient *if it runs in polynomial time given the ability to make unit-time calls to $CSC(\mathcal{H})$.*

## 3 Learning subject to AIF

In this section we first cast the learning problem subject to the AIF fairness constraints as the constrained optimization problem (1) and then develop an oracle efficient algorithm for solving its corresponding empirical risk minimization (ERM) problem (in the spirit of Agarwal et al. (2018)). In the coming sections we give a full analysis of the developed algorithm including its *in-sample* accuracy/fairness guarantees and define the mapping it induces from new problems to hypotheses, and finally establish *out-of-sample* bounds for this trained mapping.

---

**Fair Learning Problem subject to $(\alpha, 0)$-AIF**

$$\min_{\psi \in \Delta(\mathcal{H})^{\mathcal{F}}, \gamma \in [0,1]} \quad err(\psi; \mathcal{P}, \mathcal{Q})$$
$$\text{s.t. } \forall x \in \mathcal{X}: \quad |\mathcal{E}(x, \psi; \mathcal{Q}) - \gamma| \leq \alpha \tag{1}$$

---

**Definition 3.1** (OPT). *Consider the optimization problem (1). Given distributions $\mathcal{P}$ and $\mathcal{Q}$, and fairness approximation parameter $\alpha$, we denote the optimal solutions of (1) by $\psi^\star(\alpha; \mathcal{P}, \mathcal{Q})$ and $\gamma^\star(\alpha; \mathcal{P}, \mathcal{Q})$, and the value of the objective function at these optimal points by $OPT(\alpha; \mathcal{P}, \mathcal{Q})$.*

We will use OPT as the benchmark with respect to which we evaluate the accuracy of our trained mapping. It is worth noticing that the optimization problem (1) has a nonempty set of feasible solutions for every $\alpha$ and all distributions $\mathcal{P}$ and $\mathcal{Q}$ because the following point is always feasible: $\gamma = 0.5$ and $\psi_f = 0.5h^0 + 0.5h^1$ (i.e. random classification) for all $f \in \mathcal{F}$ where $h^0$ and $h^1$ are all-zero and all-one constant classifiers.

### 3.1 The empirical fair learning problem

We start to develop our algorithm by defining the *empirical* version of (1) for a given training data set of $n$ individuals $X = \{x_i\}_{i=1}^n$ and $m$ learning problems $F = \{f_j\}_{j=1}^m$. We will formulate the

empirical problem as finding a *restricted* mapping $\psi|_F$ by which we mean the domain of the mapping is restricted to the training set $F \subseteq \mathcal{F}$. We will later see how the dynamics of our proposed algorithm allows us to extend the restricted mapping to a mapping from the entire space $\mathcal{F}$. We slightly change notation and represent a restricted mapping $\psi|_F$ explicitly by a vector $\boldsymbol{p} = (p_1, \ldots, p_m) \in \Delta(\mathcal{H})^m$ of randomized classifiers where $p_j \in \Delta(\mathcal{H})$ corresponds to $f_j \in F$. Using the empirical versions of the individual and the overall error rates incurred by the mapping $\boldsymbol{p}$ (see Definition 2.1), we cast the empirical fair learning problem as the constrained optimization problem (2).

---

**Empirical Fair Learning Problem**

$$\min_{\boldsymbol{p} \in \Delta(\mathcal{H})^m, \gamma \in [0,1]} \quad \text{err}\left(\boldsymbol{p}; \widehat{\mathcal{P}}, \widehat{\mathcal{Q}}\right)$$

$$\text{s.t. } \forall i \in \{1, \ldots, n\}: \quad \left|\mathcal{E}\left(x_i, \boldsymbol{p}; \widehat{\mathcal{Q}}\right) - \gamma\right| \le 2\alpha \tag{2}$$

---

We use the dual perspective of constrained optimization to reduce the fair learning task (2) to a two-player game between a "Learner" (primal player) and an "Auditor" (dual player). Towards deriving the Lagrangian of (2), we first rewrite its constraints in $\boldsymbol{r}\left(\boldsymbol{p}, \gamma; \widehat{\mathcal{Q}}\right) \le 0$ form where

$$\boldsymbol{r}\left(\boldsymbol{p}, \gamma; \widehat{\mathcal{Q}}\right) = \begin{bmatrix} \mathcal{E}\left(x_i, \boldsymbol{p}; \widehat{\mathcal{Q}}\right) - \gamma - 2\alpha \\ \gamma - \mathcal{E}\left(x_i, \boldsymbol{p}; \widehat{\mathcal{Q}}\right) - 2\alpha \end{bmatrix}_{i=1}^n \in \mathbb{R}^{2n} \tag{3}$$

represents the "fairness violations" of the pair $(\boldsymbol{p}, \gamma)$ in one single vector. Let the corresponding dual variables for $\boldsymbol{r}$ be represented by $\boldsymbol{\lambda} = \left[\lambda_i^+, \lambda_i^-\right]_i \in \Lambda$, where $\Lambda = \{\boldsymbol{\lambda} \in \mathbb{R}_+^{2n} \mid ||\boldsymbol{\lambda}||_1 \le B\}$. Note we place an upper bound $B$ on the $\ell_1$-norm of $\boldsymbol{\lambda}$ in order to reason about the convergence of our proposed algorithm. $B$ will eventually factor into both the run-time and the approximation guarantees of our solution. Using Equation (3) and the introduced dual variables, we have that the Lagrangian of (2) is $\mathcal{L}\left(\boldsymbol{p}, \gamma, \boldsymbol{\lambda}\right) = \text{err}\left(\boldsymbol{p}; \widehat{\mathcal{P}}, \widehat{\mathcal{Q}}\right) + \boldsymbol{\lambda}^T \boldsymbol{r}\left(\boldsymbol{p}, \gamma; \widehat{\mathcal{Q}}\right)$. We therefore consider solving:

$$\min_{\boldsymbol{p} \in \Delta(\mathcal{H})^m, \gamma \in [0,1]} \max_{\boldsymbol{\lambda} \in \Lambda} \mathcal{L}\left(\boldsymbol{p}, \gamma, \boldsymbol{\lambda}\right) = \max_{\boldsymbol{\lambda} \in \Lambda} \min_{\boldsymbol{p} \in \Delta(\mathcal{H})^m, \gamma \in [0,1]} \mathcal{L}\left(\boldsymbol{p}, \gamma, \boldsymbol{\lambda}\right) \tag{4}$$

where strong duality holds because $\mathcal{L}$ is linear in its arguments and the domains of $(\boldsymbol{p}, \gamma)$ and $\boldsymbol{\lambda}$ are convex and compact (Sion (1958)). From a game theoretic perspective, the solution to this minmax problem can be seen as an equilibrium of a zero-sum game between two players. The primal player (Learner) has strategy space $\Delta(\mathcal{H})^m \times [0, 1]$ while the dual player (Auditor) has strategy space $\Lambda$, and given a pair of chosen strategies $(\boldsymbol{p}, \gamma, \boldsymbol{\lambda})$, the Lagrangian $\mathcal{L}\left(\boldsymbol{p}, \gamma, \boldsymbol{\lambda}\right)$ represents how much the Learner has to pay to the Auditor — i.e. it defines the payoff function of a zero sum game in which the Learner is the minimization player, and the Auditor is the maximization player. Using no regret dynamics, an approximate equilibrium of this zero-sum game can be found in an iterative framework. In each iteration, we let the dual player run the *exponentiated gradient descent* algorithm and the primal player *best respond*. The best response problem of the Learner can be decoupled into $(m + 1)$ separate minimization problems and that in particular, the optimal classifiers $\boldsymbol{p}$ can be viewed as the solutions to $m$ *weighted classification problems* in $\mathcal{H}$ where all $m$ problems share the weights $\boldsymbol{w} = [\lambda_i^+ - \lambda_i^-]_i \in \mathbb{R}^n$ over the training individuals. We write the best response of the Learner in Subroutine 1 where we use the oracle $CSC(\mathcal{H})$ (see Definition 2.3) to solve the weighted classification problems. See the supplementary file for the detailed derivation.

---

**Subroutine 1: BEST**– best response of the Learner in the AIF setting

---

**Input:** dual weights $\boldsymbol{w} = [\lambda_i^+ - \lambda_i^-]_{i=1}^n \in \mathbb{R}^n$, training examples $S = \left\{x_i, (f_j(x_i))_{j=1}^m \right\}_{i=1}^n$

**for** $j = 1, \ldots, m$ **do**

$\quad c_i^1 \leftarrow (w_i + 1/n)(1 - f_j(x_i))$ and $c_i^0 \leftarrow (w_i + 1/n)f_j(x_i)$ for $i \in [n]$.

$\quad h_j \leftarrow CSC\left(\mathcal{H}; D\right)$ where $D = \{x_i, c_i^1, c_i^0\}_{i=1}^n$.

**end**

**Output:** $\left(\boldsymbol{h} = (h_1, h_2, \ldots, h_m), \gamma = \mathbb{1}\left[\sum_{i=1}^n w_i > 0\right]\right)$

---

## 3.2 Algorithm implementation and in-sample guarantees

In Algorithm 2 (**AIF-Learn**), with a slight deviation from what we described in the previous subsection, we implement the proposed algorithm. The deviation arises when the Auditor updates the dual variables $\boldsymbol{\lambda}$ in each round, and is introduced in the service of arguing for generalization. To counteract the inherent adaptivity of the algorithm (which makes the quantities estimated at each round data dependent), at each round $t$ of the algorithm, we draw a *fresh* batch of $m_0$ problems. From another viewpoint – which is the way the algorithm is actually implemented – similar to usual batch learning models we assume we have a training set $F$ of $m$ learning problems upfront. However, in our proposed algorithm that runs for $T$ iterations, we partition $F$ into $T$ equally-sized ($m_0$) subsets $\{F_t\}_{t=1}^T$ uniformly at random and use only the batch $F_t$ at round $t$ to update $\boldsymbol{\lambda}$. Without loss of generality and to avoid technical complications, we assume $|F_t| = m_0 = m/T$ is a natural number. This is represented in Algorithm 2 by writing $\widehat{\mathcal{Q}}_t = \mathcal{U}(F_t)$ for the uniform distribution over the batch of problems $F_t$, and $\boldsymbol{h}_t|_{F_t}$ for the associated classifiers for $F_t$.

Notice **AIF-Learn** takes as input an approximation parameter $\nu \in [0,1]$ which will quantify how close the output of the algorithm is to an equilibrium of the introduced game, and it will accordingly propagate to the accuracy bounds. One important aspect of **AIF-Learn** is that it maintains a weight vector $\boldsymbol{w}_t \in \mathbb{R}^n$ over the training individuals $X$ and that each $\widehat{p}_j$ learned by our algorithm is in fact an average over $T$ classifiers where classifier $t$ is the solution to a CSC problem on $X$ weighted by $\boldsymbol{w}_t$. As a consequence, we propose to extend the learned restricted mapping $\widehat{\boldsymbol{p}}$ to a mapping $\widehat{\psi} = \widehat{\psi}(X, \widehat{W})$ that takes *any* problem $f \in \mathcal{F}$ as input (represented to $\widehat{\psi}$ by the labels it induces on the training data), uses the individuals $X$ along with the set of weights $\widehat{W}$ to solve $T$ CSC problems in a similar fashion, and outputs the average of the learned classifiers denoted by $\widehat{\psi}_f \in \Delta(\mathcal{H})$. This extension is consistent with $\widehat{\boldsymbol{p}}$ in the sense that $\widehat{\psi}$ restricted to $F$ will be exactly the $\widehat{\boldsymbol{p}}$ output by our algorithm. The pseudocode for $\widehat{\psi}$ (output by **AIF-Learn**) is written in detail in Mapping 3.

---

**Algorithm 2: AIF-Learn** – learning subject to AIF

---

**Input:** fairness parameter $\alpha$, approximation parameter $\nu$, data $X = \{x_i\}_{i=1}^n$ and $F = \{f_j\}_{j=1}^m$

$B \leftarrow \frac{1+2\nu}{\alpha}, \;\; T \leftarrow \frac{16B^2(1+2\alpha)^2 \log(2n+1)}{\nu^2}, \;\; \eta \leftarrow \frac{\nu}{4(1+2\alpha)^2 B}, \;\; m_0 \leftarrow \frac{m}{T}, \;\; S \leftarrow \left\{x_i, (f_j(x_i))_j\right\}_i$

Partition $F$: $\{F_t\}_{t=1}^T$ where $|F_t| = m_0$.
$\boldsymbol{\theta}_1 \leftarrow \boldsymbol{0}$
**for** $t = 1, \ldots, T$ **do**
    $\lambda_{i,t}^\bullet \leftarrow B \frac{\exp(\theta_{i,t}^\bullet)}{1 + \sum_{i',\bullet'} \exp(\theta_{i',t}^{\bullet'})}$ for $i \in [n]$ and $\bullet \in \{+,-\}$
    $\boldsymbol{w}_t \leftarrow [\lambda_{i,t}^+ - \lambda_{i,t}^-]_{i=1}^n \in \mathbb{R}^n$
    $(\boldsymbol{h}_t, \gamma_t) \leftarrow \textbf{BEST}(\boldsymbol{w}_t; S)$
    $\boldsymbol{\theta}_{t+1} \leftarrow \boldsymbol{\theta}_t + \eta \cdot \boldsymbol{r}\left(\boldsymbol{h}_t|_{F_t}, \gamma_t; \widehat{\mathcal{Q}}_t\right)$
**end**
$\widehat{\gamma} \leftarrow \frac{1}{T} \sum_{t=1}^T \gamma_t, \quad \widehat{\boldsymbol{p}} \leftarrow \frac{1}{T} \sum_{t=1}^T \boldsymbol{h}_t, \quad \widehat{\boldsymbol{\lambda}} \leftarrow \frac{1}{T} \sum_{t=1}^T \boldsymbol{\lambda}_t, \quad \widehat{W} \leftarrow \{\boldsymbol{w}_t\}_{t=1}^T$

**Output:** average plays $\left(\widehat{\boldsymbol{p}}, \widehat{\gamma}, \widehat{\boldsymbol{\lambda}}\right)$, mapping $\widehat{\psi} = \widehat{\psi}\left(X, \widehat{W}\right)$ (see Mapping 3)

---

We defer a complete in-sample analysis of Algorithm 2 to the supplementary file. At a high level, we start by establishing the regret bound of the Auditor and choosing $T$ and $\eta$ such that her regret $\leq \nu$. There will be an extra $\widetilde{O}(\sqrt{1/m_0})$ term originating from a high probability (Chernoff) bound in the regret of the Auditor because she is using a batch of only $m_0$ randomly selected problems to update the fairness violation vector $\boldsymbol{r}$. We therefore have to assume $m_0$ is sufficiently large to control the regret. Once the regret bound is established, we can show that the average played strategies $(\widehat{\boldsymbol{p}}, \widehat{\gamma}, \widehat{\boldsymbol{\lambda}})$ output by Algorithm 2 forms a $\nu$-approximate equilibrium of the game by which we mean: neither player would gain more than $\nu$ if they deviated from these proposed strategies. Finally we can take this guarantee and turn it into accuracy and fairness guarantees of the pair $(\widehat{\boldsymbol{p}}, \widehat{\gamma})$ with respect to the empirical distributions $(\widehat{\mathcal{P}}, \widehat{\mathcal{Q}})$, which results in Theorem 3.1.

| **Mapping 3:** $\widehat{\psi}\,(X,\widehat{W})$ – pseudocode |
| --- |
| **Input:** $f \in \mathcal{F}$ (represented as $\{f(x_i)\}_{i=1}^n$) |
| **for** $t = 1, \ldots, T$ **do** |
| $\quad c_i^1 \leftarrow (w_{i,t} + 1/n)(1 - f(x_i))$ for $i \in [n]$. |
| $\quad c_i^0 \leftarrow (w_{i,t} + 1/n)f(x_i)$ for $i \in [n]$. |
| $\quad D \leftarrow \{x_i, c_i^1, c_i^0\}_{i=1}^n$. |
| $\quad h_{f,\boldsymbol{w}_t} \leftarrow CSC\,(\mathcal{H}; D)$ |
| **end** |
| **Output:** $\widehat{\psi}_f = \frac{1}{T}\sum_{t=1}^T h_{f,\boldsymbol{w}_t} \in \Delta(\mathcal{H})$ |

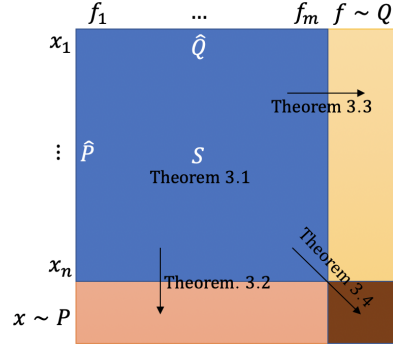

Figure 1: Illustration of generalization directions.

**Theorem 3.1** (In-sample Accuracy and Fairness). *Suppose $m_0 \geq O\left(\log\left(nT/\delta\right)/\alpha^2\nu^2\right)$. Let $(\widehat{\boldsymbol{p}},\widehat{\gamma})$ be the output of Algorithm 2 and let $(\boldsymbol{p},\gamma)$ be any feasible pair of variables for (2). We have that with probability $1 - \delta$, $err\,(\widehat{\boldsymbol{p}};\widehat{\mathcal{P}},\widehat{\mathcal{Q}}) \leq err\,(\boldsymbol{p};\widehat{\mathcal{P}},\widehat{\mathcal{Q}}) + 2\nu$, and that $\widehat{\boldsymbol{p}}$ satisfies $(3\alpha, 0)$-AIF with respect to the empirical distributions $(\widehat{\mathcal{P}},\widehat{\mathcal{Q}})$. In other words, for all $i \in [n]$, $|\mathcal{E}(x_i,\widehat{\boldsymbol{p}};\widehat{\mathcal{Q}}) - \widehat{\gamma}| \leq 3\alpha$.*

### 3.3 Generalization theorems

When it comes to out-of-sample performance in our framework, unlike in usual learning settings, there are two distributions we need to reason about: the individual distribution $\mathcal{P}$ and the problem distribution $\mathcal{Q}$ (see Figure 1 for a visual illustration of generalization directions in our framework). We need to argue that $\widehat{\psi}$ induces a mapping that is accurate with respect to $\mathcal{P}$ and $\mathcal{Q}$, and is fair for almost every individual $x \sim \mathcal{P}$, where fairness is defined with respect to the true problem distribution $\mathcal{Q}$. Given these two directions for generalization, we state our generalization guarantees in three steps visualized by arrows in Figure 1. First, in Theorem 3.2, we fix the empirical distribution of the problems $\widehat{\mathcal{Q}}$ and show that the output $\widehat{\psi}$ of Algorithm 2 is accurate and fair with respect to the underlying individual distribution $\mathcal{P}$ as long as $n$ is sufficiently large. Second, in Theorem 3.3, we fix the empirical distribution of individuals $\widehat{\mathcal{P}}$ and consider generalization along the underlying problem generating distribution $\mathcal{Q}$. It will follow from the dynamics of the algorithm that the mapping $\widehat{\psi}$ will remain accurate and fair with respect to $\mathcal{Q}$. We will eventually put these pieces together in Theorem 3.4 and argue that $\widehat{\psi}$ is accurate and fair with respect to the underlying distributions $(\mathcal{P}, \mathcal{Q})$ simultaniously, given that both $n$ and $m$ are large enough. We will use OPT (see Definition 3.1) as a benchmark for the accuracy of the mapping $\widehat{\psi}$. See the supplementary file for detailed proofs.

**Theorem 3.2** (Generalization over $\mathcal{P}$). *Let $0 < \delta < 1$. Let $\widehat{\psi}$ and $\widehat{\gamma}$ be the outputs of Algorithm 2 and suppose $n \geq \widetilde{O}\left((m\,d_{\mathcal{H}} + \log\left(1/\nu^2\delta\right))/\alpha^2\beta^2\right)$. We have that with probability $1 - 5\delta$, the mapping $\widehat{\psi}$ satisfies $(5\alpha, \beta)$-AIF with respect to the distributions $(\mathcal{P},\widehat{\mathcal{Q}})$, i.e., $\mathbb{P}_{x \sim \mathcal{P}}\left(|\mathcal{E}(x,\widehat{\psi};\widehat{\mathcal{Q}}) - \widehat{\gamma}| > 5\alpha\right) \leq \beta$ and that $err\,(\widehat{\psi};\mathcal{P},\widehat{\mathcal{Q}}) \leq OPT\,(\alpha;\mathcal{P},\widehat{\mathcal{Q}}) + O\,(\nu) + O\,(\alpha\beta)$.*

**Theorem 3.3** (Generalization over $\mathcal{Q}$). *Let $0 < \delta < 1$. Let $\widehat{\psi}$ and $\widehat{\gamma}$ be the outputs of Algorithm 2 and suppose $m \geq \widetilde{O}\left(\log\left(n\right)\log\left(n/\delta\right)/\nu^4\alpha^4\right)$. We have that with probability $1 - 6\delta$, the mapping $\widehat{\psi}$ satisfies $(4\alpha, 0)$-AIF with respect to the distributions $(\widehat{\mathcal{P}}, \mathcal{Q})$, i.e., $\mathbb{P}_{x \sim \widehat{\mathcal{P}}}\left(|\mathcal{E}(x,\widehat{\psi};\mathcal{Q}) - \widehat{\gamma}| > 4\alpha\right) = 0$ and that $err\,(\widehat{\psi};\widehat{\mathcal{P}},\mathcal{Q}) \leq OPT\,(\alpha;\widehat{\mathcal{P}},\mathcal{Q}) + O\,(\nu)$.*

**Theorem 3.4** (Simultaneous Generalization over $\mathcal{P}$ and $\mathcal{Q}$). *Let $0 < \delta < 1$. Let $\widehat{\psi}$ and $\widehat{\gamma}$ be the outputs of Algorithm 2 and suppose $n \geq \widetilde{O}\left((m\,d_{\mathcal{H}} + \log\left(1/\nu^2\delta\right))/\alpha^2\beta^2\right)$ and $m \geq \widetilde{O}\left(\log\left(n\right)\log\left(n/\delta\right)/\nu^4\alpha^4\right)$. We have that with probability $1-12\delta$, the mapping $\widehat{\psi}$ satisfies $(6\alpha, 2\beta)$-AIF with respect to the distributions $(\mathcal{P}, \mathcal{Q})$, i.e., $\mathbb{P}_{x \sim \mathcal{P}}\left(|\mathcal{E}(x,\widehat{\psi};\mathcal{Q}) - \widehat{\gamma}| > 6\alpha\right) \leq 2\beta$ and that $err\,(\widehat{\psi};\mathcal{P},\mathcal{Q}) \leq OPT\,(\alpha;\mathcal{P},\mathcal{Q}) + O\,(\nu) + O\,(\alpha\beta)$.*

Note that the bounds on $n$ and $m$ in Theorem 3.4 are mutually dependent: $n$ must be *linear* in $m$, but $m$ need only be *logarithmic* in $n$, and so both bounds can be simultaneously satisfied with sample complexity that is only polynomial in the parameters of the problem.

# 4 Experimental evaluation

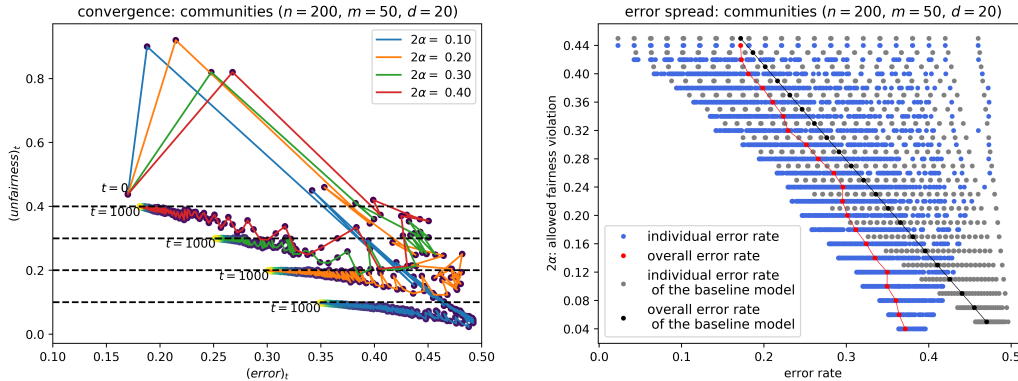

Figure 2: (a) Error-unfairness trajectory plots illustrating the convergence of algorithm **AIF-Learn**. (b) In-sample error-unfairness tradeoffs and individual errors for **AIF-Learn** vs. the baseline model: simple mixtures of the error-optimal model and random classification. Gray dots are shifted upwards slightly to avoid occlusions.

We have implemented the **AIF-Learn** algorithm and conclude with a brief experimental demonstration of its practical efficacy using the Communities and Crime dataset[3], which contains U.S. census records with demographic information at the neighborhood level. To obtain a challenging instance of our multi-problem framework, we treated each of the first $n = 200$ neighborhoods as the "individuals" in our sample, and binarized versions of the first $m = 50$ variables as distinct prediction problems. Another $d = 20$ of the variables were used as features for learning. For the base learning oracle assumed by **AIF-Learn**, we used a linear threshold learning heuristic that has worked well in other oracle-efficient reductions (Kearns et al. (2018)).

Despite the absence of worst-case guarantees for the linear threshold heuristic, **AIF-Learn** seems to empirically enjoy the strong convergence properties suggested by the theory. In Figure 2(a) we show trajectory plots of the learned model's error ($x$ axis) versus its fairness violation (variation in cross-problem individual error rates, $y$ axis) over 1000 iterations of the algorithm for varying values of the allowed fairness violation $2\alpha$ (dashed horizontal lines). In each case we see the trajectory eventually converge to a point which saturates the fairness constraint with the optimal error.

In Figure 2(b) we provide a more detailed view of the behavior and performance of **AIF-Learn**. The $x$ axis measures error rates, while the $y$ axis measures the allowed fairness violation. For each value of the allowed fairness violation $2\alpha$ (which is the allowed gap between the smallest and largest individual errors on input $\alpha$), there is a horizontal row of 200 blue dots showing the error rates for each individual, and a single red dot representing the overall average of those individual error rates. As expected, for large $\alpha$ (weak or no fairness constraint), the overall error rate is lowest, but the spread of individual error rates (unfairness) is greatest. As $\alpha$ is decreased, the spread of individual error rates is greatly narrowed, at a cost of greater overall error.

A trivial way of achieving zero variability in individual error rates is to make all predictions randomly. So as a baseline comparison for **AIF-Learn**, the gray dots in Figure 2(b) show the individual error rates achieved by different mixtures of the unconstrained error-optimal model with random classifications, with a black dot representing the overall average of these rates. When the weight on random classification is low (weak or no fairness, top row of gray dots), the overall error is lowest and the individual variation (unfairness) is highest. As we increase the weight on random classification, variation or unfairness decreases and the overall error gets worse. It is clear from the figure that **AIF-Learn** is considerably outperforming this baseline, both in terms of the average errors (red vs. black lines) and the individual errors (blue vs. gray dots).

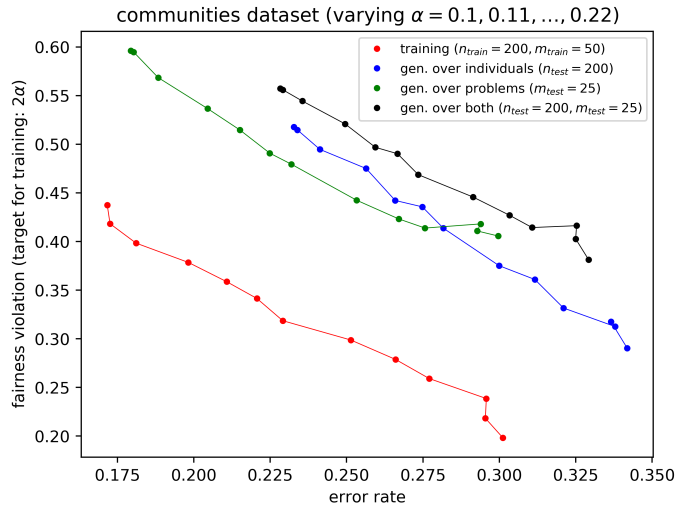

Figure 3: Pareto frontier of error and fairness violation rates on training and test data sets.

Finally we present out-of-sample performance of **AIF-Learn** in Figure 3. To be consistent with in-sample results reported in Figure 2(b), for each value of $\alpha$, we trained a mapping on exactly the same subset of the Communities and Crime data set ($n = 200$ individuals, $m = 50$ problems) that we used before. Thus the red curve labelled "training" in Figure 3 is the same as the red curve appearing in Figure 2(b). We used a completely fresh holdout consisting of $n = 200$ individuals and $m = 25$ problems (binarized features from the dataset that weren't previously used) to evaluate our generalization performance over both individuals and problems, in terms of both accuracy and fairness violation. Similar to the presentation of generalization theorems in Section 3.3, we demonstrate experimental evaluation of generalization in three steps. The blue and green curves in Figure 3 represent generalization results over individuals (test data: test individuals and training problems) and problems (test data: training individuals and test problems) respectively. The black curve represent generalization across both individuals and problems where test individuals and test problems were used to evaluate the performance of the trained models.

Two things stand out from Figure 3:

1. As predicted by the theory, our test curves track our training curves, but with higher error and unfairness. In particular, the ordering of the models (each corresponds to one $\alpha$) on the Pareto frontier is the same in testing as in training, meaning that the training curve can indeed be used to manage the trade-off out-of-sample as well.

2. The gap in error is substantially smaller than would be predicted by our theory: since our training data set is so small, our theoretical guarantees are vacuous, but all points plotted in our test Pareto curves are non-trivial in terms of both accuracy and fairness. Presumably the gap in error would narrow on larger training data sets.

We have additional experimental results on a synthetic data set in the supplement.

## Footnotes

[1]We will use the terms: problem, task, and labeling interchangeably.

[2]Throughout we will use subscript $i$ to denote individuals and $j$ to denote learning problems.

[3]Described in detail and available for download at `http://archive.ics.uci.edu/ml/datasets/communities+and+crime`

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
