[Supplementary Material · supplement.pdf]

# Supplement

This supplementary document is divided into four sections.

- In section A we provide missing parts from the analysis of our first algorithm **AIF-Learn**, including the derivation of the best response of the Learner in the AIF setting, as well as the algorithm's complete in-sample analysis.

- In section B we introduce the "FPAIF" notion of fairness which essentially asks for equalizing *false positive rates* across individuals. We develop an algorithm called **FPAIF-Learn** for learning mappings subject to the FPAIF notion of fairness and provide its analysis and generalization theorems. Our method will obviously generalize to learning subject to *false negative rate parity* and so we avoid replicating the results for false negative rates.

- In section C we will present all the proofs. Most of the proofs of the FPAIF section – especially the proofs for generalization theorems – will be similar to their counterparts in the AIF section and we will avoid stating them.

- In section D we provide additional experimental results along with more details on implementation of the algorithm **AIF-Learn**.

## A  Learning subject to AIF: missing parts

### A.1  BEST: The Learner's Best Response

In this subsection we show how the Learner's best response problem can be decoupled into $(m + 1)$ minimization problems. Consider the described iterative algorithm where in each iteration the Auditor uses the exponentiated gradient descent and the Learner uses its best response. We will formally describe the best response problem of the Learner in this subsection and will summarize it in a subroutine called **BEST**. At iteration $t$ of the algorithm, the Learner is given $\boldsymbol{\lambda}_t \in \Lambda$ from the Auditor and they want to solve the following minimization problem.

$$
\underset{\boldsymbol{p} \in \Delta(\mathcal{H})^m, \gamma \in [0,1]}{\arg\min} \quad \mathcal{L}\left(\boldsymbol{p}, \gamma, \boldsymbol{\lambda}_t\right)
$$

$$
\equiv \underset{\boldsymbol{p} \in \Delta(\mathcal{H})^m, \gamma \in [0,1]}{\arg\min} \quad \mathrm{err}\left(\boldsymbol{p}; \widehat{\mathcal{P}}, \widehat{\mathcal{Q}}\right) + \sum_{i=1}^{n} \left\{ \lambda_{i,t}^+ \left( \mathcal{E}\left(x_i, \boldsymbol{p}; \widehat{\mathcal{Q}}\right) - \gamma \right) + \lambda_{i,t}^- \left( \gamma - \mathcal{E}\left(x_i, \boldsymbol{p}; \widehat{\mathcal{Q}}\right) \right) \right\}
$$

$$
\equiv \underset{\boldsymbol{p} \in \Delta(\mathcal{H})^m, \gamma \in [0,1]}{\arg\min} \quad \frac{1}{n} \sum_{i=1}^{n} \mathcal{E}\left(x_i, \boldsymbol{p}; \widehat{\mathcal{Q}}\right) + \sum_{i=1}^{n} \left\{ \lambda_{i,t}^+ \left( \mathcal{E}\left(x_i, \boldsymbol{p}; \widehat{\mathcal{Q}}\right) - \gamma \right) + \lambda_{i,t}^- \left( \gamma - \mathcal{E}\left(x_i; \boldsymbol{p}; \widehat{\mathcal{Q}}\right) \right) \right\}
$$

$$
\equiv \underset{\boldsymbol{p} \in \Delta(\mathcal{H})^m, \gamma \in [0,1]}{\arg\min} \quad \sum_{i=1}^{n} \left\{ \lambda_{i,t}^- - \lambda_{i,t}^+ \right\} \gamma + \sum_{i=1}^{n} \left( \frac{1}{n} + \lambda_{i,t}^+ - \lambda_{i,t}^- \right) \mathcal{E}\left(x_i, \boldsymbol{p}; \widehat{\mathcal{Q}}\right)
$$

$$
\equiv \underset{\boldsymbol{p} \in \Delta(\mathcal{H})^m, \gamma \in [0,1]}{\arg\min} \quad \sum_{i=1}^{n} \left\{ \lambda_{i,t}^- - \lambda_{i,t}^+ \right\} \gamma + \frac{1}{m} \sum_{j=1}^{m} \left\{ \sum_{i=1}^{n} \left( \frac{1}{n} + \lambda_{i,t}^+ - \lambda_{i,t}^- \right) \underset{h_j \sim p_j}{\mathbb{P}} \left[ h_j(x_i) \neq f_j(x_i) \right] \right\}
$$

Therefore, the minimization problem of the Learner gets nicely decoupled into $(m + 1)$ minimization problems. Let $w_{i,t} = \lambda_{i,t}^+ - \lambda_{i,t}^-$ for all $i$, and accordingly, let $\boldsymbol{w}_t = [w_{1,t}, \ldots, w_{n,t}]^\top \in \mathbb{R}^n$. First,

the optimal value for $\gamma$ is chosen according to

$$\gamma_t = \mathbb{1}\left[\sum_{i=1}^{n} w_{i,t} > 0\right] \tag{1}$$

And for learning problem $j$, the following minimization problem must be solved.

$$\underset{p_j \in \Delta(\mathcal{H})}{\arg\min} \sum_{i=1}^{n} (1/n + w_{i,t}) \underset{h_j \sim p_j}{\mathbb{P}} [h_j(x_i) \neq f_j(x_i)] \equiv \underset{h_j \in \mathcal{H}}{\arg\min} \sum_{i=1}^{n} (1/n + w_{i,t}) \mathbb{1}[h_j(x_i) \neq f_j(x_i)]$$

where the equivalence holds since without loss of generality, the Learner can always choose to put all the probability mass on a single classifier because of linearity. This minimization problem is now exactly a weighted classification problem. Since we work with cost sensitive classification (CSC) problems in this paper and assume access to CSC oracles, we further reduce the weighted classification problem to a CSC problem that can be solved by a call to the cost sensitive classification oracle for $\mathcal{H}$ ($CSC(\mathcal{H})$). For problem $j \in [m]$, let

$$c_{i,j}^1 = (w_{i,t} + 1/n)(1 - f_j(x_i)) \quad , \quad c_{i,j}^0 = (w_{i,t} + 1/n)f_j(x_i)$$

be the costs associated with individual $i \in [n]$. Observe that the above weighted classification problem can be now casted as the following CSC problem.

$$h_j = \underset{h \in \mathcal{H}}{\arg\min} \sum_{i=1}^{n} c_{i,j}^1 \, h(x_i) + c_{i,j}^0 \, (1 - h(x_i)) \tag{2}$$

To sum up, at iteration $t$ of the algorithm the Auditor first updates the dual variable $\boldsymbol{\lambda}_t$ (or correspondingly the vector of weights $\boldsymbol{w}_t$) and then the Learner picks $\gamma_t = \mathbb{1}\left[\sum_{i=1}^{n} w_{i,t} > 0\right]$ and solves $m$ cost sensitive classification problems casted in 2 by calling the cost sensitive classification oracle $CSC(\mathcal{H})$ for all $1 \leq j \leq m$. We have the best response of the Learner written in Subroutine 1. This subroutine will be called in each iteration of the final AIF learning algorithm.

---

**Subroutine 1: BEST**– best response of the Learner in the AIF setting

---

**Input:** dual weights $\boldsymbol{w} \in \mathbb{R}^n$, training examples $S = \left\{x_i, (f_j(x_i))_{j=1}^{m}\right\}_{i=1}^{n}$

$\gamma \leftarrow \mathbb{1}\left[\sum_{i=1}^{n} w_i > 0\right]$
**for** $j = 1, \dots, m$ **do**
    $c_i^1 \leftarrow (w_i + 1/n)(1 - f_j(x_i))$ for $i \in [n]$
    $c_i^0 \leftarrow (w_i + 1/n)f_j(x_i)$ for $i \in [n]$
    $D = \{x_i, c_i^1, c_i^0\}_{i=1}^{n}$
    $h_j \leftarrow CSC(\mathcal{H}; D)$
**end**
$\boldsymbol{h} \leftarrow (h_1, h_2, \dots, h_m)$

**Output:** $\left(\boldsymbol{h}, \gamma\right)$

---

### A.2 The in-sample analysis of AIF-Learn algorithm in full detail

We start the analysis of **AIF-Learn** by establishing the regret bound of the Auditor over $T$ rounds. The regret bound will help us pick the number of iterations $T$ and the learning rate $\eta$ so that the Auditor has sufficiently small regret (bounded by $\nu$). Notice the Learner uses its best response in each round of the algorithm which implies that it has zero regret.

**Lemma A.1** (Regret of the Auditor). *Let $0 < \delta < 1$. Let $\{\boldsymbol{\lambda}_t\}_{t=1}^{T}$ be the sequence of exponentiated gradient descent plays (with learning rate $\eta$) by the Auditor to the given $\{\boldsymbol{h}_t, \gamma_t\}_{t=1}^{T}$ of the Learner over $T$ rounds of **AIF-Learn**. We have that for any set of observed individuals $X$, with probability at least $1 - \delta$ over the observed problems F: for any $\boldsymbol{\lambda} \in \Lambda$,*

$$\frac{1}{T}\sum_{t=1}^{T} \mathcal{L}(\boldsymbol{h}_t, \gamma_t, \boldsymbol{\lambda}) - \frac{1}{T}\sum_{t=1}^{T} \mathcal{L}(\boldsymbol{h}_t, \gamma_t, \boldsymbol{\lambda}_t) \leq B\sqrt{\frac{\log(2nT/\delta)}{2m_0}} + \frac{B\log(2n+1)}{\eta T} + \eta B (1 + 2\alpha)^2$$

The last two terms appearing in the above bound come from the usual regret analysis of the exponentiated gradient descent algorithm. However, the first term originates from high probability Chernoff-type inequalities because as explained before, the algorithm is using, instead of the whole problems $F$, only the batch $F_t$ to estimate the vector of fairness violations at round $t$. Hence at round $t$, the difference of fairness violation estimates, one with respect to $F_t$ and another with respect to $F$, will appear in the regret of the Auditor which can be bounded by the Chernoff-Hoeffding's inequality. We will therefore have to assume that $m_0$ is sufficiently large to make the above regret bound small enough.

**Assumption A.1.** *For a given confidence parameter $\delta$, inputs $\alpha$ and $\nu$ of Algorithm **AIF-Learn**, we suppose throughout this section that the number of fresh problems $m_0$ used in each round of the algorithm satisfies $m_0 \geq O\left(\frac{\log(nT/\delta)}{\alpha^2 \nu^2}\right)$, or equivalently $m = m_0 \cdot T \geq O\left(\frac{T \log(nT/\delta)}{\alpha^2 \nu^2}\right)$.*

Following Lemma A.1 and Assumption A.1, we characterize the average play $\left(\widehat{\boldsymbol{p}}, \widehat{\gamma}, \widehat{\boldsymbol{\lambda}}\right)$ output by **AIF-Learn** in the following theorem. This theorem, informally speaking, guarantees that neither player would gain more than $\nu$ if they deviated from the average play strategies. This is what we call a "$\nu$-approximate equilibrium" of the game. The proof of the theorem follows from the regret analysis of the Auditor and is fully presented in section C of this document.

**Theorem A.2** (Average Play Characterization). *Let $0 < \delta < 1$. Let $\left(\widehat{\boldsymbol{p}}, \widehat{\gamma}, \widehat{\boldsymbol{\lambda}}\right)$ be the average plays output by Algorithm **AIF-Learn**. We have that under Assumption A.1, for any set of observed individuals $X$, with probability at least $1 - \delta$ over the observed labelings $F$, the average plays $\left(\widehat{\boldsymbol{p}}, \widehat{\gamma}, \widehat{\boldsymbol{\lambda}}\right)$ forms a $\nu$-approximate equilibrium of the game, i.e.,*

$$\mathcal{L}\left(\widehat{\boldsymbol{p}}, \widehat{\gamma}, \widehat{\boldsymbol{\lambda}}\right) \leq \mathcal{L}\left(\boldsymbol{p}, \gamma, \widehat{\boldsymbol{\lambda}}\right) + \nu \quad \text{for all } \boldsymbol{p} \in \Delta(\mathcal{H})^m, \gamma \in [0, 1]$$

$$\mathcal{L}\left(\widehat{\boldsymbol{p}}, \widehat{\gamma}, \widehat{\boldsymbol{\lambda}}\right) \geq \mathcal{L}\left(\widehat{\boldsymbol{p}}, \widehat{\gamma}, \boldsymbol{\lambda}\right) - \nu \quad \text{for all } \boldsymbol{\lambda} \in \Lambda$$

We are now ready to present the main theorem of this subsection which takes the guarantees provided in Theorem A.2 and turns them into "accuracy" and "fairness" guarantees of the pair $(\widehat{\boldsymbol{p}}, \widehat{\gamma})$ using the specific form of the Lagrangian. The theorem will in fact show that the set of randomized classifiers $\widehat{\boldsymbol{p}}$ achieves optimal accuracy up to $O(\nu)$ and that it also satisfies $(O(\alpha), 0)$-AIF notion of fairness, all with respect to the empirical distributions $\widehat{\mathcal{P}}$ and $\widehat{\mathcal{Q}}$.

**Theorem A.3** (In-sample Accuracy and Fairness). *Let $0 < \delta < 1$ and suppose Assumption A.1 holds. Let $(\widehat{\boldsymbol{p}}, \widehat{\gamma})$ be the output of **AIF-Learn** and let $(\boldsymbol{p}, \gamma)$ be any feasible pair of variables for the empirical fair learning problem. We have that for any set of observed individuals $X$, with probability at least $1 - \delta$ over the observed labelings $F$,*

$$err\left(\widehat{\boldsymbol{p}}; \widehat{\mathcal{P}}, \widehat{\mathcal{Q}}\right) \leq err\left(\boldsymbol{p}; \widehat{\mathcal{P}}, \widehat{\mathcal{Q}}\right) + 2\nu$$

*and that $\widehat{\boldsymbol{p}}$ satisfies $(3\alpha, 0)$-AIF with respect to the empirical distributions $(\widehat{\mathcal{P}}, \widehat{\mathcal{Q}})$. In other words, for all $i \in [n]$,*

$$\left| \mathcal{E}\left(x_i, \widehat{\boldsymbol{p}}; \widehat{\mathcal{Q}}\right) - \widehat{\gamma} \right| \leq 3\alpha$$

# B   Learning subject to FPAIF

**Definition B.1** (Individual False Positive/Negative Error Rates). *For a given individual $x \in \mathcal{X}$, a mapping $\psi \in \Delta(\mathcal{H})^{\mathcal{F}}$, and distribution $\mathcal{Q}$ over the space of problems, define the Individual false positive/negative rate:*

$$\mathcal{E}_{FP}(x, \psi; \mathcal{Q}) = \frac{1}{\underset{f \sim \mathcal{Q}}{\mathbb{P}}[f(x) = 0]} \underset{f \sim \mathcal{Q}}{\mathbb{E}}\left[\underset{h \sim \psi_f}{\mathbb{P}}[h(x) = 1, \, f(x) = 0]\right]$$

$$\mathcal{E}_{FN}(x, \psi; \mathcal{Q}) = \frac{1}{\underset{f \sim \mathcal{Q}}{\mathbb{P}}[f(x) = 1]} \underset{f \sim \mathcal{Q}}{\mathbb{E}}\left[\underset{h \sim \psi_f}{\mathbb{P}}[h(x) = 0, \, f(x) = 1]\right]$$

**Definition B.2** (FPAIF fairness notion). *In our framework, we say a mapping $\psi \in \Delta(\mathcal{H})^{\mathcal{F}}$ satisfies "$(\alpha, \beta)$-FPAIF" (reads $(\alpha, \beta)$-approximate False Positive Average Individual Fairness) with respect to the distributions $(\mathcal{P}, \mathcal{Q})$ if there exists $\gamma \geq 0$ such that*

$$\mathop{\mathbb{P}}_{x \sim \mathcal{P}} \left( \left| \mathcal{E}_{FP}\left(x, \psi; \mathcal{Q}\right) - \gamma \right| > \alpha \right) \leq \beta$$

In this section we consider the learning problem subject to the FPAIF notion of fairness. We will be less wordy in this section as the ideas and the approach that we take are mostly similar to those developed in AIF learning section. We start off by casting the fair learning problem as the constrained optimization problem 3 where a mapping $\psi$ is to be found such that all individual false positive rates incurred by $\psi$ are within $2\alpha$ of each other. As before, we denote the optimal error of the optimization problem 3 by OPT and will consider that as a benchmark to evaluate the accuracy of our algorithm's trained mapping.

---

**Fair Learning Problem subject to $(\alpha, 0)$-FPAIF**

$$\min_{\psi \in \Delta(\mathcal{H})^{\mathcal{F}}, \gamma \in [0,1]} \quad \text{err}\left(\psi; \mathcal{P}, \mathcal{Q}\right)$$
$$\text{s.t. } \forall x \in \mathcal{X}: \quad \left| \mathcal{E}_{\text{FP}}\left(x, \psi; \mathcal{Q}\right) - \gamma \right| \leq \alpha \tag{3}$$

---

**Definition B.3.** *Consider the optimization problem 3. Given distributions $\mathcal{P}$ and $\mathcal{Q}$ and fairness approximation parameter $\alpha$, we denote the optimal solutions of 3 by $\psi^{\star}\left(\alpha; \mathcal{P}, \mathcal{Q}\right) = \psi^{\star}$ and $\gamma^{\star}\left(\alpha; \mathcal{P}, \mathcal{Q}\right) = \gamma^{\star}$, and the value of the objective function at these optimal points by $OPT\left(\alpha; \mathcal{P}, \mathcal{Q}\right)$. In other words*

$$OPT\left(\alpha; \mathcal{P}, \mathcal{Q}\right) = err\left(\psi^{\star}; \mathcal{P}, \mathcal{Q}\right)$$

It is important to observe that the optimization problem 3 has a nonempty set of feasible points for any $\alpha$ and any distributions $\mathcal{P}$ and $\mathcal{Q}$. Take $\gamma = 0$ and $\psi_f = h^0$ for all $f \in \mathcal{F}$ where $h^0$ is the all-zero constant classifier, and observe that the constraint is satisfied. Since the distributions $\mathcal{P}$ and $\mathcal{Q}$ are generally not known, we instead solve the *empirical* version of 3. Consider a training data set consisting of $n$ individuals $X = \{x_i\}_{i=1}^n$ drawn independently from $\mathcal{P}$ and $m$ problems $F = \{f_j\}_{j=1}^m$ drawn independently from $\mathcal{Q}$. We formulate the empirical fair learning problem in 4 where we find an optimal fair mapping of the problems $F$ to $\Delta(\mathcal{H})$ given the individuals $X$.

---

**Empirical Fair Learning Problem**

$$\min_{\boldsymbol{p} \in \Delta(\mathcal{H})^m, \gamma \in [0,1]} \quad \text{err}\left(\boldsymbol{p}; \widehat{\mathcal{P}}, \widehat{\mathcal{Q}}\right)$$
$$\text{s.t. } \forall i \in \{1, \ldots, n\}: \quad \left| \mathcal{E}_{\text{FP}}\left(x_i, \boldsymbol{p}; \widehat{\mathcal{Q}}\right) - \gamma \right| \leq 2\alpha \tag{4}$$

---

As also discussed in the AIF section of the paper, solving the empirical problem 4 will only give us a – restricted – mapping $\boldsymbol{p} = \psi|_F$ by which we mean we learn $\psi$ only on the finite domain $F \subseteq \mathcal{F}$. It is not clear for now how we can extend the restricted mapping $\psi|_F \in \Delta(\mathcal{H})^m$ to a mapping $\psi \in \Delta(\mathcal{H})^{\mathcal{F}}$ over the whole function space; however, we will see the specific form of our algorithm (as in the AIF setting: learning a set of weights over training individuals) will allow us to come up with such an extension.

We once again use the dual perspective of constrained optimization problems followed by no regret dynamics to reduce 4 to a two-player zero-sum game between a Learner and an Auditor, and design an iterative algorithm to get an approximate equilibrium of the game. To do so, we first rewrite the constraints of 4 in $\boldsymbol{r}_{\text{FP}}\left(\boldsymbol{p}, \gamma; \widehat{\mathcal{Q}}\right) \leq 0$ form where

$$\boldsymbol{r}_{\text{FP}}\left(\boldsymbol{p}, \gamma; \widehat{\mathcal{Q}}\right) = \begin{bmatrix} \mathcal{E}_{\text{FP}}\left(x_i, \boldsymbol{p}; \widehat{\mathcal{Q}}\right) - \gamma - 2\alpha \\ \gamma - \mathcal{E}_{\text{FP}}\left(x_i, \boldsymbol{p}; \widehat{\mathcal{Q}}\right) - 2\alpha \end{bmatrix}_{i=1}^n \in \mathbb{R}^{2n} \tag{5}$$

stores the "fairness violations" of the pair $(\boldsymbol{p}, \gamma)$ in one single vector. Let the corresponding dual variables be represented by $\boldsymbol{\lambda} = \left[\boldsymbol{\lambda}_i^+, \boldsymbol{\lambda}_i^-\right]_{i=1}^n \in \Lambda$, where $\Lambda = \{\boldsymbol{\lambda} \in \mathbb{R}_+^{2n} \mid ||\boldsymbol{\lambda}||_1 \leq B\}$. Using Equation 5 and the introduced dual variables, we have that the Lagrangian of 4 is

$$\mathcal{L}\left(\boldsymbol{p}, \gamma, \boldsymbol{\lambda}\right) = \text{err}\left(\boldsymbol{p}; \widehat{\mathcal{P}}, \widehat{\mathcal{Q}}\right) + \sum_{i=1}^n \lambda_i^+ \left(\mathcal{E}_{\text{FP}}\left(x_i, \boldsymbol{p}; \widehat{\mathcal{Q}}\right) - \gamma - 2\alpha\right) + \lambda_i^- \left(\gamma - \mathcal{E}_{\text{FP}}\left(x_i, \boldsymbol{p}; \widehat{\mathcal{Q}}\right) - 2\alpha\right)$$

$$= \text{err}\left(\boldsymbol{p}; \widehat{\mathcal{P}}, \widehat{\mathcal{Q}}\right) + \boldsymbol{\lambda}^T \boldsymbol{r}_{\text{FP}}\left(\boldsymbol{p}, \gamma; \widehat{\mathcal{Q}}\right) \tag{6}$$

Therefore, we focus on solving the following minmax problem:

$$\min_{\boldsymbol{p} \in \Delta(\mathcal{H})^m, \gamma \in [0,1]} \max_{\lambda \in \Lambda} \mathcal{L}\left(\boldsymbol{p}, \gamma, \boldsymbol{\lambda}\right) = \max_{\lambda \in \Lambda} \min_{\boldsymbol{p} \in \Delta(\mathcal{H})^m, \gamma \in [0,1]} \mathcal{L}\left(\boldsymbol{p}, \gamma, \boldsymbol{\lambda}\right) \tag{7}$$

Using no regret dynamics, an approximate equilibrium of this zero-sum game (i.e. a saddle point of $\mathcal{L}$) can be found in an iterative framework. In each iteration, we let the dual player run the exponentiated gradient descent algorithm and the primal player run an approximate version of its best response using the oracle $CSC(\mathcal{H})$ that solves cost sensitive classification problems in $\mathcal{H}$.

In the following subsection, we will first describe the best response of the Learner, and show how the best response depends on the estimates $\widehat{\boldsymbol{\rho}} = [\widehat{\rho}_i]_{i=1}^n \in \mathbb{R}^n$ – where $\widehat{\rho}_i$ is the fraction of problems in $F$ that maps $x_i$ to $0$ – in addition to the weights $\boldsymbol{w}_t \in \mathbb{R}^n$ maintained by the Auditor. We will then argue that to avoid injecting correlation into the algorithm (so as to argue later on about generalization) we will have to "perturb" the best response of the Learner by using some other set of estimates $\widetilde{\boldsymbol{\rho}} = [\widetilde{\rho}_i]_{i=1}^n$ which we assume is given to the algorithm and is independent of $F$. This is why the Learner is using an *approximate* version of its best response.

**Definition B.4.** *For an individual $x \in \mathcal{X}$, let $\rho_x$ and $\widehat{\rho}_x$ represent the probability that $x$ is labelled $0$ by a randomly sampled function $f \sim \mathcal{Q}$ and $f \sim \widehat{\mathcal{Q}}$, respectively. In other words*

$$\rho_x = \mathbb{P}_{f \sim \mathcal{Q}}[f(x) = 0] \quad , \quad \widehat{\rho}_x = \mathbb{P}_{f \sim \widehat{\mathcal{Q}}}[f(x) = 0]$$

*we will use $\rho_i \equiv \rho_{x_i}$ and $\widehat{\rho}_i \equiv \widehat{\rho}_{x_i}$ to denote the corresponding probabilities for $x_i$ in the training set.*

**Remark B.1.** *Observe that using the introduced notation, for a mapping $\psi$ and $x \in \mathcal{X}$,*

$$\mathcal{E}_{FP}\left(x, \psi; \mathcal{Q}\right) = \left(\frac{1}{\rho_x}\right) \mathbb{E}_{f \sim \mathcal{Q}}\left[\mathbb{P}_{h \sim \psi_f}[h(x) = 1, \, f(x) = 0]\right]$$

$$\mathcal{E}_{FN}\left(x, \psi; \mathcal{Q}\right) = \left(\frac{1}{1 - \rho_x}\right) \mathbb{E}_{f \sim \mathcal{Q}}\left[\mathbb{P}_{h \sim \psi_f}[h(x) = 0, \, f(x) = 1]\right]$$

*and that $\mathcal{E}\left(x, \psi; \mathcal{Q}\right)$ can be written as a linear combination of $\mathcal{E}_{FP}\left(x, \psi; \mathcal{Q}\right)$ and $\mathcal{E}_{FN}\left(x, \psi; \mathcal{Q}\right)$:*

$$\mathcal{E}\left(x, \psi; \mathcal{Q}\right) = \rho_x \cdot \mathcal{E}_{FP}\left(x, \psi; \mathcal{Q}\right) + (1 - \rho_x) \cdot \mathcal{E}_{FN}\left(x, \psi; \mathcal{Q}\right)$$

## B.1  aBEST$_{\text{FP}}$: The Learner's approximate Best Response

At iteration $t$ of the algorithm, the Learner is given $\boldsymbol{\lambda}_t$ of the dual player and they want to solve the following minimization problem.

$$\min_{\boldsymbol{p} \in \Delta(\mathcal{H})^m, \gamma \in [0,1]} \mathcal{L}\left(\boldsymbol{p}, \gamma, \boldsymbol{\lambda}_t\right)$$

$$\equiv \min_{\boldsymbol{p}, \gamma} \text{err}\left(\boldsymbol{p}; \widehat{\mathcal{P}}, \widehat{\mathcal{Q}}\right) + \sum_{i=1}^n \left\{\lambda_{i,t}^+ \left(\mathcal{E}_{\text{FP}}\left(x_i, \boldsymbol{p}; \widehat{\mathcal{Q}}\right) - \gamma\right) + \lambda_{i,t}^- \left(\gamma - \mathcal{E}_{\text{FP}}\left(x_i, \boldsymbol{p}; \widehat{\mathcal{Q}}\right)\right)\right\}$$

$$\equiv \min_{\boldsymbol{p}, \gamma} \frac{1}{n} \sum_{i=1}^n \mathcal{E}\left(x_i, \boldsymbol{p}; \widehat{\mathcal{Q}}\right) + \sum_{i=1}^n \left\{\lambda_{i,t}^+ \left(\mathcal{E}_{\text{FP}}\left(x_i, \boldsymbol{p}; \widehat{\mathcal{Q}}\right) - \gamma\right) + \lambda_{i,t}^- \left(\gamma - \mathcal{E}_{\text{FP}}\left(x_i; \boldsymbol{p}; \widehat{\mathcal{Q}}\right)\right)\right\}$$

Let $\widehat{\rho}_i$ be defined as in Definition B.4. We have that by Remark B.1

$$\mathcal{E}\left(x_i, \boldsymbol{p}; \widehat{\mathcal{Q}}\right) = \widehat{\rho}_i \cdot \mathcal{E}_{\text{FP}}\left(x_i, \boldsymbol{p}; \widehat{\mathcal{Q}}\right) + (1 - \widehat{\rho}_i) \cdot \mathcal{E}_{\text{FN}}\left(x_i, \boldsymbol{p}; \widehat{\mathcal{Q}}\right)$$

Let $w_{i,t} = \lambda_{i,t}^+ - \lambda_{i,t}^-$ for all $i$ (Accordingly let $\boldsymbol{w}_t = [w_{1,t}, \ldots, w_{n,t}]^\top$). We have that the above minimization problem is equivalent to

$$\equiv \min_{\boldsymbol{p}, \gamma} -\gamma \sum_{i=1}^n w_{i,t} + \sum_{i=1}^n \left\{ \left( \frac{\widehat{\rho}_i}{n} + w_{i,t} \right) \mathcal{E}_{\mathrm{FP}} \left( x_i, \boldsymbol{p}; \widehat{\mathcal{Q}} \right) + \left( \frac{1 - \widehat{\rho}_i}{n} \right) \mathcal{E}_{\mathrm{FN}} \left( x_i, \boldsymbol{p}; \widehat{\mathcal{Q}} \right) \right\}$$

Now we can use the fact that

$$\mathcal{E}_{\mathrm{FP}} \left( x_i, \boldsymbol{p}; \widehat{\mathcal{Q}} \right) = \frac{1}{m\widehat{\rho}_i} \sum_{j=1}^m \mathbb{P}_{h_j \sim p_j} [h_j(x_i) = 1, f_j(x_i) = 0]$$

$$\mathcal{E}_{\mathrm{FN}} \left( x_i, \boldsymbol{p}; \widehat{\mathcal{Q}} \right) = \frac{1}{m(1 - \widehat{\rho}_i)} \sum_{j=1}^m \mathbb{P}_{h_j \sim p_j} [h_j(x_i) = 0, f_j(x_i) = 1]$$

to conclude that the best response of the Learner is equivalent to minimizing the following function over the space of $(\boldsymbol{p}, \gamma) \in \Delta(\mathcal{H})^m \times [0, 1]$.

$$-\gamma \sum_{i=1}^n w_{i,t}$$

$$+ \frac{1}{m} \sum_{j=1}^m \left\{ \sum_{i=1}^n \left( \frac{1}{n} + \frac{w_{i,t}}{\widehat{\rho}_i} \right) \mathbb{P}_{h_j \sim p_j} [h_j(x_i) = 1, f_j(x_i) = 0] + \left( \frac{1}{n} \right) \mathbb{P}_{h_j \sim p_j} [h_j(x_i) = 0, f_j(x_i) = 1] \right\}$$

Therefore, as in the AIF setting, the minimization problem of the Learner gets nicely decoupled into $(m + 1)$ disjoint minimization problems. First, the optimal value for $\gamma$ is chosen according to

$$\gamma_t = \mathbb{1} \left[ \sum_{i=1}^n w_{i,t} > 0 \right] \tag{8}$$

and that for learning problem $j$, the following cost sensitive classification problem must be solved.

$$h_j = \arg\min_{h \in \mathcal{H}} \sum_{i=1}^n c_{i,j}^1 h(x_i) + c_{i,j}^0 (1 - h(x_i)) \tag{9}$$

where the costs are

$$c_{i,j}^1 = \left( \frac{1}{n} + \frac{w_{i,t}}{\widehat{\rho}_i} \right) (1 - f_j(x_i)) \quad , \quad c_{i,j}^0 = \left( \frac{1}{n} \right) f_j(x_i)$$

One major distinction between the Learner's best response in the FPAIF setting versus that of the AIF setting is that the empirical quantities $\{\widehat{\rho}_i\}_{i=1}^n$ (which is estimated using the data set $F$) appear in the costs of the CSC problems for the FPAIF setting. As a major consequence, the generalization (with respect to $\mathcal{Q}$) arguments we had in the AIF setting won't work in this section because now the labels $\{f_j(x_i)\}_{i,j}$ and the estimates $\{\widehat{\rho}_i\}_{i=1}^n$ are correlated. We therefore assume in this section that each individual $x_i \in X$ comes with an estimate $\widetilde{\rho}_i$ of the rate $\rho_i$ that is independent of the data set $F$. More precisely, we assume our algorithm has access to estimates $\{\widetilde{\rho}_i\}_{i=1}^n$ such that for all $i \in [n]$,

$$|\widetilde{\rho}_i - \rho_i| \leq \sqrt{\frac{\log(n)}{2m_0}}$$

where $m_0$ (will be specified exactly in our proposed algorithm) as in the AIF setting will be essentially the number of fresh problems that the Auditor is using in each round of the Algorithm. In fact, similar to what we did for the AIF setting, we will randomly partition $F$ into $T$ batch of size $m_0$: $F = \{F_t\}_{t=1}^T$ and will let the Auditor use only $F_t$ at round $t \in [T]$ of the algorithm to update the vector of fairness violations $\boldsymbol{r}_{\mathrm{FP}}$. Notice assuming access to the estimates $\{\widetilde{\rho}_i\}_{i=1}^n$ is not farfetched because we can assume there was one more batch of $m_0$ problems, say $F_0$, and that the quantities $\{\widetilde{\rho}_i\}_{i=1}^n$ were estimated using the batch $F_0$ which is independent of $F = \{F_t\}_{t=1}^T$. The upper bound we required for the difference $|\widetilde{\rho}_i - \rho_i|$ will just simply follow from a Chernoff-Hoeffding's bound.

**Assumption B.1.** *For $m_0$ specified later on, we assume in this section that our algorithm has access to quantities $\{\widetilde{\rho}_i\}_{i=1}^n$, where we have that for all $i \in [n]$:*

$$|\widetilde{\rho}_i - \rho_i| \leq \sqrt{\frac{\log(n)}{2m_0}}$$

Under Assumption B.1, we now modify the best response of the Learner and let it use the estimates $\{\widetilde{\rho}_i\}_{i=1}^n$ instead of $\{\widehat{\rho}_i\}_{i=1}^n$. This will consequently make the learner to accumulate regret over the course of the algorithm and that this is why we will call it the *approximate* best response of the Learner. We have the approximate best response of the Learner (called **aBEST**$_{\text{FP}}$) written in Subroutine 2.

---

**Subroutine 2: aBEST$_{\text{FP}}$** – approximate best response of the Learner in the FPAIF setting

---

**Input:** dual weights $\boldsymbol{w} \in \mathbb{R}^n$, estimates $\widetilde{\boldsymbol{\rho}} \in \mathbb{R}^n$, training examples $S = \left\{ x_i, (f_j(x_i))_{j=1}^m \right\}_{i=1}^n$

$\gamma \leftarrow \mathbb{1}\left[\sum_{i=1}^n w_i > 0\right]$
**for** $j = 1, \ldots, m$ **do**
    $c_i^1 \leftarrow (w_i/\widetilde{\rho}_i + 1/n)(1 - f_j(x_i))$ for $i \in [n]$
    $c_i^0 \leftarrow (1/n)f_j(x_i)$ for $i \in [n]$
    $D \leftarrow \{x_i, c_i^1, c_i^0\}_{i=1}^n$
    $h_j \leftarrow CSC(\mathcal{H}; D)$
**end**
$\boldsymbol{h} \leftarrow (h_1, h_2, \ldots, h_m)$

**Output:** $\left(\boldsymbol{h}, \gamma\right)$

---

## B.2 Algorithm Implementation and In-sample Guarantees

We implement the introduced game theoretic framework in Algorithm 3 and call it **FPAIF-Learn**. The overall style of the algorithm is similar to **AIF-Learn** except that **FPAIF-Learn** takes a set of estimates $\widetilde{\boldsymbol{\rho}} \in \mathbb{R}^n$ as input and that $\widetilde{\boldsymbol{\rho}}$ is used in the approximate best response of the Learner. Once again we split $F$ into $T$ batches of size $m_0$ uniformly at random and let the Auditor use only a fresh batch of $m_0$ problems in each round of the algorithm to update the dual variables $\boldsymbol{\lambda}$, and accordingly the weights $\boldsymbol{w}$. The algorithm will terminate after $T = O\left(\log(n)/\left(\nu^2\alpha^2\right)\right)$ iterations and output the average plays of the Learner and the Auditor, along with a mapping

$$\widehat{\psi} = \widehat{\psi}\left(X, \widetilde{\boldsymbol{\rho}}, \widehat{W}\right) \in \Delta(\mathcal{H})^{\mathcal{F}}$$

which is the object we wanted to learn. In fact, $\widehat{\psi}$ extends the learned restricted mapping $\widehat{\boldsymbol{p}} = \widehat{\psi}|_F$ of the algorithm from the finite domain $F$ to the whole space $\mathcal{F}$. We have the pseudocode for the mapping $\widehat{\psi}$ written in Mapping 4.

The analysis of Algorithm 3 will follow the same style and uses the same ideas as in the AIF learning section. We will start off by establishing the regret bounds for the Learner and the Auditor in Lemma B.1 and Lemma B.2, respectively, and will show just as before how these regret bounds can be eventually turned into in-sample accuracy and fairness guarantees. One major difference is that when working with false positive rates, the quantities $\rho_x, \widehat{\rho}_x, \widetilde{\rho}_x$ introduced before will show up in the algorithm's analysis. Define for the training individuals $X = \{x_i\}_{i=1}^n$

$$\rho_{\min} = \min_{i \in [n]} \rho_i \quad , \quad \widehat{\rho}_{\min} = \min_{i \in [n]} \widehat{\rho}_i \quad , \quad \widetilde{\rho}_{\min} = \min_{i \in [n]} \widetilde{\rho}_i$$

We will in fact see $\widetilde{\rho}_{\min}$ and $\widehat{\rho}_{\min}$ will appear in the regret bounds of the Learner and the Auditor.

**Lemma B.1** (Regret of the Learner). *Let $\{\boldsymbol{h}_t, \gamma_t\}_{t=1}^T$ be the sequence of approximate best response plays by the Learner to the given $\{\boldsymbol{\lambda}_t\}_{t=1}^T$ of the Auditor over $T$ rounds of Algorithm 3. We have that*

**Algorithm 3: FPAIF-Learn** – learning subject to FPAIF

---

**Input:** fairness parameter $\alpha$, approximation parameter $\nu$, estimates $\widetilde{\boldsymbol{\rho}} = [\widetilde{\rho}_i]_{i=1}^n$
    training data set $X = \{x_i\}_{i=1}^n$ and $F = \{f_j\}_{j=1}^m$

Set
$B \leftarrow \frac{1+2\nu}{\alpha}, \ \ T \leftarrow \frac{16B^2(1+2\alpha)^2 \log(2n+1)}{\nu^2}, \ \ \eta \leftarrow \frac{\nu}{4(1+2\alpha)^2 B}, \ \ m_0 \leftarrow \frac{m}{T}, \ \ S \leftarrow \left\{ x_i, (f_j(x_i))_j \right\}_i$

Partition $F$ uniformly at random: $F = \{F_t\}_{t=1}^T$ where $|F_t| = m_0$.
$\boldsymbol{\theta}_1 \leftarrow \mathbf{0} \in \mathbb{R}^{2n}$
**for** $t = 1, \dots, T$ **do**
$\quad$ $\lambda_{i,t}^\bullet \leftarrow B \frac{\exp(\theta_{i,t}^\bullet)}{1 + \sum_{i',\bullet'} \exp(\theta_{i',t}^{\bullet'})}$ for $1 \le i \le n$ and $\bullet \in \{+, -\}$
$\quad$ $\boldsymbol{w}_t \leftarrow [\lambda_{i,t}^+ - \lambda_{i,t}^-]_{i=1}^n \in \mathbb{R}^n$
$\quad$ $(\boldsymbol{h}_t, \gamma_t) \leftarrow \textbf{aBEST}_{\text{FP}}(\boldsymbol{w}_t; \widetilde{\boldsymbol{\rho}}, S)$
$\quad$ $\boldsymbol{\theta}_{t+1} \leftarrow \boldsymbol{\theta}_t + \eta \cdot \boldsymbol{r}_{\text{FP}}\left( \boldsymbol{h}_t |_{F_t}, \gamma_t; \widehat{\mathcal{Q}}_t \right)$
**end**
$\widehat{\gamma} \leftarrow \frac{1}{T} \sum_{t=1}^T \gamma_t, \quad \widehat{\boldsymbol{p}} \leftarrow \frac{1}{T} \sum_{t=1}^T \boldsymbol{h}_t, \quad \widehat{\boldsymbol{\lambda}} \leftarrow \frac{1}{T} \sum_{t=1}^T \boldsymbol{\lambda}_t, \quad \widehat{W} \leftarrow \{\boldsymbol{w}_t\}_{t=1}^T$

**Output:** average plays $\left( \widehat{\boldsymbol{p}}, \widehat{\gamma}, \widehat{\boldsymbol{\lambda}} \right)$, mapping $\widehat{\psi} = \widehat{\psi}\left( X, \widetilde{\boldsymbol{\rho}}, \widehat{W} \right)$ (see Mapping 4)

---

**Mapping 4:** $\widehat{\psi}\left( X, \widetilde{\boldsymbol{\rho}}, \widehat{W} \right)$ – pseudocode for the mapping $\widehat{\psi}$ output by Algorithm 3

---

**Input:** $f \in \mathcal{F}$

**for** $t = 1, \dots, T$ **do**
$\quad$ $c_i^1 \leftarrow (w_{i,t}/\widetilde{\rho}_i + 1/n)(1 - f(x_i))$ for $i \in [n]$
$\quad$ $c_i^0 \leftarrow (1/n) f(x_i)$ for $i \in [n]$
$\quad$ $D \leftarrow \{x_i, c_i^1, c_i^0\}_{i=1}^n$
$\quad$ $h_{f, \boldsymbol{w}_t} \leftarrow CSC\left( \mathcal{H}; D \right)$
**end**
$\widehat{\psi}_f \leftarrow \frac{1}{T} \sum_{t=1}^T h_{f, \boldsymbol{w}_t}$

**Output:** $\widehat{\psi}_f \in \Delta(\mathcal{H})$

---

*for any set of observed individuals $X$, with probability at least $1 - \delta/2$ over the observed problems $F$, the (average) regret of the Learner is bounded as follows.*

$$\frac{1}{T} \sum_{t=1}^T \mathcal{L}\left( \boldsymbol{h}_t, \gamma_t, \boldsymbol{\lambda}_t \right) - \frac{1}{T} \min_{\boldsymbol{p} \in \Delta(\mathcal{H})^m, \gamma \in [0,1]} \sum_{t=1}^T \mathcal{L}\left( \boldsymbol{p}, \gamma, \boldsymbol{\lambda}_t \right) \le \frac{4B}{\widetilde{\rho}_{min}} \sqrt{\frac{\log\left( 8nT/\delta \right)}{2m_0}}$$

**Lemma B.2** (Regret of the Auditor). *Let $0 < \delta < 1$. Let $\{\boldsymbol{\lambda}_t\}_{t=1}^T$ be the sequence of exponentiated gradient descent plays (with learning rate $\eta$) by the Auditor to the given $\{\boldsymbol{h}_t, \gamma_t\}_{t=1}^T$ of the Learner over $T$ rounds of Algorithm 3. We have that for any set of observed individuals $X$, with probability at least $1 - \delta/2$ over the observed problems $F$, the (average) regret of the Auditor is bounded as follows: For any $\boldsymbol{\lambda} \in \Lambda$,*

$$\frac{1}{T} \sum_{t=1}^T \mathcal{L}(\boldsymbol{h}_t, \gamma_t, \boldsymbol{\lambda}) - \frac{1}{T} \sum_{t=1}^T \mathcal{L}(\boldsymbol{h}_t, \gamma_t, \boldsymbol{\lambda}_t) \le \frac{2B}{\widehat{\rho}_{min}} \sqrt{\frac{\log\left( 8nT/\delta \right)}{2m_0}} + \frac{B \log\left( 2n+1 \right)}{\eta T} + \eta B \left( 1 + 2\alpha \right)^2$$

Observe that in order to control the regret of the Learner and the Auditor at level $O(\nu)$ we need to assume that $m_0$ is large enough such that the regret bound of the Learner and the first term appearing in the regret bound of the Auditor are sufficiently small.

**Assumption B.2.** *For a given confidence parameter $\delta$, inputs $\alpha$ and $\nu$ of Algorithm 3, we assume that the number of fresh problems $m_0$ used in each round of Algorithm 3 satisfies $m_0 \geq O\left(\frac{\log(nT/\delta)}{\alpha^2 \nu^2 \rho_{min}^2}\right)$, or equivalently $m = m_0 \cdot T \geq O\left(\frac{T \log(nT/\delta)}{\alpha^2 \nu^2 \rho_{min}^2}\right)$.*

Note that Assumption B.2 immediately implies via a Chernoff bound that $\widehat{\rho}_{\min} \geq \rho_{\min}/2$ and that it also implies via Assumption B.1 that $\widetilde{\rho}_{\min} \geq \rho_{\min}/2$. In the following theorem we characterize the average play of the Learner and the Auditor. The proof of this theorem follows from the regret bounds developed in Lemma B.1 (Regret of the Learner) and Lemma B.2 (Regret of the Auditor) and uses exactly the same techniques as in the proof of Theorem A.2.

**Theorem B.3** (Average Play Characterization). *Let $0 < \delta < 1$. Let $\left(\widehat{\boldsymbol{p}}, \widehat{\gamma}, \widehat{\boldsymbol{\lambda}}\right)$ be the average plays output by Algorithm 3. We have that under Assumption B.2, for any set of observed individuals $X$, with probability at least $1 - \delta$ over the observed labelings $F$, the average plays $\left(\widehat{\boldsymbol{p}}, \widehat{\gamma}, \widehat{\boldsymbol{\lambda}}\right)$ forms a $\nu$-approximate equilibrium of the game, i.e.,*

$$\mathcal{L}\left(\widehat{\boldsymbol{p}}, \widehat{\gamma}, \widehat{\boldsymbol{\lambda}}\right) \leq \mathcal{L}\left(\boldsymbol{p}, \gamma, \widehat{\boldsymbol{\lambda}}\right) + \nu \quad \text{for all } \boldsymbol{p} \in \Delta(\mathcal{H})^m, \gamma \in [0,1]$$

$$\mathcal{L}\left(\widehat{\boldsymbol{p}}, \widehat{\gamma}, \widehat{\boldsymbol{\lambda}}\right) \geq \mathcal{L}\left(\widehat{\boldsymbol{p}}, \widehat{\gamma}, \boldsymbol{\lambda}\right) - \nu \quad \text{for all } \boldsymbol{\lambda} \in \Lambda$$

We conclude this subsection with our main Theorem B.4 that provides in-sample accuracy and fairness guarantees for the learned set of classifiers $\widehat{\boldsymbol{p}} \in \Delta(\mathcal{H})^m$ of Algorithm 3. This theorem follows immediately from Theorem B.3 and the proof is pretty much similar in style to the proof of Theorem A.3.

**Theorem B.4** (In-sample Accuracy and Fairness). *Let $0 < \delta < 1$ and suppose Assumption B.2 holds. Let $(\widehat{\boldsymbol{p}}, \widehat{\gamma})$ be the output of Algorithm 3 and let $(\boldsymbol{p}, \gamma)$ be any feasible pair of variables for the empirical fair learning problem 4. We have that for any set of observed individuals $X$, with probability at least $1 - \delta$ over the observed labelings $F$,*

$$err\left(\widehat{\boldsymbol{p}}; \widehat{\mathcal{P}}, \widehat{\mathcal{Q}}\right) \leq err\left(\boldsymbol{p}; \widehat{\mathcal{P}}, \widehat{\mathcal{Q}}\right) + 2\nu$$

*and that $\widehat{\boldsymbol{p}}$ satisfies $(3\alpha, 0)$-FPAIF with respect to the empirical distributions $(\widehat{\mathcal{P}}, \widehat{\mathcal{Q}})$. In other words, for all $i \in [n]$,*

$$\left|\mathcal{E}_{FP}\left(x_i, \widehat{\boldsymbol{p}}; \widehat{\mathcal{Q}}\right) - \widehat{\gamma}\right| \leq 3\alpha$$

## B.3 Generalization Theorems

We now consider the mapping $\widehat{\psi}$ learned by Algorithm 3 and study the generalization bounds both for accuracy and fairness. As in the AIF learning setting, we state our generalization theorems in three steps. We first consider in Theorem B.5 the empirical distribution of the problems $\widehat{\mathcal{Q}}$ and see how we can lift the guarantees from $\widehat{\mathcal{P}}$ to the true underlying distribution of individuals $\mathcal{P}$. We will then take the same approach, but this time consider generalization only over the problem generating distribution $\mathcal{Q}$ in Theorem B.6. We will eventually in Theorem B.7 provide accuracy and fairness guarantees for the learned mapping $\widehat{\psi}$ with respect to the distributions $(\mathcal{P}, \mathcal{Q})$. We will use OPT (defined formally in Definition B.3) as a benchmark to evaluate the accuracy of the mapping $\widehat{\psi}$. The proofs for the theorems of this section are similar to those of the AIF section.

**Theorem B.5** (Generalization over $\mathcal{P}$). *Let $0 < \delta < 1$. Let $\left(\widehat{\psi}, \widehat{\gamma}\right)$ be the outputs of Algorithm 3, and suppose*

$$n \geq \widetilde{O}\left(\frac{m\, d_{\mathcal{H}} + \log\left(1/\nu^2\delta\right)}{\alpha^2\beta^2}\right)$$

*where $d_{\mathcal{H}}$ is the VC dimension of $\mathcal{H}$. We have that with probability at least $1 - 6\delta$ over the observed data set $(X, F)$, the mapping $\widehat{\psi}$ is $(5\alpha, \beta)$-FPAIF with respect to the distributions $\left(\mathcal{P}, \widehat{\mathcal{Q}}\right)$, i.e.,*

$$\mathbb{P}_{x \sim \mathcal{P}}\left(\left|\mathcal{E}_{FP}\left(x, \widehat{\psi}; \widehat{\mathcal{Q}}\right) - \widehat{\gamma}\right| > 5\alpha\right) \leq \beta$$

*and that,*

$$err\left(\widehat{\psi}; \mathcal{P}, \widehat{\mathcal{Q}}\right) \leq OPT\left(\alpha; \mathcal{P}, \widehat{\mathcal{Q}}\right) + O\left(\nu\right) + O\left(\alpha\beta\right)$$

**Theorem B.6** (Generalization over $\mathcal{Q}$)**.** *Let $0 < \delta < 1$. Let $\left(\widehat{\psi}, \widehat{\gamma}\right)$ be the outputs of Algorithm 3 and suppose*

$$m \geq \widetilde{O}\left(\frac{\log\left(n\right)\log\left(n/\delta\right)}{\rho_{min}^2 \nu^4 \alpha^4}\right)$$

*We have that for any set of observed individuals $X$, with probability at least $1 - 6\delta$ over the observed problems $F$, the learned mapping $\widehat{\psi}$ is $(4\alpha, 0)$-FPAIF with respect to the distributions $\left(\widehat{\mathcal{P}}, \mathcal{Q}\right)$, i.e.,*

$$\mathbb{P}_{x \sim \widehat{\mathcal{P}}}\left(\left|\mathcal{E}_{FP}\left(x, \widehat{\psi}; \mathcal{Q}\right) - \widehat{\gamma}\right| > 4\alpha\right) = 0$$

*and that,*

$$err\left(\widehat{\psi}; \widehat{\mathcal{P}}, \mathcal{Q}\right) \leq OPT\left(\alpha; \widehat{\mathcal{P}}, \mathcal{Q}\right) + O\left(\nu\right)$$

**Theorem B.7** (Simultaneous Generalization over $\mathcal{P}$ and $\mathcal{Q}$)**.** *Let $0 < \delta < 1$. Let $\left(\widehat{\psi}, \widehat{\gamma}\right)$ be the outputs of Algorithm 3 and suppose*

$$n \geq \widetilde{O}\left(\frac{m\, d_{\mathcal{H}} + \log\left(1/\nu^2\delta\right)}{\alpha^2 \beta^2}\right) \quad , \quad m \geq \widetilde{O}\left(\frac{\log\left(n\right)\log\left(n/\delta\right)}{\rho_{inf}^2 \nu^4 \alpha^4}\right)$$

*where $d_{\mathcal{H}}$ is the VC dimension of $\mathcal{H}$ and $\rho_{inf} = \inf_{x \in \mathcal{X}} \rho_x$. We have that with probability at least $1 - 10\delta$ over the observed individuals $X$ and the problems $F$, the learned mapping $\widehat{\psi}$ is $(6\alpha, 2\beta)$-FPAIF with respect to the distributions $(\mathcal{P}, \mathcal{Q})$, i.e.,*

$$\mathbb{P}_{x \sim \mathcal{P}}\left(\left|\mathcal{E}_{FP}\left(x, \widehat{\psi}; \mathcal{Q}\right) - \widehat{\gamma}\right| > 6\alpha\right) \leq 2\beta$$

*and that,*

$$err\left(\widehat{\psi}; \mathcal{P}, \mathcal{Q}\right) \leq OPT\left(\alpha; \mathcal{P}, \mathcal{Q}\right) + O\left(\nu\right) + O\left(\alpha\beta\right)$$

## C   Proofs of the paper

### C.1   Preliminary Tools

**Theorem C.1** (Additive Chernoff-Hoeffding Bound)**.** *Let $X = \{X_i\}_{i=1}^n$ be a sequence of i.i.d. random variables with $a \leq X_i \leq b$ and $\mathbb{E}\left[X_i\right] = \mu$ for all $i$. We have that for all $s > 0$,*

$$\mathbb{P}_X\left[\left|\frac{\sum_i X_i}{n} - \mu\right| \geq s\right] \leq 2\exp\left(\frac{-2ns^2}{(b-a)^2}\right)$$

**Lemma C.2** (Sauer's Lemma (see e.g. Kearns and Vazirani (1994)))**.** *Let $\mathcal{H}$ be a class of binary functions defined on $\mathcal{X}$ where the VC dimension of $\mathcal{H}$, $d_{\mathcal{H}}$, is finite. Let $X = \{x_i\}_{i=1}^n$ be a data set of size $n$ drwan from $\mathcal{X}$ and let $\mathcal{H}(X) = \{(h(x_1), \ldots, h(x_n)) : h \in \mathcal{H}\}$ be the set of induced labelings of $\mathcal{H}$ on $X$. We have that $|\mathcal{H}(X)| \leq O\left(n^{d_{\mathcal{H}}}\right)$.*

**Theorem C.3** (Exponentiated Gradient Descent Regret (see corollary 2.14 of Shalev-Shwartz (2012)))**.** *Let $\Lambda' = \left\{\boldsymbol{\lambda}' \in \mathbb{R}_+^d : ||\boldsymbol{\lambda}'||_1 = B\right\}$. Suppose the exponentiated gradient descent algorithm with learning rate $\eta$ is run on the sequence of linear functions $\left\{f_t(\boldsymbol{\lambda}') = (\boldsymbol{\lambda}')^\top \boldsymbol{r}_t\right\}_{t=1}^T$ where $\boldsymbol{\lambda}' \in \Lambda'$ and $||\boldsymbol{r}_t||_\infty \leq L$ for all $t$. Let $\boldsymbol{\lambda}'_t$ denote the exponentiated gradient descent play at round $t$. We have that the regret of the algorithm over $T$ rounds is:*

$$Regret_T(\Lambda') = \max_{\boldsymbol{\lambda}' \in \Lambda'} \sum_{t=1}^T (\boldsymbol{\lambda}')^\top \boldsymbol{r}_t - \sum_{t=1}^T (\boldsymbol{\lambda}'_t)^\top \boldsymbol{r}_t \leq \frac{B\log\left(d\right)}{\eta} + \eta B L^2 T$$

*and that for $\eta = O\left(1/\sqrt{T}\right)$, we have that $Regret_T(\Lambda') = O\left(\sqrt{T}\right)$.*

## C.2 Missing proofs: Learning subject to AIF

### C.2.1 Algorithm analysis and in-sample guarantees

*Proof of Lemma A.1.* Fix the set of observed individuals $X \in \mathcal{X}^n$. Recall that $\Lambda = \{\boldsymbol{\lambda} \in \mathbb{R}_+^{2n} : ||\boldsymbol{\lambda}||_1 \le B\}$ is the set of strategies for the Auditor. Now let $\Lambda' = \{\boldsymbol{\lambda}' \in \mathbb{R}_+^{2n+1} : ||\boldsymbol{\lambda}'||_1 = B\}$. Any $\boldsymbol{\lambda} \in \Lambda$ is associated with a $\boldsymbol{\lambda}' \in \Lambda'$ which is equal to $\boldsymbol{\lambda}$ on the first $2n$ coordinates and has the remaining mass on the last one. Let $\widehat{\boldsymbol{r}}_t = \boldsymbol{r}(h_t|_{F_t}, \gamma_t; \widehat{\mathcal{Q}}_t)$ be the vector of fairness violations – estimated over only the $m_0$ problems $F_t$ of round $t$ – that the Auditor is using in Algorithm **AIF-Learn**, and let $\widehat{\boldsymbol{r}}_t' \in \mathbb{R}^{2n+1}$ be equal to $\widehat{\boldsymbol{r}}_t$ on the first $2n$ coordinates and zero in the last one. We have that for any $\boldsymbol{\lambda} \in \Lambda$ and its associated $\boldsymbol{\lambda}' \in \Lambda'$, and in particular for $\boldsymbol{\lambda}_t$ and $\boldsymbol{\lambda}_t'$ of Algorithm **AIF-Learn**, and all $t \in [T]$,

$$\boldsymbol{\lambda}^\top \widehat{\boldsymbol{r}}_t = (\boldsymbol{\lambda}')^\top \widehat{\boldsymbol{r}}_t' \quad , \quad \boldsymbol{\lambda}_t^\top \widehat{\boldsymbol{r}}_t = (\boldsymbol{\lambda}_t')^\top \widehat{\boldsymbol{r}}_t' \tag{10}$$

Now by Theorem C.3, and using the observation that $||\widehat{\boldsymbol{r}}_t'||_\infty = ||\widehat{\boldsymbol{r}}_t||_\infty \le 1 + 2\alpha$, we have that for any $\boldsymbol{\lambda}' \in \Lambda'$,

$$\sum_{t=1}^T (\boldsymbol{\lambda}')^\top \widehat{\boldsymbol{r}}_t' \le \sum_{t=1}^T (\boldsymbol{\lambda}_t')^\top \widehat{\boldsymbol{r}}_t' + \frac{B \log (2n+1)}{\eta} + \eta (1+2\alpha)^2 BT$$

Consequently by Equation 10, we have that for any $\boldsymbol{\lambda} \in \Lambda$,

$$\sum_{t=1}^T \boldsymbol{\lambda}^\top \widehat{\boldsymbol{r}}_t \le \sum_{t=1}^T \boldsymbol{\lambda}_t^\top \widehat{\boldsymbol{r}}_t + \frac{B \log (2n+1)}{\eta} + \eta (1+2\alpha)^2 BT$$

Now let $\boldsymbol{r}_t = \boldsymbol{r}\left(h_t, \gamma_t; \widehat{\mathcal{Q}}\right)$ be the vector of fairness violations estimated over all problems $F$ and notice that the regret bound must be with respect to $\boldsymbol{r}_t$ and not $\widehat{\boldsymbol{r}}_t$. With that goal in mind, we have that for any $\boldsymbol{\lambda} \in \Lambda$,

$$\sum_{t=1}^T \boldsymbol{\lambda}^\top \boldsymbol{r}_t \le \sum_{t=1}^T \boldsymbol{\lambda}_t^\top \boldsymbol{r}_t + \sum_{t=1}^T (\boldsymbol{\lambda}_t - \boldsymbol{\lambda})^\top (\widehat{\boldsymbol{r}}_t - \boldsymbol{r}_t) + \frac{B \log (2n+1)}{\eta} + \eta (1+2\alpha)^2 BT$$

We will use Chernoff-Hoeffding's inequality to bound the difference $(\widehat{\boldsymbol{r}}_t - \boldsymbol{r}_t)$ in $\ell_\infty$ norm. Let $\widehat{\psi}_t = \widehat{\psi}(X, \boldsymbol{w}_t)$ where $\boldsymbol{w}_t$ is the vector of weights used in round $t$ of the algorithm and observe that we can rewrite $\widehat{\boldsymbol{r}}_t$ and $\boldsymbol{r}_t$ in terms of $\widehat{\psi}_t$:

$$\widehat{\boldsymbol{r}}_t = \begin{bmatrix} \mathcal{E}\left(x_i, \widehat{\psi}_t; \widehat{\mathcal{Q}}_t\right) - \gamma_t - 2\alpha \\ \gamma_t - \mathcal{E}\left(x_i, \widehat{\psi}_t; \widehat{\mathcal{Q}}_t\right) - 2\alpha \end{bmatrix}_{i=1}^n \quad , \quad \boldsymbol{r}_t = \begin{bmatrix} \mathcal{E}\left(x_i, \widehat{\psi}_t; \widehat{\mathcal{Q}}\right) - \gamma_t - 2\alpha \\ \gamma_t - \mathcal{E}\left(x_i, \widehat{\psi}_t; \widehat{\mathcal{Q}}\right) - 2\alpha \end{bmatrix}_{i=1}^n$$

Hence, bounding the difference $(\widehat{\boldsymbol{r}}_t - \boldsymbol{r}_t)$ in $\ell_\infty$ norm involves bounding the terms

$$\left| \mathcal{E}\left(x_i, \widehat{\psi}_t; \widehat{\mathcal{Q}}_t\right) - \mathcal{E}\left(x_i, \widehat{\psi}_t; \widehat{\mathcal{Q}}\right) \right|$$

for all $i$. Notice we can now view the batch of problems $F_t$ as independent draws from the distribution $\widehat{\mathcal{Q}}$. Therefore, it follows from the Chernoff-Hoeffding's Theorem C.1 that with probability at least $1 - \delta$ over the set of problems $F$, for any $\boldsymbol{\lambda} \in \Lambda$,

$$\sum_{t=1}^T (\boldsymbol{\lambda}_t - \boldsymbol{\lambda})^\top (\widehat{\boldsymbol{r}}_t - \boldsymbol{r}_t) \le \sum_{t=1}^T ||\boldsymbol{\lambda}_t - \boldsymbol{\lambda}||_1 \cdot ||\widehat{\boldsymbol{r}}_t - \boldsymbol{r}_t||_\infty \le B \sum_{t=1}^T ||\widehat{\boldsymbol{r}}_t - \boldsymbol{r}_t||_\infty \le BT \sqrt{\frac{\log (2nT/\delta)}{2m_0}}$$

which implies with probability at least $1 - \delta$ over the problems $F$, for any $\boldsymbol{\lambda} \in \Lambda$,

$$\frac{1}{T} \sum_{t=1}^T \mathcal{L}\left(h_t, \gamma_t, \boldsymbol{\lambda}\right) - \frac{1}{T} \sum_{t=1}^T \mathcal{L}\left(h_t, \gamma_t, \boldsymbol{\lambda}_t\right) \le B \sqrt{\frac{\log (2nT/\delta)}{2m_0}} + \frac{B \log (2n+1)}{\eta T} + \eta (1+2\alpha)^2 B$$

completing the proof. $\qquad\square$

*Proof of Theorem A.2.* Let

$$R_{\boldsymbol{\lambda}} := B\sqrt{\frac{\log\left(2nT/\delta\right)}{2m_0}} + \frac{B\log\left(2n+1\right)}{\eta T} + \eta\left(1+2\alpha\right)^2 B$$

be the average regret of the Auditor. We have that for any $\boldsymbol{p} \in \Delta(\mathcal{H})^m$ and $\gamma \in [0,1]$,

$$
\begin{aligned}
\mathcal{L}\left(\boldsymbol{p},\gamma,\widehat{\boldsymbol{\lambda}}\right) &= \frac{1}{T}\sum_{t=1}^{T}\mathcal{L}\left(\boldsymbol{p},\gamma,\boldsymbol{\lambda}_t\right) \quad \text{(by linearity of } \mathcal{L}) \\
&\geq \frac{1}{T}\sum_{t=1}^{T}\mathcal{L}\left(\boldsymbol{h}_t,\gamma_t,\boldsymbol{\lambda}_t\right) \quad ((\boldsymbol{h}_t,\gamma_t) \text{ is Learner's Best Response)} \\
&\geq \frac{1}{T}\sum_{t=1}^{T}\mathcal{L}\left(\boldsymbol{h}_t,\gamma_t,\widehat{\boldsymbol{\lambda}}\right) - R_{\boldsymbol{\lambda}} \quad \text{(w.p. } 1-\delta \text{ over } F \text{ by Lemma A.1)} \\
&= \mathcal{L}\left(\widehat{\boldsymbol{p}},\widehat{\gamma},\widehat{\boldsymbol{\lambda}}\right) - R_{\boldsymbol{\lambda}}
\end{aligned}
$$

And that for any $\boldsymbol{\lambda} \in \Lambda$ we have:

$$
\begin{aligned}
\mathcal{L}\left(\widehat{\boldsymbol{p}},\widehat{\gamma},\boldsymbol{\lambda}\right) &= \frac{1}{T}\sum_{t=1}^{T}\mathcal{L}\left(\boldsymbol{h}_t,\gamma_t,\boldsymbol{\lambda}\right) \quad \text{(by linearity of } \mathcal{L}) \\
&\leq \frac{1}{T}\sum_{t=1}^{T}\mathcal{L}\left(\boldsymbol{h}_t,\gamma_t,\boldsymbol{\lambda}_t\right) + R_{\boldsymbol{\lambda}} \quad \text{(w.p. } 1-\delta \text{ over } F \text{ by Lemma A.1)} \\
&\leq \frac{1}{T}\sum_{t=1}^{T}\mathcal{L}\left(\widehat{\boldsymbol{p}},\widehat{\gamma},\boldsymbol{\lambda}_t\right) + R_{\boldsymbol{\lambda}} \quad ((\boldsymbol{h}_t,\gamma_t) \text{ is Learner's Best Response)} \\
&= \mathcal{L}\left(\widehat{\boldsymbol{p}},\widehat{\gamma},\widehat{\boldsymbol{\lambda}}\right) + R_{\boldsymbol{\lambda}}
\end{aligned}
$$

Now we have to pick $T$ and $\eta$ of Algorithm **AIF-Learn** such that $R_{\boldsymbol{\lambda}} \leq \nu$ where $\nu$ is the approximation parameter input to the algorithm. First let $B = \frac{1+2\nu}{\alpha}$ and observe that if $m_0 \geq \frac{2B^2\log(2nT/\delta)}{\nu^2} = \frac{2(1+2\nu)^2\log(2nT/\delta)}{\nu^2\alpha^2}$ (see Assumption A.1 on $m_0$) we have that

$$R_{\boldsymbol{\lambda}} \leq \frac{\nu}{2} + \frac{B\log\left(2n+1\right)}{\eta T} + \eta\left(1+2\alpha\right)^2 B$$

Next, let $T = \frac{16B^2(1+2\alpha)^2\log(2n+1)}{\nu^2}$, and $\eta = \frac{\nu}{4(1+2\alpha)^2 B}$ which makes the last two terms appearing in the RHS of the inequality to sum to $\nu/2$, and accordingly $R_{\boldsymbol{\lambda}} \leq \nu$. Therefore, we have shown that for any $X$, with probability at least $1-\delta$ over the observed labelings $F$,

$$
\begin{aligned}
\mathcal{L}\left(\widehat{\boldsymbol{p}},\widehat{\gamma},\widehat{\boldsymbol{\lambda}}\right) &\leq \mathcal{L}\left(\boldsymbol{p},\gamma,\widehat{\boldsymbol{\lambda}}\right) + \nu \quad \text{for all } \boldsymbol{p} \in \Delta(\mathcal{H})^m, \gamma \in [0,1] \\
\mathcal{L}\left(\widehat{\boldsymbol{p}},\widehat{\gamma},\widehat{\boldsymbol{\lambda}}\right) &\geq \mathcal{L}\left(\widehat{\boldsymbol{p}},\widehat{\gamma},\boldsymbol{\lambda}\right) - \nu \quad \text{for all } \boldsymbol{\lambda} \in \Lambda
\end{aligned}
$$

$\square$

*Proof of Theorem A.3.* Let $(\boldsymbol{p},\gamma)$ be any feasible point of the empirical fair learning problem (note as discussed in the paper there is at least one) and define $\boldsymbol{\lambda}^{\star} \in \Lambda$ to be

$$\boldsymbol{\lambda}^{\star} := \begin{cases} 0 & \text{if } r_{k^{\star}}\left(\widehat{\boldsymbol{p}},\widehat{\gamma};\widehat{\mathcal{Q}}\right) \leq 0 \\ Be_{k^{\star}} & \text{if } r_{k^{\star}}\left(\widehat{\boldsymbol{p}},\widehat{\gamma};\widehat{\mathcal{Q}}\right) > 0 \end{cases}$$

where $k^{\star} = \arg\max_{1\leq k\leq 2n} r_k\left(\widehat{\boldsymbol{p}},\widehat{\gamma};\widehat{\mathcal{Q}}\right)$. Observe that with probability $1-\delta$ over $F$,

$$
\begin{aligned}
\mathcal{L}\left(\widehat{\boldsymbol{p}},\widehat{\gamma},\widehat{\boldsymbol{\lambda}}\right) &\leq \mathcal{L}\left(\boldsymbol{p},\gamma,\widehat{\boldsymbol{\lambda}}\right) + \nu \quad \text{(by Theorem A.2)} \\
&= \text{err}\left(\boldsymbol{p};\widehat{\mathcal{P}},\widehat{\mathcal{Q}}\right) + \widehat{\boldsymbol{\lambda}}^{\top}\boldsymbol{r}\left(\boldsymbol{p},\gamma;\widehat{\mathcal{Q}}\right) + \nu \\
&\leq \text{err}\left(\boldsymbol{p};\widehat{\mathcal{P}},\widehat{\mathcal{Q}}\right) + \nu
\end{aligned}
$$

and that,

$$\mathcal{L}\left(\widehat{\boldsymbol{p}}, \widehat{\gamma}, \widehat{\boldsymbol{\lambda}}\right) \geq \mathcal{L}\left(\widehat{\boldsymbol{p}}, \widehat{\gamma}, \boldsymbol{\lambda}^{\star}\right) - \nu \quad \text{(by Theorem A.2)}$$

$$= \text{err}\left(\widehat{\boldsymbol{p}}; \widehat{\mathcal{P}}, \widehat{\mathcal{Q}}\right) + (\boldsymbol{\lambda}^{\star})^{\top} \boldsymbol{r}\left(\widehat{\boldsymbol{p}}, \widehat{\gamma}; \widehat{\mathcal{Q}}\right) - \nu$$

$$\geq \text{err}\left(\widehat{\boldsymbol{p}}; \widehat{\mathcal{P}}, \widehat{\mathcal{Q}}\right) - \nu$$

Combining the above upper and lower bounds on $\mathcal{L}\left(\widehat{\boldsymbol{p}}, \widehat{\gamma}, \widehat{\boldsymbol{\lambda}}\right)$ implies

$$\text{err}\left(\widehat{\boldsymbol{p}}; \widehat{\mathcal{P}}, \widehat{\mathcal{Q}}\right) \leq \text{err}\left(\boldsymbol{p}; \widehat{\mathcal{P}}, \widehat{\mathcal{Q}}\right) + 2\nu$$

Now let's prove the bound on fairness violation. Once again using the above upper and lower bounds, we have that

$$(\boldsymbol{\lambda}^{\star})^{\top} \boldsymbol{r}\left(\widehat{\boldsymbol{p}}, \widehat{\gamma}; \widehat{\mathcal{Q}}\right) \leq \text{err}\left(\boldsymbol{p}; \widehat{\mathcal{P}}, \widehat{\mathcal{Q}}\right) - \text{err}\left(\widehat{\boldsymbol{p}}; \widehat{\mathcal{P}}, \widehat{\mathcal{Q}}\right) + 2\nu \leq 1 + 2\nu$$

By definition of $\boldsymbol{\lambda}^{\star}$,

$$\max_{1 \leq k \leq 2n} r_k\left(\widehat{\boldsymbol{p}}, \widehat{\gamma}; \widehat{\mathcal{Q}}\right) \leq \frac{1 + 2\nu}{B}$$

which implies for all $1 \leq i \leq n$,

$$\left|\mathcal{E}\left(x_i, \widehat{\boldsymbol{p}}; \widehat{\mathcal{Q}}\right) - \widehat{\gamma}\right| \leq 2\alpha + \frac{1 + 2\nu}{B}$$

And the proof is completed by the choice of $B = (1 + 2\nu)/\alpha$ in Algorithm **AIF-Learn**. $\square$

### C.2.2 Proof of theorem: Generalization over $\mathcal{P}$

When arguing about generalization over $\mathcal{P}$, we will have to come up with a *uniform convergence* result for all – randomized – classifiers. It is usually the case in learning theory – for example when considering the concentration of the classifiers' errors – that a uniform convergence for pure classifiers will immediately imply the uniform convergence for randomized classifiers by simply pulling out the expectation over classifiers. However, in our setting and in particular for our notation of fairness, this approach "may" not be possible. We will therefore consider directly arguing about uniform convergence of randomized classifiers. Although there are possibly infinitely many of those randomized classifiers, we actually need to consider only the $\widetilde{O}\left(1/\alpha^2\right)$-sparse classifiers by which we mean the distributions over $\mathcal{H}$ with support of size at most $\widetilde{O}\left(1/\alpha^2\right)$. This $1/\alpha^2$ factor will accordingly show up in our final sample complexity bound for $n$.

In Lemma C.4 we prove a uniform convergence of $r$-sparse randomized classifiers (defined formally below) and later in the proof of our theorem, we invoke this lemma for $r = \widetilde{O}\left(1/\alpha^2\right)$ to prove our result.

**Definition C.1** (Sparse Randomized Classifiers). *We say a randomized classifier $p \in \Delta(\mathcal{H})$ is $r$-sparse if the support of $p$ is of size at most $r$. We denote the set of all $r$-sparse randomized classifiers of the hypothesis class $\mathcal{H}$ by $\Delta_r(\mathcal{H})$, and the elements of $\Delta_r(\mathcal{H})$ by $p^{(r)}$.*

**Lemma C.4** (Uniform Convergence of AIF Fairness Notion in $\Delta_r(\mathcal{H})^m$). *Let $0 < \delta < 1$ and $r \in \mathbb{N}$ and let $T$ be the number of iterations in Algorithm **AIF-Learn**. Suppose*

$$n \geq \widetilde{O}\left(\frac{r\, m\, d_{\mathcal{H}} + \log(T/\delta)}{\beta^2}\right)$$

*We have that for any set of problems $F$, with probability at least $1 - \delta$ over the individuals $X$: for all $\boldsymbol{p}^{(r)} \in \Delta_r(\mathcal{H})^m$ and all $\gamma \in \left\{0, \frac{1}{T}, \frac{2}{T}, \ldots, 1\right\}$,*

$$\left|\Pr_{x \sim \mathcal{P}}\left(\left|\mathcal{E}\left(x, \boldsymbol{p}^{(r)}; \widehat{\mathcal{Q}}\right) - \gamma\right| > \alpha\right) - \Pr_{x \sim \widehat{\mathcal{P}}}\left(\left|\mathcal{E}\left(x, \boldsymbol{p}^{(r)}; \widehat{\mathcal{Q}}\right) - \gamma\right| > \alpha\right)\right| \leq \beta$$

*Proof of Lemma C.4.* The proof of this Lemma will use standard techniques for proving uniform convergence in learning theory such as the "two-sample trick" and Sauer's Lemma C.2. To simplify notation, let's call, for any $\boldsymbol{p}^{(r)} \in \Delta_r(\mathcal{H})^m$ and any $\gamma \in S_T := \left\{0, \frac{1}{T}, \frac{2}{T}, \ldots, 1\right\}$,

$$\mathbb{P}_{x \sim \mathcal{P}}\left(\left|\mathcal{E}\left(x, \boldsymbol{p}^{(r)}; \widehat{\mathcal{Q}}\right) - \gamma\right| > \alpha\right) := g\left(\boldsymbol{p}^{(r)}, \gamma; \mathcal{P}\right) \tag{11}$$

Define event $A(X)$ as follows:

$$A(X) = \left\{\exists \boldsymbol{p}^{(r)} \in \Delta_r(\mathcal{H})^m, \gamma \in S_T : \left|g\left(\boldsymbol{p}^{(r)}, \gamma; \mathcal{P}\right) - g\left(\boldsymbol{p}^{(r)}, \gamma; \widehat{\mathcal{P}}_X\right)\right| > \beta\right\}$$

where $\widehat{\mathcal{P}}_X \equiv \widehat{\mathcal{P}}$ represents the uniform distribution over $X$. Our ultimate goal is to show that given the sample complexity for $n$ stated in the lemma, $\mathbb{P}_X[A(X)]$ is small. Suppose besides the original data set $X$, we also have an "imaginary" data set of individuals $X' = \{x'_i\}_{i=1}^n \in \mathcal{X}^n$ sampled *i.i.d.* from the distribution $\mathcal{P}$. Define event $B(X, X')$ as follows:

$$B(X, X') = \left\{\exists \boldsymbol{p}^{(r)} \in \Delta_r(\mathcal{H})^m, \gamma \in S_T : \left|g\left(\boldsymbol{p}^{(r)}, \gamma; \widehat{\mathcal{P}}_{X'}\right) - g\left(\boldsymbol{p}^{(r)}, \gamma; \widehat{\mathcal{P}}_X\right)\right| > \frac{\beta}{2}\right\}$$

**Claim**: $\mathbb{P}_X[A(X)] \leq 2\mathbb{P}_{(X,X')}[B(X, X')]$. Proof: It suffices to show that

$$\mathbb{P}_{(X,X')}[B(X, X') \mid A(X)] \geq 1/2$$

because

$$\mathbb{P}_{(X,X')}[B(X, X')] \geq \mathbb{P}_{(X,X')}[B(X, X') \mid A(X)]\,\mathbb{P}_X[A(X)]$$

Let the event $A(X)$ hold and suppose the pair $\boldsymbol{p}_\star^{(r)}$ and $\gamma_\star$ satisfy

$$\left|g\left(\boldsymbol{p}_\star^{(r)}, \gamma_\star; \mathcal{P}\right) - g\left(\boldsymbol{p}_\star^{(r)}, \gamma_\star; \widehat{\mathcal{P}}_X\right)\right| > \beta$$

We have that by the triangle inequality and a Chernoff-Hoeffding's bound:

$\mathbb{P}_{(X,X')}[B(X, X') \mid A(X)]$

$$\geq \mathbb{P}_{(X,X')}\left[\left|g\left(\boldsymbol{p}_\star^{(r)}, \gamma_\star; \widehat{\mathcal{P}}_{X'}\right) - g\left(\boldsymbol{p}_\star^{(r)}, \gamma_\star; \widehat{\mathcal{P}}_X\right)\right| > \frac{\beta}{2}\right]$$

$$\geq \mathbb{P}_{(X,X')}\left[\left|g\left(\boldsymbol{p}_\star^{(r)}, \gamma_\star; \mathcal{P}\right) - g\left(\boldsymbol{p}_\star^{(r)}, \gamma_\star; \widehat{\mathcal{P}}_X\right)\right| - \left|g\left(\boldsymbol{p}_\star^{(r)}, \gamma_\star; \mathcal{P}\right) - g\left(\boldsymbol{p}_\star^{(r)}, \gamma_\star; \widehat{\mathcal{P}}_{X'}\right)\right| > \frac{\beta}{2}\right]$$

$$\geq \mathbb{P}_{X'}\left[\left|g\left(\boldsymbol{p}_\star^{(r)}, \gamma_\star; \mathcal{P}\right) - g\left(\boldsymbol{p}_\star^{(r)}, \gamma_\star; \widehat{\mathcal{P}}_{X'}\right)\right| < \frac{\beta}{2}\right]$$

$$\geq 1 - 2e^{-n\frac{\beta^2}{2}}$$

$$\geq 1/2$$

Following the claim, it now suffices to show that $\mathbb{P}_{(X,X')}[B(X, X')]$ is small. Consider the following thought experiment: Let $T$ and $T'$ be two empty sets. For each $i \in [n]$ toss a fair coin independently and

- if it lands on Heads, put $x_i$ in $T$ and $x'_i$ in $T'$.
- if it lands on Tails, put $x'_i$ in $T$ and $x_i$ in $T'$.

We will later denote the randomness induced by these coin flips by "coin" in our probability statements. It follows immediately by our construction that the distribution of $(T, T')$ is the same as the distribution of $(X, X')$, and therefore

$$\mathbb{P}_{(X,X')}[B(X, X')] = \mathbb{P}_{(T,T')}[B(T, T')] = \mathbb{P}_{(X,X',\text{coin})}[B(T, T')] \tag{12}$$

where

$$B(T, T') = \left\{\exists \boldsymbol{p}^{(r)} \in \Delta_r(\mathcal{H})^m, \gamma \in S_T : \left|g\left(\boldsymbol{p}^{(r)}, \gamma; \widehat{\mathcal{P}}_{T'}\right) - g\left(\boldsymbol{p}^{(r)}, \gamma; \widehat{\mathcal{P}}_T\right)\right| > \frac{\beta}{2}\right\}$$

But we have that

$$\mathbb{P}_{(X,X',\text{coin})}[B(T, T')] = \mathbb{E}_{(X,X')}[\mathbb{P}_{\text{coin}}[B(T, T')]] \tag{13}$$

where we use the fact that the coin flips are independent of the random variables $(X, X')$ and thus, conditioning on $(X, X')$ won't change the distribution of "coin". Following Equations 12 and 13, it now suffices to show that for – any $(X, X')$ – $\mathbb{P}_{\text{coin}}[B(T, T')]$ is small. Fix the data sets $(X, X')$. Fix $\boldsymbol{p}^{(r)} \in \Delta_r(\mathcal{H})^m$ and $\gamma \in S_T$ and let

$$ I = \left\{ i \in [n] : g\left(\boldsymbol{p}^{(r)}, \gamma; x_i\right) \neq g\left(\boldsymbol{p}^{(r)}, \gamma; x_i'\right) \right\} $$

where recall that by Equation 11:

$$ g\left(\boldsymbol{p}^{(r)}, \gamma; x_i\right) = \mathbb{1}\left[\left|\mathcal{E}\left(x_i, \boldsymbol{p}^{(r)}; \widehat{\mathcal{Q}}\right) - \gamma\right| > \alpha\right] $$

Let $|I| = n' \leq n$. Now observe that

$$ \mathbb{P}_{\text{coin}}\left[\left|g\left(\boldsymbol{p}^{(r)}, \gamma; \widehat{\mathcal{P}}_{T'}\right) - g\left(\boldsymbol{p}^{(r)}, \gamma; \widehat{\mathcal{P}}_T\right)\right| > \frac{\beta}{2}\right] = \mathbb{P}_{\text{coin}}\left[|\text{heads}(I) - \text{tails}(I)| > \frac{n\beta}{2}\right] $$

where $\text{heads}(I)$ and $\text{tails}(I)$ denote the number of heads and tails of the coin on indices $I$. But a triangle inequality followed by two Chernoff-Hoeffding bounds imply

$$
\begin{aligned}
\mathbb{P}_{\text{coin}}\left[|\text{heads}(I) - \text{tails}(I)| > \frac{n\beta}{2}\right] &\leq \mathbb{P}_{\text{coin}}\left[\left|\frac{\text{heads}(I)}{n'} - \frac{1}{2}\right| > \frac{n\beta}{4n'}\right] \\
&\quad + \mathbb{P}_{\text{coin}}\left[\left|\frac{1}{2} - \frac{\text{tails}(I)}{n'}\right| > \frac{n\beta}{4n'}\right] \\
&\leq 4e^{-2n'(n\beta/4n')^2} \\
&\leq 4e^{-n\beta^2/8}
\end{aligned}
$$

Therefore, we have proved that for any $\boldsymbol{p}^{(r)} \in \Delta_r(\mathcal{H})^m$ and any $\gamma \in S_T$,

$$ \mathbb{P}_{\text{coin}}\left[\left|g\left(\boldsymbol{p}^{(r)}, \gamma; \widehat{\mathcal{P}}_{T'}\right) - g\left(\boldsymbol{p}^{(r)}, \gamma; \widehat{\mathcal{P}}_T\right)\right| > \frac{\beta}{2}\right] \leq 4e^{-n\beta^2/8} $$

Now it's time to apply the Sauer's Lemma C.2 to get a uniform convergence for all $\boldsymbol{p}^{(r)} \in \Delta_r(\mathcal{H})^m$ and all $\gamma \in S_T$. Notice once the data sets $(X, X')$ are fixed, there are at most $O\left((2n)^{d_{\mathcal{H}} r m}\right)$ number of randomized classifiers $\boldsymbol{p}^{(r)} \in \Delta_r(\mathcal{H})^m$ induced on the set $\{X, X'\}$. Here $2n$ comes from the fact that $X$ and $X'$ have a combined number of $2n$ points. $d_{\mathcal{H}}$ is the VC dimension, and $r$ and $m$ show up in the bound because each $p_j^{(r)}$ in $\boldsymbol{p}^{(r)}$ is $r$-sparse and that there are a total $m$ randomized classifiers $p_j^{(r)}$ in $\boldsymbol{p}^{(r)}$. We also have that $|S_T| = T + 1$. Consequently by a union bound over all induced $\boldsymbol{p}^{(r)}$ and all $\gamma \in S_T$:

$$
\begin{aligned}
\mathbb{P}_{\text{coin}}[B(T, T')] = \mathbb{P}_{\text{coin}}&\left[\exists \boldsymbol{p}^{(r)} \in \Delta_r(\mathcal{H})^m, \gamma \in S_T : \left|g\left(\boldsymbol{p}^{(r)}, \gamma; \widehat{\mathcal{P}}_{T'}\right) - g\left(\boldsymbol{p}^{(r)}, \gamma; \widehat{\mathcal{P}}_T\right)\right| > \frac{\beta}{2}\right] \\
&\leq \sum_{\boldsymbol{p}^{(r)}, \gamma} \mathbb{P}_{\text{coin}}\left[\left|g\left(\boldsymbol{p}^{(r)}, \gamma; \widehat{\mathcal{P}}_{T'}\right) - g\left(\boldsymbol{p}^{(r)}, \gamma; \widehat{\mathcal{P}}_T\right)\right| > \frac{\beta}{2}\right] \\
&\leq O\left((2n)^{d_{\mathcal{H}} r m}\right) \cdot (T + 1) \cdot 4e^{-n\beta^2/8}
\end{aligned}
$$

where the second sum is actually over all the induced $r$-sparse randomized classifiers on the set $\{X, X'\}$ and all $\gamma \in S_T$. This shows so long as

$$ n \geq \widetilde{O}\left(\frac{r\, m\, d_{\mathcal{H}} + \log(T/\delta)}{\beta^2}\right) $$

we have that for any $(X, X')$, $\mathbb{P}_{\text{coin}}[B(T, T')] \leq \delta/2$. Therefore

$$ \mathbb{P}_X[A(X)] \leq 2\,\mathbb{P}_{(X, X')}[B(X, X')] = 2\,\mathbb{E}_{(X, X')}[\mathbb{P}_{\text{coin}}[B(T, T')]] \leq \delta $$

completing the proof. $\qquad\square$

*Proof of Theorem 3.2 (Generalization over $\mathcal{P}$).* We will use Lemma C.4 in the proof of this theorem. We are interested in the fairness violation and accuracy of the mapping $\widehat{\psi} = \widehat{\psi}(X, F)$ returned by Algorithm **AIF-Learn** with respect to the distributions $\mathcal{P}$ and $\widehat{\mathcal{Q}}$. Fix the set of problems $F$ and observe that when we work with the empirical distribution of the problems $\widehat{\mathcal{Q}}$, we have that

$$\mathcal{E}\left(x, \widehat{\psi}; \widehat{\mathcal{Q}}\right) = \mathcal{E}\left(x, \widehat{\boldsymbol{p}}; \widehat{\mathcal{Q}}\right) \quad , \quad \mathrm{err}\left(\widehat{\psi}; \mathcal{P}, \widehat{\mathcal{Q}}\right) = \mathrm{err}\left(\widehat{\boldsymbol{p}}; \mathcal{P}, \widehat{\mathcal{Q}}\right)$$

where $\widehat{\boldsymbol{p}}$ is the set of $m$ randomized classifiers output by Algorithm **AIF-Learn**. Let's first prove the generalization for fairness. Define event $A(X)$:

$$A(X) = \left\{ \mathbb{P}_{x \sim \mathcal{P}}\left(\left|\mathcal{E}\left(x, \widehat{\boldsymbol{p}}; \widehat{\mathcal{Q}}\right) - \widehat{\gamma}\right| > 5\alpha\right) - \mathbb{P}_{x \sim \widehat{\mathcal{P}}}\left(\left|\mathcal{E}\left(x, \widehat{\boldsymbol{p}}; \widehat{\mathcal{Q}}\right) - \widehat{\gamma}\right| > 3\alpha\right) > \beta \right\}$$

We will eventually show that under the stated sample complexity for $n$ in the theorem, $\mathbb{P}_X[A(X)]$ is small. Consider the distributions $\widehat{\boldsymbol{p}} = (\widehat{p}_1, \widehat{p}_2, \ldots, \widehat{p}_m)$ over $\mathcal{H}$. Let

$$r = \frac{\log(12nm/\delta)}{2\alpha^2}$$

For each $j \in [m]$, consider drawing $r$ independent samples from the distribution $\widehat{p}_j$ over $\mathcal{H}$ and define $\widehat{p}_j^{(r)}$ to be the uniform distribution over the drawn samples. We will abuse notation and use $\widehat{p}_j^{(r)}$ to denote both the drawn samples and the uniform distribution over them. Now define

$$\widehat{\boldsymbol{p}}^{(r)} = \left(\widehat{p}_1^{(r)}, \widehat{p}_2^{(r)}, \ldots, \widehat{p}_m^{(r)}\right) \in \Delta_r(\mathcal{H})^m$$

which is the "$r$-sparsified" version $\widehat{\boldsymbol{p}}$. One important observation that follows from the Chernoff-Hoeffding's theorem is that for – any – $X' = \{x_i'\}_{i=1}^n \in \mathcal{X}^n$, with probability at least $1 - \delta/6$ over the draws of $\widehat{\boldsymbol{p}}^{(r)}$, we have that for all $i \in [n]$:

$$\left|\mathcal{E}\left(x_i', \widehat{\boldsymbol{p}}; \widehat{\mathcal{Q}}\right) - \mathcal{E}\left(x_i', \widehat{\boldsymbol{p}}^{(r)}; \widehat{\mathcal{Q}}\right)\right| \leq \alpha \tag{14}$$

In other words, for any set of individuals $X' \in \mathcal{X}^n$, we have that:

$$\mathbb{P}_{\widehat{\boldsymbol{p}}^{(r)}}\left[\mathbb{P}_{x \sim \widehat{\mathcal{P}}'}\left(\left|\mathcal{E}\left(x, \widehat{\boldsymbol{p}}; \widehat{\mathcal{Q}}\right) - \mathcal{E}\left(x, \widehat{\boldsymbol{p}}^{(r)}; \widehat{\mathcal{Q}}\right)\right| > \alpha\right) \neq 0\right] \leq \delta/6 \tag{15}$$

where $\widehat{\mathcal{P}}'$ denotes the uniform distribution over $X'$. Now define events $B, C, D$ as follows:

$$B\left(X, \widehat{\boldsymbol{p}}^{(r)}\right) = \left\{\left|\mathbb{P}_{x \sim \mathcal{P}}\left(\left|\mathcal{E}\left(x, \widehat{\boldsymbol{p}}^{(r)}; \widehat{\mathcal{Q}}\right) - \widehat{\gamma}\right| > 4\alpha\right) - \mathbb{P}_{x \sim \widehat{\mathcal{P}}}\left(\left|\mathcal{E}\left(x, \widehat{\boldsymbol{p}}^{(r)}; \widehat{\mathcal{Q}}\right) - \widehat{\gamma}\right| > 4\alpha\right)\right| > \frac{\beta}{2}\right\}$$

$$C\left(X, \widehat{\boldsymbol{p}}^{(r)}\right) = \left\{\mathbb{P}_{x \sim \mathcal{P}}\left(\left|\mathcal{E}\left(x, \widehat{\boldsymbol{p}}; \widehat{\mathcal{Q}}\right) - \mathcal{E}\left(x, \widehat{\boldsymbol{p}}^{(r)}; \widehat{\mathcal{Q}}\right)\right| > \alpha\right) > \frac{\beta}{2}\right\}$$

$$D\left(X, \widehat{\boldsymbol{p}}^{(r)}\right) = \left\{\mathbb{P}_{x \sim \widehat{\mathcal{P}}}\left(\left|\mathcal{E}\left(x, \widehat{\boldsymbol{p}}; \widehat{\mathcal{Q}}\right) - \mathcal{E}\left(x, \widehat{\boldsymbol{p}}^{(r)}; \widehat{\mathcal{Q}}\right)\right| > \alpha\right) \neq 0\right\}$$

It follows by the triangle inequality that

$$\mathbb{P}_X[A(X)] = \mathbb{P}_{X, \widehat{\boldsymbol{p}}^{(r)}}[A(X)]$$

$$\leq \underbrace{\mathbb{P}_{X, \widehat{\boldsymbol{p}}^{(r)}}\left[B\left(X, \widehat{\boldsymbol{p}}^{(r)}\right)\right]}_{\text{term 1}} + \underbrace{\mathbb{P}_{X, \widehat{\boldsymbol{p}}^{(r)}}\left[C\left(X, \widehat{\boldsymbol{p}}^{(r)}\right)\right]}_{\text{term 2}} + \underbrace{\mathbb{P}_{X, \widehat{\boldsymbol{p}}^{(r)}}\left[D\left(X, \widehat{\boldsymbol{p}}^{(r)}\right)\right]}_{\text{term 3}}$$

So to prove that $\mathbb{P}_X[A(X)]$ is small, it suffices to show that all the three terms appearing in the RHS of the above inequality are small. We will in fact show each term $\leq \delta/3$:

**term 1**: Notice to bound this term we need to prove a *uniform convergence* for all $r$-sparse set of randomized classifiers $\boldsymbol{p}^{(r)} \in \Delta_r(\mathcal{H})^m$ and all $\gamma$ of the form $c/T$ for some nonnegative integer $c \leq T$ (because the $\widehat{\gamma}$ output by our algorithm has this form). But we already proved this uniform convergence in Lemma C.4. In fact if we define the event:

$$B_1(X) = \left\{\exists \boldsymbol{p}^{(r)} \in \Delta_r(\mathcal{H})^m, \gamma \in \left\{0, \frac{1}{T}, \frac{2}{T}, \ldots, 1\right\} : \right.$$

$$\left.\left|\mathbb{P}_{x \sim \mathcal{P}}\left(\left|\mathcal{E}\left(x, \boldsymbol{p}^{(r)}; \widehat{\mathcal{Q}}\right) - \gamma\right| > 4\alpha\right) - \mathbb{P}_{x \sim \widehat{\mathcal{P}}}\left(\left|\mathcal{E}\left(x, \boldsymbol{p}^{(r)}; \widehat{\mathcal{Q}}\right) - \gamma\right| > 4\alpha\right)\right| > \frac{\beta}{2}\right\}$$

by Lemma C.4, so long as,

$$n \geq \widetilde{O}\left(\frac{r\,m\,d_{\mathcal{H}} + \log\left(T/\delta\right)}{\beta^2}\right) \equiv \widetilde{O}\left(\frac{m\,d_{\mathcal{H}} + \log\left(1/\nu^2\delta\right)}{\alpha^2\beta^2}\right) \quad (T = O\left(\log\left(n\right)/\nu^2\alpha^2\right))$$

we have that $\mathbb{P}_X\left[B_1(X)\right] \leq \delta/3$. And this implies

$$\mathbb{P}_{X,\widehat{\boldsymbol{p}}^{(r)}}\left[B\left(X,\widehat{\boldsymbol{p}}^{(r)}\right)\right] \leq \mathbb{P}_X\left[B_1\left(X\right)\right] \leq \delta/3$$

**term 2**: Observe that

$$\mathbb{P}_{X,\widehat{\boldsymbol{p}}^{(r)}}\left[C\left(X,\widehat{\boldsymbol{p}}^{(r)}\right)\right] = \mathbb{E}_X\left[\mathbb{P}_{\widehat{\boldsymbol{p}}^{(r)}}\left[C\left(X,\widehat{\boldsymbol{p}}^{(r)}\right)\right]\right]$$

Notice when we condition on $X$, the conditional distribution of both $\widehat{\boldsymbol{p}}$ and $\widehat{\boldsymbol{p}}^{(r)}$ will actually change because they both depend on $X$, but it is the case that for any $X$, $\widehat{\boldsymbol{p}}^{(r)}$ (given $X$) will be still independent draws from $\widehat{\boldsymbol{p}}$ (given $X$) and this is what we will actually need. We will show for any $X$:

$$\mathbb{P}_{\widehat{\boldsymbol{p}}^{(r)}}\left[C\left(X,\widehat{\boldsymbol{p}}^{(r)}\right)\right] \leq \delta/3$$

Fix $X$. Let $X' = \{x_i'\}_{i=1}^n$ be a new data set of individuals drawn independently from the distribution $\mathcal{P}$ and let $\widehat{\mathcal{P}}'$ denote the uniform distribution over $X'$. Define events:

$$C_1\left(X',\widehat{\boldsymbol{p}}^{(r)}\right) =$$

$$\left\{\left|\mathbb{P}_{x\sim\mathcal{P}}\left(\left|\mathcal{E}\left(x,\widehat{\boldsymbol{p}};\widehat{\mathcal{Q}}\right) - \mathcal{E}\left(x,\widehat{\boldsymbol{p}}^{(r)};\widehat{\mathcal{Q}}\right)\right| > \alpha\right) - \mathbb{P}_{x\sim\widehat{\mathcal{P}}'}\left(\left|\mathcal{E}\left(x,\widehat{\boldsymbol{p}};\widehat{\mathcal{Q}}\right) - \mathcal{E}\left(x,\widehat{\boldsymbol{p}}^{(r)};\widehat{\mathcal{Q}}\right)\right| > \alpha\right)\right| > \frac{\beta}{2}\right\}$$

$$C_2\left(X',\widehat{\boldsymbol{p}}^{(r)}\right) = \left\{\mathbb{P}_{x\sim\widehat{\mathcal{P}}'}\left(\left|\mathcal{E}\left(x,\widehat{\boldsymbol{p}};\widehat{\mathcal{Q}}\right) - \mathcal{E}\left(x,\widehat{\boldsymbol{p}}^{(r)};\widehat{\mathcal{Q}}\right)\right| > \alpha\right) \neq 0\right\}$$

It follows by a triangle inequality that:

$$\mathbb{P}_{\widehat{\boldsymbol{p}}^{(r)}}\left[C\left(X,\widehat{\boldsymbol{p}}^{(r)}\right)\right] = \mathbb{P}_{X',\widehat{\boldsymbol{p}}^{(r)}}\left[C\left(X,\widehat{\boldsymbol{p}}^{(r)}\right)\right]$$

$$\leq \mathbb{P}_{X',\widehat{\boldsymbol{p}}^{(r)}}\left[C_1\left(X',\widehat{\boldsymbol{p}}^{(r)}\right)\right] + \mathbb{P}_{X',\widehat{\boldsymbol{p}}^{(r)}}\left[C_2\left(X',\widehat{\boldsymbol{p}}^{(r)}\right)\right]$$

$$= \mathbb{E}_{\widehat{\boldsymbol{p}}^{(r)}}\left[\mathbb{P}_{X'}\left[C_1\left(X',\widehat{\boldsymbol{p}}^{(r)}\right)\right]\right] + \mathbb{E}_{X'}\left[\mathbb{P}_{\widehat{\boldsymbol{p}}^{(r)}}\left[C_2\left(X',\widehat{\boldsymbol{p}}^{(r)}\right)\right]\right]$$

Given $\widehat{\boldsymbol{p}}^{(r)}$, we have that by a Chernoff-Hoeffding's inequality that as long as the sample complexity for $n$ is met:

$$\mathbb{P}_{X'}\left[C_1\left(X',\widehat{\boldsymbol{p}}^{(r)}\right)\right] \leq \delta/6$$

On the other hand, given $X'$, it follows by Equation 15 that,

$$\mathbb{P}_{\widehat{\boldsymbol{p}}^{(r)}}\left[C_2\left(X',\widehat{\boldsymbol{p}}^{(r)}\right)\right] \leq \delta/6$$

Consequently:

$$\mathbb{P}_{X,\widehat{\boldsymbol{p}}^{(r)}}\left[C\left(X,\widehat{\boldsymbol{p}}^{(r)}\right)\right] \leq \delta/3$$

**term 3**: Observe that Equation 15 implies:

$$\mathbb{P}_{X,\widehat{\boldsymbol{p}}^{(r)}}\left[D\left(X,\widehat{\boldsymbol{p}}^{(r)}\right)\right] = \mathbb{E}_X\left[\mathbb{P}_{\widehat{\boldsymbol{p}}^{(r)}}\left[D\left(X,\widehat{\boldsymbol{p}}^{(r)}\right)\right]\right] \leq \delta/3$$

We finally proved $\mathbb{P}_X\left[A\left(X\right)\right] \leq \delta$. Or in other words, we have proved that for any $F$, with probability at least $1 - \delta$ over the individuals $X$,

$$\mathbb{P}_{x\sim\mathcal{P}}\left(\left|\mathcal{E}\left(x,\widehat{\boldsymbol{p}};\widehat{\mathcal{Q}}\right) - \widehat{\gamma}\right| > 5\alpha\right) - \mathbb{P}_{x\sim\widehat{\mathcal{P}}}\left(\left|\mathcal{E}\left(x,\widehat{\boldsymbol{p}};\widehat{\mathcal{Q}}\right) - \widehat{\gamma}\right| > 3\alpha\right) \leq \beta$$

On the other hand, the in-sample guarantees provided in Theorem A.3 implies for any $X$, with probability $1 - \delta$ over the observed problems $F$, the pair $(\widehat{\boldsymbol{p}}, \widehat{\gamma})$ of Algorithm **AIF-Learn** satisfies

$$\mathbb{P}_{x\sim\widehat{\mathcal{P}}}\left(\left|\mathcal{E}\left(x,\widehat{\boldsymbol{p}};\widehat{\mathcal{Q}}\right) - \widehat{\gamma}\right| > 3\alpha\right) = 0$$

Hence, as long as

$$n \geq \widetilde{O}\left(\frac{m\, d_{\mathcal{H}} + \log\left(1/\nu^2 \delta\right)}{\alpha^2 \beta^2}\right) \tag{16}$$

we have that with probability at least $1 - 2\delta$ over the observed data set $(X, F)$,

$$\mathbb{P}_{x \sim \mathcal{P}}\left(\left|\mathcal{E}\left(x, \widehat{\boldsymbol{p}}; \widehat{\mathcal{Q}}\right) - \widehat{\gamma}\right| > 5\alpha\right) \leq \beta$$

which shows $\widehat{\boldsymbol{p}}$, or equivalently the mapping $\widehat{\psi}$, is $(5\alpha, \beta)$-AIF with respect to the distributions $\mathcal{P}$ and $\widehat{\mathcal{Q}}$. Now let's look at the generalization (with respect to $\mathcal{P}$) error of the set of classifiers $\widehat{\boldsymbol{p}}$. Let

$$\psi^\star = \psi^\star\left(\alpha; \mathcal{P}, \widehat{\mathcal{Q}}\right), \quad \gamma^\star = \gamma^\star\left(\alpha; \mathcal{P}, \widehat{\mathcal{Q}}\right), \quad \text{err}\left(\psi^\star; \mathcal{P}, \widehat{\mathcal{Q}}\right) = \text{OPT}\left(\alpha; \mathcal{P}, \widehat{\mathcal{Q}}\right)$$

be the optimal solutions of the fair learning problem. Let also $\psi^\star|_F = \boldsymbol{p}^\star \in \Delta(\mathcal{H})^m$. Since we are working with the empirical distribution of problems $\widehat{\mathcal{Q}}$,

$$\text{err}\left(\boldsymbol{p}^\star; \mathcal{P}, \widehat{\mathcal{Q}}\right) = \text{err}\left(\psi^\star; \mathcal{P}, \widehat{\mathcal{Q}}\right) = \text{OPT}\left(\alpha; \mathcal{P}, \widehat{\mathcal{Q}}\right)$$

We would like to compare the accuracy of $\widehat{\boldsymbol{p}}$ with respect to the distributions $\mathcal{P}$ and $\widehat{\mathcal{Q}}$, i.e. $\text{err}\left(\widehat{\boldsymbol{p}}; \mathcal{P}, \widehat{\mathcal{Q}}\right)$, to $\text{OPT}\left(\alpha; \mathcal{P}, \widehat{\mathcal{Q}}\right)$ as a benchmark. We will establish this comparison in three steps:

1. a bound on the difference between $\text{err}\left(\widehat{\boldsymbol{p}}; \mathcal{P}, \widehat{\mathcal{Q}}\right)$ and its in-sample version $\text{err}\left(\widehat{\boldsymbol{p}}; \widehat{\mathcal{P}}, \widehat{\mathcal{Q}}\right)$.
   Note a uniform convergence can be achieved for all $\boldsymbol{p} \in \Delta(\mathcal{H})^m$ by standard generalization techniques (like Sauer's Lemma and the two-sample trick, as discussed in the proof of Lemma C.4), without even needing to consider the uniform convergence for $r$-sparse randomized classifiers. Because when investigating the concentration of the overall error rate which is in the form of nested expectations, the linearity of expectation helps us to directly turn a uniform convergence for all sets of pure classifiers $\boldsymbol{h} \in \mathcal{H}^m$ into a uniform convergence for all randomized classifiers $\boldsymbol{p} \in \Delta(\mathcal{H})^m$ without blowing up the sample complexity by a factor of $1/\alpha^2$. We can therefore guarantee that for all $F$, with probability at least $1 - \delta$ over $X$, as long as 16 is satisfied:

$$\left|\text{err}\left(\widehat{\boldsymbol{p}}; \mathcal{P}, \widehat{\mathcal{Q}}\right) - \text{err}\left(\widehat{\boldsymbol{p}}; \widehat{\mathcal{P}}, \widehat{\mathcal{Q}}\right)\right| \leq O\left(\alpha\beta\right) \tag{17}$$

2. a bound on the difference between $\text{err}\left(\boldsymbol{p}^\star; \mathcal{P}, \widehat{\mathcal{Q}}\right)$ and its in-sample version $\text{err}\left(\boldsymbol{p}^\star; \widehat{\mathcal{P}}, \widehat{\mathcal{Q}}\right)$. This is very simple and it follows immediately from the Chernoff-Hoeffding's theorem that for all $F$, with probability at least $1 - \delta$ over $X$, as long as 16 is satisfied:

$$\left|\text{OPT}\left(\alpha; \mathcal{P}, \widehat{\mathcal{Q}}\right) - \text{err}\left(\boldsymbol{p}^\star; \widehat{\mathcal{P}}, \widehat{\mathcal{Q}}\right)\right| \leq O\left(\alpha\beta\right)$$

3. the pair $(\boldsymbol{p}^\star, \gamma^\star)$ is feasible in the empirical learning problem. As a consequence, Theorem A.3 implies for any $X$, with probability $1 - \delta$ over $F$:

$$\text{err}\left(\widehat{\boldsymbol{p}}; \widehat{\mathcal{P}}, \widehat{\mathcal{Q}}\right) \leq \text{err}\left(\boldsymbol{p}^\star; \widehat{\mathcal{P}}, \widehat{\mathcal{Q}}\right) + 2\nu$$

Finally, putting together all three pieces explained above, we have that with probability at least $1 - 3\delta$ over the observed data set $(X, F)$,

$$\text{err}\left(\widehat{\boldsymbol{p}}; \mathcal{P}, \widehat{\mathcal{Q}}\right) \leq \text{OPT}\left(\alpha; \mathcal{P}, \widehat{\mathcal{Q}}\right) + O\left(\nu\right) + O\left(\alpha\beta\right)$$

completing the proof. $\qquad\square$

### C.2.3 Proof of Theorem: Generalization over $\mathcal{Q}$

*Proof of Theorem 3.3 (Generalization over $\mathcal{Q}$).* Fix the set of observed individuals $X = \{x_i\}_{i=1}^n$ and recall $\widehat{\mathcal{P}}$ denotes the uniform distribution over $X$. Let

$$\widehat{\psi} = \widehat{\psi}\left(X, \widehat{W}\right) \quad \text{where} \quad \widehat{W} = \{\boldsymbol{w}_t\}_{t=1}^T$$

be the mapping returned by Algorithm **AIF-Learn**. We would like to lift the accuracy and fairness guarantees of $\widehat{\psi}$ from the empirical distribution $\widehat{\mathcal{Q}}$ up to the true underlying distribution $\mathcal{Q}$. First observe that by the definition of $\widehat{\psi}$, for any $x \in \mathcal{X}$ and any distribution $Q$:

$$\mathcal{E}\left(x, \widehat{\psi}; \mathcal{Q}\right) = \frac{1}{T}\sum_{t=1}^T \mathcal{E}\left(x, \widehat{\psi}_t; \mathcal{Q}\right) \tag{18}$$

where $\widehat{\psi}_t = \widehat{\psi}\left(X, \boldsymbol{w}_t\right)$ is a mapping defined only the weights $\boldsymbol{w}_t$ of round $t$ over the individuals $X$. $\widehat{\psi}_t$ in fact takes a function $f$, solves one CSC problem on $X$ weighted by $\boldsymbol{w}_t$ and then returns the learned classifier. For any $t \in [T]$, we are interested in bounding the difference

$$\left| \mathcal{E}\left(x, \widehat{\psi}_t; \mathcal{Q}\right) - \mathcal{E}\left(x, \widehat{\psi}_t; \widehat{\mathcal{Q}}\right) \right|$$

Notice following the dynamics of our algorithm, $\boldsymbol{w}_t$ (and accordingly $\widehat{\psi}_t$) has dependence on the previous batches of problems $\{F_{t'}\}_{t'=1}^{t-1}$ and therefore we cannot directly invoke the Chernoff-Hoeffding's theorem to bound the above difference. We instead use the estimates $\mathcal{E}(x, \widehat{\psi}_t; \widehat{\mathcal{Q}}_t)$ where $\widehat{\mathcal{Q}}_t = \mathcal{U}(F_t)$ as an intermediary step to bound the difference. Observe that a simple triangle inequality implies

$$\left| \mathcal{E}\left(x, \widehat{\psi}_t; \mathcal{Q}\right) - \mathcal{E}\left(x, \widehat{\psi}_t; \widehat{\mathcal{Q}}\right) \right| \leq \left| \mathcal{E}\left(x, \widehat{\psi}_t; \mathcal{Q}\right) - \mathcal{E}\left(x, \widehat{\psi}_t; \widehat{\mathcal{Q}}_t\right) \right| + \left| \mathcal{E}\left(x, \widehat{\psi}_t; \widehat{\mathcal{Q}}_t\right) - \mathcal{E}\left(x, \widehat{\psi}_t; \widehat{\mathcal{Q}}\right) \right|$$

Now we can invoke the Chernoff-Hoeffding's Theorem C.1 to bound each term appearing above separately. In fact the batch of problems $F_t$ can be seen as independent draws from both distributions $\mathcal{Q}$ and $\widehat{\mathcal{Q}}$. It therefore follows that with probability $1 - \delta$ over the draws of the batch $F_t$,

$$\left| \mathcal{E}\left(x, \widehat{\psi}_t; \mathcal{Q}\right) - \mathcal{E}\left(x, \widehat{\psi}_t; \widehat{\mathcal{Q}}\right) \right| \leq 2\sqrt{\frac{\log\left(4/\delta\right)}{2m_0}}$$

We can now use Equation 18 to translate this bound to a bound for $\widehat{\psi}$. We have that for any $x \in \mathcal{X}$, with probability at least $1 - \delta$ over all problems $F$,

$$\left| \mathcal{E}\left(x, \widehat{\psi}; \mathcal{Q}\right) - \mathcal{E}\left(x, \widehat{\psi}; \widehat{\mathcal{Q}}\right) \right| \leq 2\sqrt{\frac{\log\left(4T/\delta\right)}{2m_0}} \tag{19}$$

And therefore, with probability at least $1 - \delta$ over all problems $F$, for all $i \in [n]$,

$$\left| \mathcal{E}\left(x_i, \widehat{\psi}; \mathcal{Q}\right) - \mathcal{E}\left(x_i, \widehat{\psi}; \widehat{\mathcal{Q}}\right) \right| \leq 2\sqrt{\frac{\log\left(4nT/\delta\right)}{2m_0}} \leq \nu\alpha \tag{20}$$

where the second inequality follows from Assumption A.1 on $m_0$. Now by a simple triangle inequality

$$\mathbb{P}_{x \sim \widehat{\mathcal{P}}}\left( \left| \mathcal{E}\left(x, \widehat{\psi}; \mathcal{Q}\right) - \widehat{\gamma} \right| > 4\alpha \right)$$

$$\leq \mathbb{P}_{x \sim \widehat{\mathcal{P}}}\left( \left| \mathcal{E}\left(x, \widehat{\psi}; \mathcal{Q}\right) - \mathcal{E}(x, \widehat{\psi}; \widehat{\mathcal{Q}}) \right| > \alpha \right) + \mathbb{P}_{x \sim \widehat{\mathcal{P}}}\left( \left| \mathcal{E}(x, \widehat{\psi}; \widehat{\mathcal{Q}}) - \widehat{\gamma} \right| > 3\alpha \right)$$

The first term appearing in the RHS of the above inequality is zero with probability $1 - \delta$ over $F$ following inequality 20. The second term is zero with probability $1 - \delta$ over $F$ as well following the in-sample guarantees provided in Theorem A.3. Therefore, we conclude that for any set of observed individuals $X$, with probability at least $1 - 2\delta$ over the observed labelings $F$, as long as Assumption A.1:

$$m = T \cdot m_0 \geq O\left( \frac{T\log\left(nT/\delta\right)}{\nu^2\alpha^2} \right) = \widetilde{O}\left( \frac{\log\left(n\right)\log\left(n/\delta\right)}{\nu^4\alpha^4} \right) \tag{21}$$

holds, the mapping $\widehat{\psi}$ satisfies $(4\alpha, 0)$-AIF with respect to the distributions $\left(\widehat{\mathcal{P}}, \mathcal{Q}\right)$:

$$\mathop{\mathbb{P}}_{x \sim \widehat{\mathcal{P}}} \left( \left| \mathcal{E}\left(x, \widehat{\psi}; \mathcal{Q}\right) - \widehat{\gamma} \right| > 4\alpha \right) = 0$$

Now let's look at the generalization (with respect to $\mathcal{Q}$) error of $\widehat{\psi}$. Let

$$\psi^\star = \psi^\star\left(\alpha; \widehat{\mathcal{P}}, \mathcal{Q}\right), \quad \gamma^\star = \gamma^\star\left(\alpha; \widehat{\mathcal{P}}, \mathcal{Q}\right), \quad \mathrm{err}\left(\psi^\star; \widehat{\mathcal{P}}, \mathcal{Q}\right) = \mathrm{OPT}\left(\alpha; \widehat{\mathcal{P}}, \mathcal{Q}\right)$$

be the optimal solutions of the fair learning problem on distributions $\widehat{\mathcal{P}}$ and $\mathcal{Q}$. We would like to compare the accuracy of $\widehat{\psi}$ with respect to the distributions $\widehat{\mathcal{P}}$ and $\mathcal{Q}$, i.e. $\mathrm{err}\left(\widehat{\psi}; \widehat{\mathcal{P}}, \mathcal{Q}\right)$, to $\mathrm{OPT}\left(\alpha; \widehat{\mathcal{P}}, \mathcal{Q}\right)$ as a benchmark. We follow the same approach we used in the previous proof to achieve an upper bound for the difference between $\widehat{\psi}$'s error and the optimal error. It follows directly from inequality 20 that with probability $1 - \delta$ over the problems $F$, as long as the sample complexity 21 is satisfied,

$$\left| \mathrm{err}\left(\widehat{\psi}; \widehat{\mathcal{P}}, \mathcal{Q}\right) - \mathrm{err}\left(\widehat{\psi}; \widehat{\mathcal{P}}, \widehat{\mathcal{Q}}\right) \right| = \left| \frac{1}{n} \sum_{i=1}^{n} \mathcal{E}\left(x_i, \widehat{\psi}; \mathcal{Q}\right) - \frac{1}{n} \sum_{i=1}^{n} \mathcal{E}\left(x_i, \widehat{\psi}; \widehat{\mathcal{Q}}\right) \right| \leq \nu\alpha \quad (22)$$

and that by a simple application of the Chernoff-Hoeffding's inequality, as long as the sample complexity 21 is satisfied, with probability $1 - \delta$ over $F$ we have that

$$\left| \mathrm{err}\left(\psi^\star; \widehat{\mathcal{P}}, \mathcal{Q}\right) - \mathrm{err}\left(\psi^\star; \widehat{\mathcal{P}}, \widehat{\mathcal{Q}}\right) \right| \leq O(\nu\alpha) \quad (23)$$

On the other hand, with probability $1 - \delta$ over $F$, the pair $(\psi^\star|_F, \gamma^\star)$ – where $\psi^\star|_F$ is the mapping $\psi^\star$ restricted to the problems $F$ – is feasible in the empirical learning problem. Because, with probability $1 - \delta$ as long as the sample complexity 21 is satisfied:

$$\left| \mathcal{E}\left(x_i, \psi^\star|_F; \widehat{\mathcal{Q}}\right) - \gamma^\star \right| \leq \left| \mathcal{E}\left(x_i, \psi^\star|_F; \widehat{\mathcal{Q}}\right) - \mathcal{E}\left(x_i, \psi^\star; \mathcal{Q}\right) \right| + \left| \mathcal{E}\left(x_i, \psi^\star; \mathcal{Q}\right) - \gamma^\star \right| \leq \alpha + \alpha = 2\alpha$$

Therefore, by Theorem A.3, we have that with probability $1 - \delta$ over $F$,

$$\mathrm{err}\left(\widehat{\psi}; \widehat{\mathcal{P}}, \widehat{\mathcal{Q}}\right) = \mathrm{err}\left(\widehat{p}; \widehat{\mathcal{P}}, \widehat{\mathcal{Q}}\right) \leq \mathrm{err}\left(\psi^\star|_F; \widehat{\mathcal{P}}, \widehat{\mathcal{Q}}\right) + 2\nu = \mathrm{err}\left(\psi^\star; \widehat{\mathcal{P}}, \widehat{\mathcal{Q}}\right) + 2\nu \quad (24)$$

Putting together inequalities 22, 23, and 24, we conclude that for any $X$, with probability at least $1 - 4\delta$ over the problems $F$,

$$\mathrm{err}\left(\widehat{\psi}; \widehat{\mathcal{P}}, \mathcal{Q}\right) \leq \mathrm{OPT}\left(\alpha; \widehat{\mathcal{P}}, \mathcal{Q}\right) + O(\nu)$$

$\square$

### C.2.4   Proof of theorem: Simultaneous Generalization over $\mathcal{P}$ and $\mathcal{Q}$

We have proved so far:

- Generalization over $\mathcal{P}$: $\left(\widehat{\mathcal{P}}, \widehat{\mathcal{Q}}\right) \xrightarrow{\text{lifted}} \left(\mathcal{P}, \widehat{\mathcal{Q}}\right)$.

- Generalization over $\mathcal{Q}$: $\left(\widehat{\mathcal{P}}, \widehat{\mathcal{Q}}\right) \xrightarrow{\text{lifted}} \left(\widehat{\mathcal{P}}, \mathcal{Q}\right)$

Before we prove our next theorem which considers simultaneous generalization over both distributions $\mathcal{P}$ and $\mathcal{Q}$, we first prove the following Lemma C.5. Notice when proving the final generalization theorem, we take the following natural two steps in lifting the guarantees:

$$\left(\widehat{\mathcal{P}}, \widehat{\mathcal{Q}}\right) \to \left(\mathcal{P}, \widehat{\mathcal{Q}}\right) \to (\mathcal{P}, \mathcal{Q})$$

The first step of the above scheme is exactly what we have proved in the Theorem for generalization over $\mathcal{P}$. However, we cannot directly invoke the other generalization Theorem to prove the second step since we had the empirical distribution of individuals – i.e. $\widehat{\mathcal{P}}$ – in that theorem. In the following Lemma, we basically prove the second step where the distribution over individuals is $\mathcal{P}$.

**Lemma C.5.** *Suppose*

$$n \geq \widetilde{O}\left(\frac{m\,d_{\mathcal{H}} + \log\left(1/\nu^2\delta\right)}{\alpha^2\beta^2}\right) \quad , \quad m \geq \widetilde{O}\left(\frac{\log\left(n\right)\log\left(n/\delta\right)}{\nu^4\alpha^4}\right)$$

*and let $\widehat{\psi} = \widehat{\psi}\left(X, \widehat{W}\right)$ be the output of Algorithm **AIF-Learn**. We have that for any $X$, with probability at least $1 - 2\delta$ over the problems F,*

$$\mathbb{P}_{x \sim \mathcal{P}}\left(\left|\mathcal{E}\left(x, \widehat{\psi}; \mathcal{Q}\right) - \mathcal{E}\left(x, \widehat{\psi}; \widehat{\mathcal{Q}}\right)\right| > \alpha\right) \leq \beta$$

*and that*

$$\left|err\left(\widehat{\psi}; \mathcal{P}, \mathcal{Q}\right) - err\left(\widehat{\psi}; \mathcal{P}, \widehat{\mathcal{Q}}\right)\right| \leq O\left(\alpha\beta\right) + O\left(\nu\alpha\right)$$

*Proof.* Let's prove the first part of the Lemma. Let $X' = \{x'_i\}_{i=1}^n \subseteq \mathcal{X}^n$ be – any – set of $n$ individuals. Observe that inequality 19 implies with probability $1 - \delta$ over the problems $F$, for all $i \in [n]$:

$$\left|\mathcal{E}\left(x'_i, \widehat{\psi}; \mathcal{Q}\right) - \mathcal{E}\left(x'_i, \widehat{\psi}; \widehat{\mathcal{Q}}\right)\right| \leq 2\sqrt{\frac{\log\left(4nT/\delta\right)}{2m_0}} \leq \nu\alpha$$

In other words, if $\widehat{\mathcal{P}}'$ represents the uniform distribution over $X'$, we have that for any $X' \subseteq \mathcal{X}^n$, with probability at least $1 - \delta$ over $F$,

$$\mathbb{P}_{x \sim \widehat{\mathcal{P}}'}\left(\left|\mathcal{E}\left(x, \widehat{\psi}; \mathcal{Q}\right) - \mathcal{E}\left(x, \widehat{\psi}; \widehat{\mathcal{Q}}\right)\right| > \alpha\right) = 0 \tag{25}$$

This doesn't give us what we want because this inequality works for any distribution with support of size at most $n$ while we want a guarantee for any $x \sim \mathcal{P}$ and that $\mathcal{P}$ might have infinite sized support. For random variables $F$, consider the event $A\left(F\right)$:

$$A\left(F\right) = \left\{\mathbb{P}_{x \sim \mathcal{P}}\left(\left|\mathcal{E}\left(x, \widehat{\psi}; \mathcal{Q}\right) - \mathcal{E}\left(x, \widehat{\psi}; \widehat{\mathcal{Q}}\right)\right| > \alpha\right) > \beta\right\}$$

We eventually want to show that $\mathbb{P}_F\left[A\left(F\right)\right]$ is small. Following the guarantee of 25 we consider an auxiliary data set of individuals $X'$ to argue that $\mathbb{P}_F\left[A\left(F\right)\right]$ is in fact small. To formalize our argument, let $X' = \{x'_i\}_{i=1}^n \sim \mathcal{P}^n$ be a new set of $n$ individuals drawn independently from the distribution $\mathcal{P}$ and let $\widehat{\mathcal{P}}'$ denote the uniform distribution over $X'$. Define new events $B\left(F, X'\right)$ and $C\left(F, X'\right)$ which depend on both $F$ and $X'$:

$B\left(F, X'\right) =$

$$\left\{\left|\mathbb{P}_{x \sim \mathcal{P}}\left(\left|\mathcal{E}\left(x, \widehat{\psi}; \mathcal{Q}\right) - \mathcal{E}\left(x, \widehat{\psi}; \widehat{\mathcal{Q}}\right)\right| > \alpha\right) - \mathbb{P}_{x \sim \widehat{\mathcal{P}}'}\left(\left|\mathcal{E}\left(x, \widehat{\psi}; \mathcal{Q}\right) - \mathcal{E}\left(x, \widehat{\psi}; \widehat{\mathcal{Q}}\right)\right| > \alpha\right)\right| > \beta\right\}$$

$$C\left(F, X'\right) = \left\{\mathbb{P}_{x \sim \widehat{\mathcal{P}}'}\left(\left|\mathcal{E}\left(x, \widehat{\psi}; \mathcal{Q}\right) - \mathcal{E}\left(x, \widehat{\psi}; \widehat{\mathcal{Q}}\right)\right| > \alpha\right) \neq 0\right\}$$

We have that

$$\begin{aligned}
\mathbb{P}_F\left[A\left(F\right)\right] &= \mathbb{P}_{F,X'}\left[A\left(F\right)\right] \\
&\leq \mathbb{P}_{F,X'}\left[B\left(F, X'\right)\right] + \mathbb{P}_{F,X'}\left[C\left(F, X'\right)\right] \quad \text{(follows from a triangle inequality)} \\
&= \mathbb{E}_F\left[\mathbb{P}_{X'}\left[B\left(F, X'\right)\right]\right] + \mathbb{E}_{X'}\left[\mathbb{P}_F\left[C\left(F, X'\right)\right]\right]
\end{aligned}$$

But inequality 25 implies for any $X'$, $\mathbb{P}_F\left[C\left(F, X'\right)\right] \leq \delta$. And that by a simple application of the Chernoff-Hoeffding's inequality, as long as the sample complexity for $n$ stated in the Lemma is met, we have that for any $F$, $\mathbb{P}_{X'}\left[B\left(F, X'\right)\right] \leq \delta$. We have therefore shown that

$$\mathbb{P}_F\left[A\left(F\right)\right] \leq 2\delta$$

proving the first part of the Lemma. The second part of the Lemma can be proved similarly with the same idea of considering an auxiliary data set $X' \sim \mathcal{P}^n$. Skipping the details and the formal proof,

we can in fact show that with probability $1 - 2\delta$ over $F$,

$$\left| \mathrm{err}\left(\widehat{\psi}; \mathcal{P}, \mathcal{Q}\right) - \mathrm{err}\left(\widehat{\psi}; \mathcal{P}, \widehat{\mathcal{Q}}\right) \right|$$

$$\leq \mathop{\mathbb{E}}_{x \sim \mathcal{P}} \left| \mathcal{E}\left(x, \widehat{\psi}; \mathcal{Q}\right) - \mathcal{E}\left(x, \widehat{\psi}; \widehat{\mathcal{Q}}\right) \right|$$

$$\leq \left| \mathop{\mathbb{E}}_{x \sim \mathcal{P}} \left| \mathcal{E}\left(x, \widehat{\psi}; \mathcal{Q}\right) - \mathcal{E}\left(x, \widehat{\psi}; \widehat{\mathcal{Q}}\right) \right| - \mathop{\mathbb{E}}_{x \sim \widehat{\mathcal{P}}'} \left| \mathcal{E}\left(x, \widehat{\psi}; \mathcal{Q}\right) - \mathcal{E}\left(x, \widehat{\psi}; \widehat{\mathcal{Q}}\right) \right| \right|$$

$$+ \mathop{\mathbb{E}}_{x \sim \widehat{\mathcal{P}}'} \left| \mathcal{E}\left(x, \widehat{\psi}; \mathcal{Q}\right) - \mathcal{E}\left(x, \widehat{\psi}; \widehat{\mathcal{Q}}\right) \right|$$

$$\leq O\left(\alpha\beta\right) + O\left(\nu\alpha\right)$$

$\square$

*Proof of Theorem 3.4 (Simultaneous Generalization over $\mathcal{P}$ and $\mathcal{Q}$).* First, observe that by a traingle inequality

$$\mathop{\mathbb{P}}_{x \sim \mathcal{P}} \left( \left| \mathcal{E}\left(x, \widehat{\psi}; \mathcal{Q}\right) - \widehat{\gamma} \right| > 6\alpha \right)$$

$$\leq \mathop{\mathbb{P}}_{x \sim \mathcal{P}} \left( \left| \mathcal{E}\left(x, \widehat{\psi}; \mathcal{Q}\right) - \mathcal{E}\left(x, \widehat{\psi}; \widehat{\mathcal{Q}}\right) \right| > \alpha \right) + \mathop{\mathbb{P}}_{x \sim \mathcal{P}} \left( \left| \mathcal{E}\left(x, \widehat{\psi}; \widehat{\mathcal{Q}}\right) - \widehat{\gamma} \right| > 5\alpha \right)$$

We will argue that as long as,

$$n \geq \widetilde{O}\left( \frac{m\, d_{\mathcal{H}} + \log\left(1/\nu^2\delta\right)}{\alpha^2\beta^2} \right) \quad , \quad m \geq \widetilde{O}\left( \frac{\log\left(n\right)\log\left(n/\delta\right)}{\nu^4\alpha^4} \right)$$

we can further bound the above inequality by $2\beta$. Lemma C.5 implies for any $X$, the first term appearing in the RHS of the above inequality is bounded by $\beta$ with probability $1 - 2\delta$ over the problems $F$. The second term is bounded by $\beta$ as well by the generalization over $\mathcal{P}$ theorem with probability $1 - 3\delta$ over the observed individuals and problems $(X, F)$ (in the theorem we actually state w.p. $1 - 6\delta$ because $3\delta$ comes from the accuracy guarantee). Putting these two pieces together, we have that with probability at least $1 - 5\delta$ over $(X, F)$,

$$\mathop{\mathbb{P}}_{x \sim \mathcal{P}} \left( \left| \mathcal{E}\left(x, \widehat{\psi}; \mathcal{Q}\right) - \widehat{\gamma} \right| > 6\alpha \right) \leq 2\beta$$

which implies $\widehat{\psi}$ is $(6\alpha, 2\beta)$-AIF with respect to the distributions $(\mathcal{P}, \mathcal{Q})$. Now let's look at the generalization error of $\widehat{\psi}$. Let

$$\psi^\star = \psi^\star\left(\alpha; \mathcal{P}, \mathcal{Q}\right), \quad \gamma^\star = \gamma^\star\left(\alpha; \mathcal{P}, \mathcal{Q}\right), \quad \mathrm{err}\left(\psi^\star; \mathcal{P}, \mathcal{Q}\right) = \mathrm{OPT}\left(\alpha; \mathcal{P}, \mathcal{Q}\right)$$

We will again use the triangle inequality to write:

$$\left| \mathrm{err}\left(\widehat{\psi}; \mathcal{P}, \mathcal{Q}\right) - \mathrm{err}\left(\widehat{\psi}; \widehat{\mathcal{P}}, \widehat{\mathcal{Q}}\right) \right| \leq \left| \mathrm{err}\left(\widehat{\psi}; \mathcal{P}, \mathcal{Q}\right) - \mathrm{err}\left(\widehat{\psi}; \mathcal{P}, \widehat{\mathcal{Q}}\right) \right|$$

$$+ \left| \mathrm{err}\left(\widehat{\psi}; \mathcal{P}, \widehat{\mathcal{Q}}\right) - \mathrm{err}\left(\widehat{\psi}; \widehat{\mathcal{P}}, \widehat{\mathcal{Q}}\right) \right|$$

It follows from Lemma C.5 that the first term appearing in the RHS is bounded by $O\left(\alpha\beta\right) + O\left(\nu\alpha\right)$ with probability $1 - 2\delta$ over the problems $F$, and that the second term (see 17) is bounded by $O\left(\alpha\beta\right)$ as well with probability $1 - \delta$ over the individuals $X$. Therefore, with probability $1 - 3\delta$ over the individuals $X$ and problems $F$,

$$\left| \mathrm{err}\left(\widehat{\psi}; \mathcal{P}, \mathcal{Q}\right) - \mathrm{err}\left(\widehat{\psi}; \widehat{\mathcal{P}}, \widehat{\mathcal{Q}}\right) \right| \leq O\left(\alpha\beta\right) + O\left(\nu\alpha\right)$$

It follows similarly that with probability $1 - 2\delta$ over $X$ and $F$,

$$\left| \mathrm{err}\left(\psi^\star; \mathcal{P}, \mathcal{Q}\right) - \mathrm{err}\left(\psi^\star; \widehat{\mathcal{P}}, \widehat{\mathcal{Q}}\right) \right| \leq O\left(\nu\alpha\right) + O\left(\alpha\beta\right)$$

We can also show that with probability $1 - \delta$ over $F$, the pair $(\psi^\star|_F, \gamma^\star)$ – where $\psi^\star|_F$ is the mapping $\psi^\star$ restricted to the domain $F$ – is feasible in the empirical fair learning problem, which again implies by Theorem A.3 that with probability $1 - \delta$ over $F$

$$\mathrm{err}\left(\widehat{\psi}; \widehat{\mathcal{P}}, \widehat{\mathcal{Q}}\right) = \mathrm{err}\left(\widehat{p}; \widehat{\mathcal{P}}, \widehat{\mathcal{Q}}\right) \leq \mathrm{err}\left(\psi^\star|_F; \widehat{\mathcal{P}}, \widehat{\mathcal{Q}}\right) + 2\nu = \mathrm{err}\left(\psi^\star; \widehat{\mathcal{P}}, \widehat{\mathcal{Q}}\right) + 2\nu$$

Putting all inequalities together, we have that with probability $1 - 7\delta$ over $(X, F)$,

$$\mathrm{err}\left(\widehat{\psi}; \mathcal{P}, \mathcal{Q}\right) \leq \mathrm{OPT}\left(\alpha; \mathcal{P}, \mathcal{Q}\right) + O\left(\nu\right) + O\left(\alpha\beta\right)$$

$\square$

## C.3 Missing Proofs: Learning subject to FPAIF

### C.3.1 Algorithm analysis and in-sample guarantees

*Proof of Lemma B.1.* Recall the Learner is accumulating regret only because they are using the given quantities $\{\widetilde{\rho}_i\}_{i=1}^n$ instead of $\{\widehat{\rho}_i\}_{i=1}^n$ that depend on the observed data $F$. In other words, at round $t \in [T]$ of the algorithm instead of best responding to

$$
\mathcal{L}\left(\boldsymbol{h}, \gamma, \boldsymbol{\lambda}_t\right) = -\gamma \sum_{i=1}^n w_{i,t}
$$
$$
+ \frac{1}{m} \sum_{j=1}^m \left\{ \sum_{i=1}^n \left( \frac{1}{n} + \frac{w_{i,t}}{\widehat{\rho}_i} \right) \mathbb{1}\left[h_j(x_i) = 1, f_j(x_i) = 0\right] + \left(\frac{1}{n}\right) \mathbb{1}\left[h_j(x_i) = 0, f_j(x_i) = 1\right] \right\}
$$

The Learner is in fact minimizing

$$
\widetilde{\mathcal{L}}\left(\boldsymbol{h}, \gamma, \boldsymbol{\lambda}_t\right) = -\gamma \sum_{i=1}^n w_{i,t}
$$
$$
+ \frac{1}{m} \sum_{j=1}^m \left\{ \sum_{i=1}^n \left( \frac{1}{n} + \frac{w_{i,t}}{\widetilde{\rho}_i} \right) \mathbb{1}\left[h_j(x_i) = 1, f_j(x_i) = 0\right] + \left(\frac{1}{n}\right) \mathbb{1}\left[h_j(x_i) = 0, f_j(x_i) = 1\right] \right\}
$$

Observe that for any pair of $\left(\boldsymbol{h}, \gamma\right)$,

$$
\left| \mathcal{L}\left(\boldsymbol{h}, \gamma, \boldsymbol{\lambda}_t\right) - \widetilde{\mathcal{L}}\left(\boldsymbol{h}, \gamma, \boldsymbol{\lambda}_t\right) \right| \leq \frac{1}{m} \sum_{j=1}^m \sum_{i=1}^n \left| \frac{w_{i,t}}{\widetilde{\rho}_i} - \frac{w_{i,t}}{\widehat{\rho}_i} \right| \mathbb{1}\left[h_j(x_i) = 1, f_j(x_i) = 0\right]
$$
$$
\leq \sum_{i=1}^n \left| \frac{w_{i,t}}{\widetilde{\rho}_i} - \frac{w_{i,t}}{\widehat{\rho}_i} \right| \frac{1}{m} \sum_{j=1}^m \mathbb{1}\left[f_j(x_i) = 0\right]
$$
$$
= \sum_{i=1}^n \widehat{\rho}_i \left| \frac{w_{i,t}}{\widetilde{\rho}_i} - \frac{w_{i,t}}{\widehat{\rho}_i} \right|
$$
$$
\leq \sum_{i=1}^n \frac{|\widehat{\rho}_i - \widetilde{\rho}_i| \cdot |w_{i,t}|}{\widetilde{\rho}_i}
$$

A simple application of Chernoff-Hoeffding's inequality (Theorem C.1) implies with probability $1 - \delta$ over the observed labelings $F$, uniformly for all $i$,

$$
|\widehat{\rho}_i - \rho_i| \leq \sqrt{\frac{\log\left(2n/\delta\right)}{2m_0}}
$$

On the other hand by Assumption B.1, we have that for all $i \in [n]$,

$$
|\widetilde{\rho}_i - \rho_i| \leq \sqrt{\frac{\log\left(n\right)}{2m_0}}
$$

Therefore for a given $t$, with probability $1 - \delta$ over $F$,

$$
\left| \mathcal{L}\left(\boldsymbol{h}, \gamma, \boldsymbol{\lambda}_t\right) - \widetilde{\mathcal{L}}\left(\boldsymbol{h}, \gamma, \boldsymbol{\lambda}_t\right) \right| \leq \frac{2}{\widetilde{\rho}_{\min}} \sqrt{\frac{\log\left(2n/\delta\right)}{2m_0}} \sum_{i=1}^n |w_{i,t}|
$$
$$
= \frac{2}{\widetilde{\rho}_{\min}} \sqrt{\frac{\log\left(2n/\delta\right)}{2m_0}} \sum_{i=1}^n |\lambda_{i,t}^+ - \lambda_{i,t}^-|
$$
$$
\leq \frac{2B}{\widetilde{\rho}_{\min}} \sqrt{\frac{\log\left(2n/\delta\right)}{2m_0}}
$$

Now, let $\left(\boldsymbol{h}_t^\star, \gamma_t^\star\right) = \arg\min_{\boldsymbol{h},\gamma} \mathcal{L}\left(\boldsymbol{h},\gamma,\boldsymbol{\lambda}_t\right)$ be the best response of the Learner to $\mathcal{L}$ and recall the actual play of the Learner is in fact $\left(\boldsymbol{h}_t,\gamma_t\right) = \arg\min_{\boldsymbol{h},\gamma} \widetilde{\mathcal{L}}\left(\boldsymbol{h},\gamma,\boldsymbol{\lambda}_t\right)$. Observe that the above inequality implies for any $t$, with probability $1-\delta$ over $F$,

$$\mathcal{L}\left(\boldsymbol{h}_t,\gamma_t,\boldsymbol{\lambda}_t\right) \leq \widetilde{\mathcal{L}}\left(\boldsymbol{h}_t,\gamma_t,\boldsymbol{\lambda}_t\right) + \frac{2B}{\widetilde{\rho}_{\min}}\sqrt{\frac{\log\left(4n/\delta\right)}{2m_0}}$$

$$\leq \widetilde{\mathcal{L}}\left(\boldsymbol{h}_t^\star,\gamma_t^\star,\boldsymbol{\lambda}_t\right) + \frac{2B}{\widetilde{\rho}_{\min}}\sqrt{\frac{\log\left(4n/\delta\right)}{2m_0}}$$

$$\leq \mathcal{L}\left(\boldsymbol{h}_t^\star,\gamma_t^\star,\boldsymbol{\lambda}_t\right) + \frac{4B}{\widetilde{\rho}_{\min}}\sqrt{\frac{\log\left(4n/\delta\right)}{2m_0}}$$

It follows immediately that with probability $1-\delta$ over the observed labelings $F$, for all $t \in [T]$

$$\mathcal{L}\left(\boldsymbol{h}_t,\gamma_t,\boldsymbol{\lambda}_t\right) - \min_{\boldsymbol{p},\gamma}\mathcal{L}\left(\boldsymbol{p},\gamma,\boldsymbol{\lambda}_t\right) \leq \frac{4B}{\widetilde{\rho}_{\min}}\sqrt{\frac{\log\left(4nT/\delta\right)}{2m_0}}$$

which completes the proof. $\square$

*Proof of Lemma B.2.* The proof is similar to that of Lemma A.1. The terms

$$\frac{B\log\left(2n+1\right)}{\eta T} + \eta\left(1+2\alpha\right)^2 B$$

come from the regret analysis of the exponentiated gradient descent algorithm. The only difference is that now we have to bound the difference between false positive rates, one with respect to the distribution $\widehat{\mathcal{Q}}$ and another with respect to the distribution $\widehat{\mathcal{Q}}_t$, for all $i$ and $t$:

$$\left|\mathcal{E}_{\mathrm{FP}}\left(x_i,\widehat{\psi}_t;\widehat{\mathcal{Q}}\right) - \mathcal{E}_{\mathrm{FP}}\left(x_i,\widehat{\psi}_t;\widehat{\mathcal{Q}}_t\right)\right|$$

Let $\widehat{\rho}_{t,i} = \mathbb{P}_{f\sim\widehat{\mathcal{Q}}_t}\left[f(x_i)=0\right]$ and recall $\widehat{\rho}_i = \mathbb{P}_{f\sim\widehat{\mathcal{Q}}}\left[f(x_i)=0\right]$. We have that

$$\mathcal{E}_{\mathrm{FP}}\left(x_i,\widehat{\psi}_t;\widehat{\mathcal{Q}}\right) = \frac{\mathbb{E}_{f\sim\widehat{\mathcal{Q}}}\left[\mathbb{P}_{h\sim\widehat{\psi}_{t,f}}\left[h(x_i)=1,\ f(x_i)=0\right]\right]}{\widehat{\rho}_i}$$

$$\mathcal{E}_{\mathrm{FP}}\left(x_i,\widehat{\psi}_t;\widehat{\mathcal{Q}}_t\right) = \frac{\mathbb{E}_{f\sim\widehat{\mathcal{Q}}_t}\left[\mathbb{P}_{h\sim\widehat{\psi}_{t,f}}\left[h(x_i)=1,\ f(x_i)=0\right]\right]}{\widehat{\rho}_{t,i}}$$

And therefore

$$\left|\mathcal{E}_{\mathrm{FP}}\left(x_i,\widehat{\psi}_t;\widehat{\mathcal{Q}}\right) - \mathcal{E}_{\mathrm{FP}}\left(x_i,\widehat{\psi}_t;\widehat{\mathcal{Q}}_t\right)\right|$$

$$\leq \frac{1}{\widehat{\rho}_i}\left\{\left|\mathbb{E}_{f\sim\widehat{\mathcal{Q}}}\left[\mathbb{P}_{h\sim\widehat{\psi}_{t,f}}\left[h(x_i)=1,\ f(x_i)=0\right]\right] - \mathbb{E}_{f\sim\widehat{\mathcal{Q}}_t}\left[\mathbb{P}_{h\sim\widehat{\psi}_{t,f}}\left[h(x_i)=1,\ f(x_i)=0\right]\right]\right| + \left|\widehat{\rho}_i - \widehat{\rho}_{t,i}\right|\right\}$$

This inequality follows from the simple fact that for $a \leq b$ and $c \leq d$ all in $\mathbb{R}_+$:

$$\left|\frac{a}{b}-\frac{c}{d}\right| = \left|\frac{a}{b}-\frac{c}{b}+\frac{c}{b}-\frac{c}{d}\right| \leq \frac{1}{b}\left|a-c\right| + c\left|\frac{1}{b}-\frac{1}{d}\right| \leq \frac{1}{b}\left|a-c\right| + \frac{1}{b}\left|b-d\right|$$

Now by viewing $F_t$ as a random sample from $F$, we can use the Chernoff-Hoeffding's inequality to bound each difference appearing above separately. We therefore have that with probability $1-\delta$ over the draws of $F_t$ from $F$:

$$\left|\mathcal{E}_{\mathrm{FP}}\left(x_i,\widehat{\psi}_t;\widehat{\mathcal{Q}}\right) - \mathcal{E}_{\mathrm{FP}}\left(x_i,\widehat{\psi}_t;\widehat{\mathcal{Q}}_t\right)\right| \leq \frac{2}{\widehat{\rho}_i}\sqrt{\frac{\log\left(4/\delta\right)}{2m_0}}$$

The rest is similar to how the proof for Lemma A.1 proceeds. $\square$

(a) convergence plot: synthetic data          (b) error spread plot: synthetic data

Figure 1: (a) Error-unfairness trajectory plots illustrating the convergence of algorithm **AIF-Learn**. (b) Error-unfairness tradeoffs and individual errors for **AIF-Learn** vs. simple mixtures of the error-optimal model and random classification. Gray dots are shifted upwards slightly to avoid occlusions.

## D    Additional experimental results for AIF-Learn

We start this section by providing more details of **AIF-Learn**'s implementation. First, since our interest is to evaluate the *in-sample* performance of the algorithm, we obviously avoid partitioning the problems $F$ into small batches which was rather a theoretical artifact of our proof for generalization over the problem generating distribution $\mathcal{Q}$. We therefore use in practice all learning tasks to update the fairness violation vector in each round of the algorithm. As also discussed in the body of paper, we implement the learning oracle assumed by our algorithm using linear threshold heuristics — identical to Kearns et al. (2018). To elaborate, we build linear regression models for predicting the cost vectors $C_0 \in \mathbb{R}^n$ and $C_1 \in \mathbb{R}^n$ and classify an individual according to the lowest predicted cost by the regression models. Suggested by our theory, we set $B = 1/\alpha$ and $\eta = c/B$ in our implementation where the constant $c$ can greatly affect the convergence speed of the algorithm. In the setting we considered in our experiments — i.e. $n = 200$ individuals, $m = 50$ problems, $d = 20$ features — it turns out choosing $c$ in the order of 100 would lead to convergence with at most $T = 1000$ iterations of the algorithm.

In addition to the experiments on the Communities and Crime data set presented in the body of the paper, we ran additional experiments on synthetic data sets to further verify the performance of our algorithms empirically. Given the parameters $n$ (number of individuals), $m$ (number of problems), $d$ (dimension), and a coin with bias $q \geq 0.5$, an instance of the synthetic data is generated as follows:

First $n$ individuals $\{x_i\}_{i=1}^n$ are generated randomly where $x_i \in \{\pm 1\}^d$. Let the first 75% of the individuals be called "group 1" and call the rest "group 2". For each learning task $j \in [m]$, we sample two weight vectors $w_{j,\text{maj}}, w_{j,\text{min}} \in \mathbb{R}^d$. Now consider a matrix $Y \in \{0,1\}^{n \times m}$ that needs to be filled in with labels. We intentionally inject unfairness into the data set (because otherwise fairness would be achieved for free due to the random generation of data) by considering "majority" and "minority" groups for each learning task (majority group will be labelled using $w_{j,\text{maj}}$ and the minority by $w_{j,\text{min}}$). For each learning task $j$, individuals in group 1 get $q$ chance of falling into the majority group of that task and individuals in group 2 get $1 - q$ chance (this is implemented by flipping a coin with bias $q$ for each individual). Once the majority/minority groups for each task are specified, the labels $Y = [y_{ij}]$ are set according to $y_{ij} = \text{sign}(w_j^T x_i)$ where $w_j \in \{w_{j,\text{maj}}, w_{j,\text{min}}\}$. (Notice $q = 1$ corresponds to a fixed (across learning tasks) majority group of size $0.75n$, and $q = 0.5$ corresponds to an instance of the data set where the majority groups are formed completely at random and that they have size $0.5n$ in expectation.)

It is "expected" that solving the unconstrained learning problem on an instance of the above synthetic data generation process would result in discrimination against the minority groups which are formed mostly by the individuals in group 2, and therefore our algorithm should provides a fix for this unfair treatment. We therefore took an instance of the synthetic data set with parameters $n = 200, m = 50, d = 20, q = 0.8$ and ran our algorithm on varying values of allowed fairness violation $2\alpha$ (notice on an input $\alpha$ to the algorithm, the individual error rates are allowed to be within at most $2\alpha$ of each

other). We present the results (which are reported in-sample) in Figure 1 where similar to the results on the Communities and Crime data set, we get convergence after $T = 1000$ iterations (or 50,000 calls to the learning oracle, see Figure 1(a)). Furthermore, Figure 1(b) suggests that our algorithm on the synthetic data set considerably outperforms the baseline that only mixes the error-optimal model with random classification (i.e. one that labels 0 or 1 with probability 0.5).