[Reviews · NeurIPS 2019]

Reviewer 1



The central idea of this work, that it is easier to define individual fairness when individuals receive many decisions instead of a single classification is a solid idea, but it is not exactly new. The earliest paper I can find now that proposes a notion of individual fairness based on this idea is Zhang and Shah 2014 (http://papers.nips.cc/paper/5588-fairness-in-multi-agent-sequential-decision-making.pdf), but even the work by Joseph et al. (2016, 2018) cited is loosely an example of this. The motivation actually given is, however, less than satisfactory. The example cited Kearns et al. (2018) is best addressed not by this work, but by that paper cited (and on this subject, I don't love using two real genders but two fake races). The real reason is not that you forget about a sensitive attribute and therefore harm that population, but because the guarantees become trivial as the complexity of the set of attributes go to infinity. Here, on the other hand, guarantees need not do this, as the utility the individuals receive is not binary. So, I really see the proposed definition as closer to group fairness with group sizes of one than the individual fairness of Dwork et al. (2012). This outlook can help to solve the following issue, which is that I'm highly skeptical of introducing a new fairness definition for computational conveniences/ assumptions, such as realizability assumptions cited as a reason for this new fairness definition. It should therefore be pointed out that the ethical motivation here is (basically) the same as in group fairness, in a setting that allows for a simpler formulation. Nonetheless, this is a particularly clean formulation of this basic idea. It's also a well-written paper and while I did not have a chance to check the proofs, the results make sense. Edit: I am satisfied with the response to this review.

Reviewer 2



This paper presents an original re-framing of the error classification as performed over a random class of classification problems, a re-application of existing methods in a new context to solve it, and an empirical analysis to justify that analysis. The key concept, of re-framing the classification task as being random, represents a novel intuition and they immediately pull fruitful analysis from it. The quality is high; the technique and proofs appear to be sound, the experiments are clear, and the supporting code appears well organized (though I haven't run it). The clarity is high, with clear notation and writing. It is easy to follow, despite a rather abstract problem formulation. My only nitpick is the font sizes and reliance on color in Figure 2. The significance is moderately high; the approach is an intuitive improvement as a fairness metric, and the provided code increases the extensibility. I expect this will be an important comparison method for future approaches to generalizable fairness. After review of the author's/authors' rebuttal, I still believe this is a top 50% paper.

Reviewer 3



Edits to review: I have updated my score to an 8. I recommend and argue for acceptance! It's a very nice paper that makes a significant conceptual contribution. I think some version of the plot included in the rebuttal highlighting the empirical generalization behavior should be included in the main body of the paper (even if that means moving the convergence plot to the appendix). I would personally be interested in seeing the results on the synthetic task, especially if you could push the limits of what happens with lots of data (individuals and problems), but understand that that may be personal preference. If it can be included in the appendix, I think it would be great. Discussion of the plot of generalization errors should also be prioritized in my opinion. In particular, even though "the gap in error is substantially smaller than predicted by our theory" it is still quite significant. In particular, what does it mean to have a fairness violation above 0.5? Does this mean that some individuals are experiencing error rates *worse than random guessing*? (I'm now confused by 2*alpha vs. 4*alpha. As a side note, simplifying the use of c*alpha for different c would greatly help the interpretability.) Would it be possible to reproduce Figure 2(b) from the main paper for the generalization errors as well? I found this (and the corresponding appendix plot) to communicate the most information about what is happening with the learned models. I can't tell from the plot given in the rebuttal, but it would be quite compelling if the generalization in overall and individual error rates performs significantly better than interpolation with random (as in the train error rates). Even if it doesn't perform significantly better, it would be interesting to know and to speculate what would be necessary to improve things. Some technical issues/questions: - I do not understand the legend of the plot. What does generalization "over individuals" mean? Sampling new individuals on the same tasks? same for "over problems" - "the ordering on the Pareto frontier is the same in testing as in training" <-- can I read this off from the plot or is this an additional claim? Original review: High-level: Overall, this paper is well-written and clear in its presentation. The theoretical results are novel, very interesting, and I have high confidence that they are correct. The problem framing is completely novel; the work manages to apply classical techniques to analyze their problem. This is a strength, not a drawback. The empirical evaluation has many issues and hurts the paper. More detailed: This work presents a very intriguing approach towards practical notions of "individual fairness". Thus far, the research into individual fairness initiated in "Fairness through Awareness" [Dwork et al.] has remained a purely theoretical program, due to serious technical barriers (requires a similarity metric over all individuals). Thinking about individual *rates* in terms of a distribution over *problem instances* is an ingenious and truly original idea that seems to make a large conceptual leap forward in terms of providing individual-level guarantees. While the novel notion of individual error rates is intriguing, the way the work defines average individual fairness (AIF) raises some subtle conceptual concerns. AIF asks for parity of individual error rates (or FPR/FNRs) across all (in fact, most 1-beta fraction of) individuals. The work does not justify why *parity* is the best notion of fairness in this context. As a classified individual who receives a suboptimal error rate, do I feel slighted because someone else received a different error rate or because the classifier performed poorly on me? It seems to me that upper bounding the error rate experienced by (most) individuals, rather than insisting on parity, seems as (or more) justifiable in many cases. If I understand correctly, all of the results would port over to an error upper bound rather than parity, and one could search over error upper bounds for suitable models. Of course not all upper bounds will be feasible, so perhaps you would need to plot a pareto curve in terms of error tolerance and fraction of individuals achieving the tolerance. The evaluation in Section 4 / Appendix highlights this point quite well. Consider a hypothetical where individuals experience harm if their error rates are more than 0.3. By this definition, it would seem that the unconstrained predictors would be one of the most fair of the predictors evaluated in the experiments (minimizing the number of individuals experiencing harm). In particular, the most fair'' according the AIF violation makes almost 0.4 error on every individual. As the work points out, at the extreme, random guessing is AIF. I think this point deserves more discussion. In particular, it seems weird that "fairness" is achieved by significantly increasing the error rates on the vast majority of individuals rather than improving the error rates on most individuals. One other minor conceptual issue that I don't think is adequately addressed: the entire approach of generalization over new problems relies upon getting a new labeling of the *same* individuals from the original training set. I'm sure there are many settings where this is highly motivated, but it should be discussed further. It is definitely a limitation to the applicability that is not appropriately communicated. Modulo these conceptual concerns that I think should be addressed, this is a very nice theoretical paper. The experimental section, however, is inadequate and raised serious concerns for me. Plainly stated, the experiments do not evaluate the main contribution of this paper. The point of this work is a *generalization* bound over individuals and problems. The experiments only report train error, rendering them meaningless to evaluate the practical validity of the theoretical contributions. As a pair of concrete concerns from translating the theory to practice: the number of individuals depends *polynomially* on 1/beta (where beta = fraction of individuals for whom the AIF guarantee may be violated) rather than logarithmically (as we might expect in standard uniform convergence bounds); and the number of problem instances depends as 1/alpha^4nu^4 for 4*alpha=AIF violation and O(nu)=optimality of error rate. Even for trivial settings of the parameters, e.g. both alpha=0.25 and nu=0.5, it seems we would need thousands of problem instances before we get any guarantees on generalization. Point being, I disagree that these experiments "empirically verify [the] effectiveness" of the approach. Since one of the experimental setups is purely synthetic, it seems like it would be completely routine to evaluate the generalization behavior in this setup. I am very confused why these validation numbers not reported. I also do not find that the plots chosen communicate the right information. It has been established by the works of [Kearns et al.]^2 on "Preventing Fairness Gerrymandering" that oracle-efficient reductions + best respond dynamics converge nicely to stable solutions, so I do not find that Figure 1(a) communicates much new information. Am I missing something significant from this plot? Of course, the concern when validation error is not reported is overfitting. To get at this question, I'm curious whether it is possible to investigate the mixture of hypotheses learned by the algorithm. In particular, in the synthetic case, does the mixture include w_maj / w_min? A 50:50 mixture of these two hypotheses would achieve error rates of about 0.25. It seems that if the groups 1 and 2 are truly chosen at random (as in the synthetic setup), then there should be no advantage over this mixed strategy. In this sense, it might be an interesting way to address the question of overfitting: if the learned hypothesis is correlated with the group labels (i.e. g1 = +1, g2=-1), then it suggests overfitting is occurring.

[Author Response · NeurIPS 2019]

Thanks for the reviews! We take your comments to heart and will make all of the small changes suggested. Here we address the more major concerns.

**Experiments**   First, we agree with reviewer 3 that the main contribution of our paper is conceptual and theoretical: in particular, identifying randomness over the problem distribution as something that can be leveraged to give an individual level guarantee, and then deriving the algorithms and generalization bounds needed to actually realize this guarantee. The purpose of the experiments is modest: to support the theory. Because of this, we chose to illustrate experimentally the one thing that is not actually guaranteed by the theory: namely, the convergence of the algorithm. The theorem we prove only guarantees that the algorithm converges *if we have a cost sensitive classification oracle*. In practice, we do not: we use a regression heuristic — and so we chose the experiment we did to confirm that we get convergence, despite the gap between theory and practice. Reviewer 3 is correct that prior work — in particular, [Kearns et al.] — has shown with similar experiments that *other* oracle efficient algorithms converge in practice with heuristics. But that does not imply that ours will, since our algorithm differs both in the specifics of the dynamics (fictitious play in [Kearns et al.], no-regret vs. best response in our paper) and in the particular problems we are asking the classification oracles to solve. In short, because the framework of "oracle efficiency" leaves a gap between theory and practice, we think of it as good practice to accompany a proof of oracle efficiency with an empirical validation of convergence whenever possible.

That being said, Reviewer 3 is of course correct that empirically examining generalization is an interesting thing to do: we didn't do it in the submission simply because we thought it was *less* interesting than convergence (generalization is guaranteed by our theorems, without any heuristic assumptions) and were mindful of space constraints. But we have now quickly adapted our code to investigate our generalization error, both across data points and across problems. The plot is below:

To be consistent with our paper, we trained on exactly the same subset of Communities and Crime that we did in our paper ($n = 200$ datapoints, $m = 50$ problems (selected features from the dataset)). Thus the curve labelled "training" is the same as the reported in-sample results in our paper. We used a fresh holdout consisting of $n = 200$ datapoints, and $m = 25$ problems (features from the dataset that weren't previously used) to evaluate our generalization performance over both problems and data points, in terms of both accuracy and fairness violation. Two things stand out:

communities dataset (varying $\alpha = 0.1, 0.11, ..., 0.22$)

1. As predicted by the theory, our test curves track our training curves, but with higher error and unfairness. In particular, the ordering of the models on the Pareto frontier is the same in testing as in training, meaning that the training curve can indeed be used to manage the trade-off out-of-sample as well.

2. The gap in error is substantially smaller than would be predicted by our theory: since our training set is so small, our theoretical guarantees are vacuous, but all points plotted in our test pareto curves are non-trivial in terms of both accuracy and fairness. Presumably the gap in error would narrow on larger training sets. If the paper is accepted and the reviewers feel that generalization experiments should be included, we will make room to do so (and re-run the above experiment on a larger training set, which we can easily do with the luxury of time). Similarly, if the reviewers still feel that the experiments detract from the theory, we are also willing to relegate all experiments to the supplemental material and focus on the main contributions in the body.

**Other Conceptual Questions**

**Reviewer 1:** The group-fairness proposal of [Kearns et al.] indeed mitigates the "gerrymandering" concern we cite from their work, but ultimately does not eliminate it because it remains a group-level constraint that only holds for groups that are sufficiently large. With our approach, we are able to reduce these groups to size 1, which makes it an individual level constraint. (We agree this is different than "individual fairness" in Dwork et al. – we use the term more broadly, and will clarify).

**Reviewer 3:** We agree that asking for parity of error statistics is not always the right approach, and that our techniques generalize to asking for upper bounds on error rates. We consider equalizing error rates mostly as a canonical example of a popular approach; we will clarify. We will also further emphasize the limitation of needing to be able to observe labels for different problems on the same training set.

[Meta-Review · NeurIPS 2019]

Reviewers found this paper to be a clever and elegant model of fairness with respect to a distribution over individuals and classification tasks.